# Improving MoE Performance and Efficiency with Plug-and-Play Intra-Layer Specialization and Cross-Layer Coupling Losses

## Abstract

Sparse Mixture-of-Experts (MoE) models scale Transformers efficiently but suffer from expert overlap—redundant representations across experts and routing ambiguity, resulting in severely underutilized model capacity. While architectural solutions like DeepSeekMoE promote specialization, they require substantial structural modifications and rely solely on intra-layer signals. In this paper, we propose two plug-and-play auxiliary losses that enhance MoE specialization and routing efficiency without modifying routers or model architectures. First, an intra-layer specialization loss penalizes cosine similarity between experts' SwiGLU activations on identical tokens, encouraging experts to specialize in complementary knowledge. Second, a cross-layer coupling loss maximizes joint Top-$k$ routing probabilities across adjacent layers, establishing coherent expert pathways through network depth while reinforcing intra-layer specialization. Both losses are orthogonal to the standard load-balancing loss and compatible with shared-expert in DeepSeekMoE and vanilla Top-$k$ MoE architectures. We implement both losses as a drop-in Megatron-LM module. Extensive experiments across pre-training, fine-tuning, and zero-shot benchmarks demonstrate consistent task gains, higher expert specialization, and lower-entropy routing; together, these improvements translate into faster inference via more stable expert pathways.

## 1 Introduction

Sparse Mixture-of-Experts (MoE) has emerged as a standard approach for scaling Transformers by expanding parameters while keeping per-token compute roughly constant (Shazeer et al., 2017; Jacobs et al., 1991). In MoE, a learned router activates only a small subset of experts—typically feed-forward networks—for each token (Fedus et al., 2022). From early sparsely gated layers to modern large language models (Du et al., 2022; Fedus et al., 2022; Lepikhin et al., 2020; Zoph et al., 2022; Dai et al., 2024), this design has delivered strong accuracy–efficiency trade-offs. Nevertheless, a fundamental challenge remains: expert specialization progressively deteriorates during training, with tokens routed to different experts exhibiting excessive uniformity and overlap, leading multiple experts to learn redundant knowledge (Dai et al., 2024). This redundancy confronts the router with ambiguous decisions among functionally equivalent experts, eroding token-to-expert boundaries and substantially underutilizing model capacity.

Recent work has sought to encourage specialization through architectural modifications. DeepSeek-MoE (Dai et al., 2024), for example, introduces always-active shared experts to handle common patterns, allowing routed specialists to focus on more fine-grained tasks. Heterogeneous Mixture of Experts (HMoE) (Wang et al., 2024) and Mixture of Diverse Size Experts (MoDSE) (Sun et al., 2024) employ variable-sized experts within each layer: HMoE favors more frequent activation of smaller experts to better match token complexity, while MoDSE distributes diverse-sized experts across GPUs to balance load. Other variants adjust layer composition, expert granularity, or routing mechanics with scale and efficiency as primary goals. Notable examples include Mixtral (Jiang et al., 2024) (top-2 routed FFNs at scale), Mixture of a Million Experts (He et al., 2024) (extreme fine-grained expertization), and ReMoE (Wang et al., 2025b) (a differentiable ReLU-based router that enables continuous sparsity control).

In contrast to architectural modifications, this paper takes an orthogonal perspective: *treating expert specialization as a primary training objective rather than a structural property*. This training-loss-centric approach complements the aforementioned architectural solutions by directly supervising expert behavior through targeted loss functions, independent of the underlying MoE architectures. To design these training losses, we identify two modes in which expert specialization fails: (1) **Expert Overlap**, where different experts produce nearly identical activations for the same tokens, yielding redundant representations; and (2) **Routing Ambiguity**, where similar inputs are inconsistently dispatched across different experts, revealing ill-defined routing rules. When either occurs, experts collapse toward overlapping knowledge while the router confronts ambiguous choices among functionally equivalent experts, undermining MoE's core principle of specialization.

To address these failures, we introduce two complementary loss functions that work in concert:

1. **Intra-Layer Specialization Loss:** This loss penalizes high cosine similarity between the activations of different experts for the same token. It directly discourages functional redundancy and pushes each expert within a layer to develop its own unique specialization.

2. **Cross-Layer Coupling Loss:** This loss promotes coherent routing across adjacent layers by maximizing the joint probability of top-ranked expert pairs. By encouraging tokens to follow consistent sequences of experts through depth—referred to as *expert paths*—it sharpens routing distributions, reduces ambiguity, and lowers routing entropy.

Together, these loss functions translate our diagnosed failure modes into targeted supervision, producing experts that are both functionally distinct within layers and coherently utilized across them.

Our theoretical analysis establishes both the effectiveness and compatibility of our proposed losses. For effectiveness, we show that the intra-layer specialization loss induces nearly orthogonal expert activations, resulting in orthogonal parameter gradients that drive distinct learning trajectories for each expert. Additionally, we demonstrate that cross-layer coupling amplifies intra-layer specialization through high activation correlations between adjacent layers, enabling specialization to propagate through network depth. For compatibility, we justify that both losses are compatible with the load-balancing objectives commonly used in MoE training.

Extensive experiments demonstrate significant improvements in both model performance and system efficiency. Our approach consistently enhances model performance across parameter scales, reducing pre-training perplexity and improving results on diverse downstream tasks including fine-tuning and zero-shot benchmarks. These gains stem from demonstrably higher expert specialization and more decisive, lower-entropy routing distributions. Furthermore, the improved specialization translates directly into system-level efficiency: stable token-expert pathways enhance cache utilization and batching during inference, yielding higher throughput without architectural modifications. In the pre-training task, our method reduces the validation perplexity by 0.7% to 1.9%. For the supervised fine-tuning task on the Qwen3-30B-A3B-Instruct-2507 model, we achieve consistent performance improvements across four datasets with an average gain of 1.4%.

Our contributions are summarized as follows:

- **A New Perspective on Specialization.** We propose a training-loss-centric approach to MoE specialization that targets two failure modes: expert overlap and routing ambiguity. To counteract them, we introduce an Intra-Layer Specialization Loss to discourage representational overlap and a Cross-Layer Coupling Loss to build coherent routing paths.

- **Theoretical Guarantees.** Our theoretical analysis validates the effectiveness and compatibility of our losses. We show they encourage distinct expert learning trajectories through near-ortoghonal gradients and allow specialization to propagate through the network. Moreover, we justify that both losses work in concert with standard load-balancing objectives.

- **Consistent Accuracy and Efficiency Gains.** Our method yields consistent gains in both model accuracy and system efficiency. The losses reduce pre-training perplexity and improve downstream performance, while the resulting stable routing paths enhance inference throughput via better caching and batching, requiring no architectural changes.

- **Drop-in Megatron-LM Integration.** We release our method as a non-invasive module for Megatron-LM. It is activated by a single configuration flag and requires no modifications to core attention, FFN, or router logic, ensuring immediate usability.

## 2 RELATED WORK

**Balancing Losses and Specialization Objectives.** A primary strategy to prevent routing collapse and improve stability in MoE training is to enforce balanced expert utilization. Early systems such as GShard (Lepikhin et al., 2020) and Switch (Fedus et al., 2022) introduced auxiliary load-balancing terms to distribute tokens across experts, with router z-loss (Zoph et al., 2022) providing additional stabilization. BASE layers (Lewis et al., 2021) formulated routing as an optimal linear assignment problem, achieving perfectly balanced usage without auxiliary terms. Expert-Choice routing (Zhou et al., 2022) further reversed the assignment process, allowing experts to select their Top-$k$ tokens, which inherently balances load. These methods primarily regulate *how much* each expert is used. In contrast, our approach is complementary: we supervise *what* experts learn and *how* their paths align, introducing a within-layer similarity penalty to discourage activation overlap and a cross-layer coupling term to enforce coherence, while leaving existing balancing mechanisms intact.

**Architectural and Router-Level Approaches.** Another line of work promotes expert specialization by redesigning MoE architectures or router mechanisms. DeepSeekMoE (Dai et al., 2024) partitions experts more finely and introduces always-active shared experts, allowing routed specialists to focus on idiosyncratic patterns. Router-centric methods also refine gating: ReMoE (Wang et al., 2025b) replaces Top-$k$ Softmax with a differentiable ReLU router and adaptive $L_1$ regularization, while Dynamic MoE (Guo et al., 2025b) auto-tunes both the number of activated experts per token and the size of the expert pool. Several structural variants further expand capacity and specialization. Mixtral layers multiple FFNs with top-2 routing, achieving strong accuracy–efficiency trade-offs (Jiang et al., 2024); Mixture of a Million Experts pushes expert granularity to the extreme (He et al., 2024); HMoE mixes experts of different sizes and biases usage toward smaller ones to encourage division of labor (Wang et al., 2024); MoDSE deploys diverse-sized experts with pairwise allocation to stabilize routing and balance compute across devices (Sun et al., 2024); while simpler approaches such as Hash Layers (Roller et al., 2021) and THOR (Zuo et al., 2022) enforce balanced usage through fixed or randomized routing schemes.

Unlike these methods—which modify layer composition or router design and largely rely on in-layer dynamics—our approach is architecture-agnostic. We impose explicit specialization objectives, namely a within-layer similarity penalty and a cross-layer coupling loss, on top of existing designs without altering attention, FFN, or router code paths.

**Cross-layer signals and information.** Recent work shows that MoE routing decisions are often correlated across layers and leverages these correlations for system efficiency. Read-ME (Cai et al., 2024) pre-computes routing across depth to enable lookahead scheduling and caching, yielding substantial inference speedups. The Layerwise Recurrent Router (RMoE) (Wang et al., 2025a) passes routing context forward via a GRU, producing more consistent assignments and improved stability. Both methods exploit cross-layer patterns for efficiency or stability but do not explicitly shape them during training. In contrast, we turn cross-layer coherence into a learning objective: our coupling loss actively encourages tokens to follow aligned expert paths across layers, transforming a byproduct of training into a supervisory signal that strengthens specialization.

## 3 MIXTURE-OF-EXPERTS MODELS: PRELIMINARIES

The MoE model enhances standard transformers by replacing feed-forward network (FFN) layers with MoE layers. Each MoE layer contains a set of $E$ independent FFNs, called experts, and a router that dynamically selects a sparse subset of these experts for each input token. The computation for the $i$-th token within a single MoE layer $l$ proceeds in three main steps:

**Routing.** Let $x_i^{(l)} \in \mathbb{R}^h$ be the input token representation. The router first calculates a logit $q_i^{(l,e)}$ for each expert $e$ using a learnable routing vector $\mathcal{R}^{(l,e)} \in \mathbb{R}^h$. These logits are then normalized via a softmax function to produce the final routing scores $s_i^{(l,e)}$:

$$q_i^{(l,e)} = \langle x_i^{(l)}, r^{(l,e)} \rangle, \qquad s_i^{(l,e)} := \frac{\exp(q_i^{(l,e)})}{\sum_{j=1}^E \exp(q_i^{(l,j)})}, \tag{1}$$

where $\langle \cdot, \cdot \rangle$ denotes the standard inner product.

**Expert Processing.** The router uses these scores to select the top-$k$ experts (where $k \ll E$), denoted by the set $\mathbb{A}_i^{(l)}$. The original input $x_i^{(l)}$ is then processed in parallel by each activated expert $e \in \mathbb{A}_i^{(l)}$. Each expert is an FFN, often a SwiGLU network, with its own weights $(W_{\text{gate}}^{(l,e)}, W_{\text{up}}^{(l,e)}, W_{\text{down}}^{(l,e)})$:

$$z_i^{(l,e)} = \text{SwiGLU}\left(x_i^{(l)} W_{\text{gate}}^{(l,e)}\right) \odot \left(x_i^{(l)} W_{\text{up}}^{(l,e)}\right), \quad y_i^{(l,e)} = z_i^{(l,e)} W_{\text{down}}^{(l,e)}, \tag{2}$$

where $\odot$ denotes the Hadamard product. Quantity $z_i^{(l,e)}$ is referred to as the expert activation.

**Output Combination.** The final output of the MoE layer, $y_i^{(l)}$, is a weighted combination of the expert outputs, using the routing scores as the weights: $y_i^{(l)} = \sum_{e \in \mathbb{A}_i^{(l)}} s_i^{(l,e)} y_i^{(l,e)}$.

**Two failure modes.** We identify two fundamental failure modes that undermine specialization in MoE models: *Expert Overlap* and *Routing Ambiguity*. The first occurs when different experts produce nearly identical activations for the same tokens, creating redundant representations that waste model capacity. The second manifests when the router inconsistently dispatches similar tokens, which prevents experts from receiving stable data distributions and causes their learning updates to converge toward the same functionality. When these issues arise, the MoE architecture collapses into functional redundancy, defeating the core principle of a specialized division of labor.

## 4    INTRA-LAYER SPECIALIZATION LOSS

This section designs the loss function to penalize expert overlap. While load-balancing losses ensure even utilization, they neither prevent functional redundancy nor guarantee diversity among experts.

**Linking Activations to Expert Learning Trajectories.** Expert specialization requires divergent learning trajectories, which manifests as distinct parameter update directions during training. Since these update directions are determined by loss function gradients, specialization necessitates that expert parameter gradients remain maximally dissimilar—ideally orthogonal—throughout optimization. This raises a critical question: *how can we control the angle between expert gradients during training*? Our analysis reveals a surprisingly simple answer below. It establishes a direct link between the geometry of the experts' activations and the geometry of their weight gradients.

**Proposition 1.** *For any two activated experts $e, \nu \in \mathbb{A}_i^{(l)}$, the cosine similarity between the gradients of the total loss $\mathcal{L}$ with respect to their down-projection matrices, $W_{down}^{(l,e)}$ and $W_{down}^{(l,\nu)}$, is equal to the cosine similarity of their activations, $z_i^{(l,e)}$ and $z_i^{(l,\nu)}$ (Proof is in Appendix A.1):*

$$\cos\left(\frac{\partial \mathcal{L}}{\partial W_{down}^{(l,e)}}, \frac{\partial \mathcal{L}}{\partial W_{down}^{(l,\nu)}}\right) = \cos\left(z_i^{(l,e)}, z_i^{(l,\nu)}\right). \tag{3}$$

This proposition establishes that near-orthogonal expert activations with small cosine activations induce correspondingly orthogonal parameter gradients, thereby driving experts along divergent learning trajectories throughout training. We thus achieve the following insight:

> **Takeaway 1.** *To ensure the parameter gradients of different experts to be orthogonal, we can force their activations to be orthogonal.*

**Intra-Layer Specialization Loss.** The above insight directly motivates our regularization term that penalizes representational similarity between experts. For token $x_i$, we define the loss as the sum of squared cosine similarities between intermediate activations $z_i^{(l,e)}$ across all active expert pairs. Squaring amplifies larger similarities and ensures a smooth, stable optimization landscape:

$$\mathcal{R}_{\text{sp}}(x_i) = \sum_{l=1}^{L} \sum_{e, \nu \in \mathbb{A}_i^{(l)}} \left[\cos\left(z_i^{(l,e)}, z_i^{(l,\nu)}\right)\right]^2 \tag{4}$$

Minimizing $\mathcal{R}_{\text{sp}}$ directly encourages representational orthogonality, which by Proposition 1, induces the divergent parameter updates necessary for expert specialization. Specifically, this regularization

drives experts to capture distinct, non-overlapping features: orthogonal activations ensure each expert encodes complementary aspects of the input data, minimizing redundancy while maximizing representational diversity. This mechanism transforms the abstract goal of specialization into a concrete optimization objective without the need for intricate architectural designs.

**Rationale for Expert Specialization Mechanism.** Our regularization strategy specifically focuses on the intermediate activations $z$ at the $W_{\text{down}}$ composition stage, rather than applying parallel constraints to the $W_{\text{up}}$ and $W_{\text{gate}}$ pathways, based on both theoretical and practical considerations. The choice is theoretically grounded in the established identity $\cos(\nabla W_{\text{down}}^{(\ell,e)}, \nabla W_{\text{down}}^{(\ell,\nu)}) = \cos(z_i^{(\ell,e)}, z_i^{(\ell,\nu)})$, which directly links activation orthogonality to divergent gradient directions in the parameter space. This principled selectivity provides a computationally efficient mechanism to promote expert specialization while avoiding the complexity and potential optimization conflicts that could arise from imposing multiple regularization objectives.

This targeted approach is supported by the formal relationship established in Proposition 1, which demonstrates that orthogonal activations necessarily induce orthogonal gradients in expert parameters. By establishing this direct link between activation-space regularization and gradient behavior, we develop a framework that ensures distinct experts capture complementary features while minimizing functional overlap. The resulting specialization mechanism achieves efficient knowledge distribution throughout the mixture-of-experts architecture while maintaining optimization stability.

**Empirical Validation.** To validate that $\mathcal{R}_{\text{sp}}$ effectively measures expert specialization, we pretrained a 1.1B MoE model (110M activated parameters) with and without this regularization while maintaining the other settings identical. We tested four configurations: $\mathcal{L}_{\text{lb}}$ (load balance only), $\mathcal{L}_{\text{lb,sp}}$ (load balance + specialization), $\mathcal{L}_{\text{lb,z}}$ (load balance + z-loss (Zoph et al., 2022)), and $\mathcal{L}_{\text{lb,z,sp}}$ (all three losses). Figure 1 shows that incorporating our specialization loss consistently reduces perplexity across all configurations, with the combined $\mathcal{L}_{\text{lb,z,sp}}$ achieving the best performance.

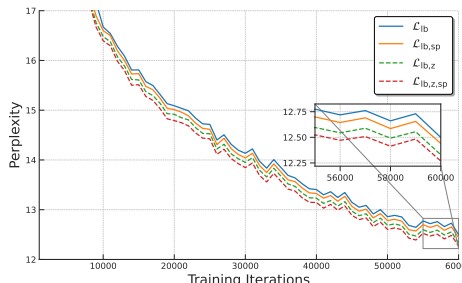

Figure 1: The perplexity for training a 1.1B model with different regularization. Setup is in Table 5.

## 5 CROSS-LAYER COUPLING LOSS

This section designs the loss function to address routing ambiguity. When near-identical tokens are scattered across multiple experts, each expert receives a mixed—and largely overlapping—data distribution, so their gradients become correlated and updates drive them toward the same functionality. Without stable, consistent assignments, experts cannot develop distinct roles, token–expert boundaries remain blurred, and the intended division of labor in MoE collapses into redundant behavior.

**The Phenomenon of Cross-Layer Coupling.** While routing ambiguity poses a significant challenge, recent research has uncovered a valuable emergent property in MoE models: cross-layer coupling (Cai et al., 2024; Yao et al., 2024). This phenomenon manifests as strong predictive relationships between expert activations across adjacent layers—the expert activated at layer $l$ reliably predicts the expert selection at layer $l + 1$. During training, models spontaneously develop these structured pathways, creating coherent information pipelines through network depth.

**Cross-Layer Coupling Amplifies Specialization.** While cross-layer coupling intuitively promotes routing stability—when tokens consistently traverse specific expert sequences (e.g., expert 3 in layer 7 followed by expert 5 in layer 8), routing ambiguity is eliminated by definition—its impact on expert specialization warrants deeper examination. We address a central question: *How does inter-layer structural consistency influence intra-layer expert differentiation?* Our theoretical analysis reveals an intriguing propagation mechanism, formalized below.

**Proposition 2.** *Let $\mathbb{A}_i^{(l)}$ denote the set of activated experts for token $x_i$ at layer $l$. Consider adjacent layers $l$ and $l + 1$ satisfying the following conditions:*

Figure 2: The probability for different experts in layer 8 to be activated conditional on the activated experts in layer 7 during the training process of a 0.4B MoE model. (Top: training with only load balance regularization; bottom: training with load balance and coupling regularization.)

1. **Representation Continuity:** *For any token $x_i$, its representations evolve smoothly across layers: $\cos(x_i^{(l)}, x_i^{(l+1)}) \geq 1 - \delta^2$ for small $\delta \in (0, 1)$.*

2. **Source Layer Specialization:** *Layer $l$ exhibits expert specialization with nearly orthogonal router weights: for experts $e_1 \in \mathbb{A}_i^{(l)}$ and $e_2 \in \mathbb{A}_j^{(l)}$ processing different tokens $x_i \neq x_j$, we have $|\cos(r^{(l,e_1)}, r^{(l,e_2)})| \leq \varepsilon$ for small $\varepsilon \in (0, 1)$.*

3. **Strong Cross-Layer Coupling:** *Adjacent layers exhibit stable expert pathways with high routing correlation. For any expert $e \in \mathbb{A}_i^{(l)}$ activated by token any $x_i$, there exists a corresponding expert $\nu \in \mathbb{A}_i^{(l+1)}$ such that both routing decisions are confident: $\cos(x_i^{(l)}, r^{(l,e)}) \geq 1 - \iota^2$ and $\cos(x_i^{(l+1)}, r^{(l+1,\nu)}) \geq 1 - \iota^2$ for small $\iota \in (0, 1)$.*

*Under these conditions, layer $l + 1$ inherits the specialization structure from layer $l$:*

$$\left| \cos\left( r^{(l+1,\nu_1)}, r^{(l+1,\nu_2)} \right) \right| \leq \varepsilon + O(\delta, \iota) \tag{5}$$

*for experts $\nu_1 \in \mathbb{A}_i^{(l+1)}$ and $\nu_2 \in \mathbb{A}_j^{(l+1)}$ processing different tokens, where the error term $O(\delta, \iota)$ vanishes as $\delta$ and $\iota$ decreases to 0 (proof in Appendix A.2).*

Proposition 2 establishes a mechanism for network-wide specialization propagation through cross-layer coupling. The result demonstrates that when layer $l$ exhibits well-specialized experts (Condition 2) and maintains strong coupling with layer $l + 1$ (Condition 3), the specialization structure transfers to the adjacent layer with bounded degradation (see Eq. 5). This propagation property enables localized specialization to cascade through network depth, ultimately producing globally specialized representations. We thus achieve the following insight:

> **Takeaway 2.** *Cross-layer coupling acts as a specialization amplifier: it transforms localized expert differentiation into network-wide functional diversity by creating stable pathways that propagate specialization across depth.*

**Cross-Layer Coupling Loss.** While cross-layer coupling emerges naturally, it develops slowly and incompletely, particularly during early training when routing ambiguity is severe. Rather than waiting for them to develop organically, we therefore introduce the loss function $\mathcal{R}_{\text{cp}}$ to actively promote stable expert pathways by maximizing joint routing probabilities between adjacent layers. For each token, we compute pathway strength as the product of routing scores $P_i^{(l,(e,\nu))}$ as listed in Eq. (6), representing the joint probability of activating expert $e$ at layer $l$ and expert $\nu$ at layer $l + 1$.

Table 1: The specialization loss during the training period for the 0.8B model with different loss.

| Iteration | 1K | 3K | 5K | 10K | 20K | 30K | 60K |
|---|---|---|---|---|---|---|---|
| $\mathcal{L}_{\text{lb,sp}}$ | 0.11551 | 0.04243 | 0.03296 | 0.02172 | 0.020822 | 0.020476 | 0.019924 |
| $\mathcal{L}_{\text{lb,sp,cp}}$ | 0.10223 | 0.03748 | 0.03044 | 0.02008 | 0.019417 | 0.019136 | 0.018862 |

The loss considers the Top-$k$ strongest inter-layer connections for each expert:

$$\mathcal{R}_{\text{cp}}(x_i) = -\sum_{l=1}^{L-1} \sum_{e \in \mathbb{A}_i^{(l)}} \sum_{\nu \in \mathbb{T}_i^{(l,e)}} P_i^{(l,(e,\nu))}, \quad \text{where} \quad P_i^{(l,(e,\nu))} = s_i^{(l,e)} s_i^{(l+1,\nu)}. \tag{6}$$

Here, $s_i$ is defined in Eq. (1), $\mathbb{T}_i^{(l,e)}$ contains the $k$ experts in layer $l+1$ with highest joint probabilities with expert $e$. Minimizing $\mathcal{R}_{\text{cp}}$ establishes coherent cross-layer expert selection pipelines that, by Proposition 2, create the structural conditions for specialization propagation throughout the network.

**Discussion on Coupling Loss Mechanism** The coupling loss $\mathcal{R}_{\text{cp}}$ is designed to stabilize inter-layer routing pathways by optimizing joint routing probabilities for consistent expert pairs across consecutive layers. This stabilization reduces routing ambiguity and minimizes token distribution overlap among experts, thereby promoting functional diversification. By encouraging orthogonal router configurations, $\mathcal{R}_{\text{cp}}$ reduces score correlation and co-activation, thereby minimizing gradient sharing and encouraging divergent specialization. This mechanism ensures that experts develop distinct roles by processing different subsets of tokens, thereby enhancing overall model efficiency and reducing functional redundancy.

**Empirical Validation.** To confirm that cross-layer coupling is a natural characteristic of MoE training worth amplifying, we pre-trained a 0.4B MoE model with 80M activated parameters and observed the conditional activation probabilities between adjacent layers. As shown in Figure 2, a clear coupling structure is present from the early stages of pre-training and becomes progressively more pronounced over time. This confirms that structured expert paths are an intrinsic feature of MoE learning, validating the premise for our coupling loss as a means to harness and accelerate this behavior. Moreover, we train this 0.4B MoE model with $\mathcal{L}_{\text{lb,sp}}$ and $\mathcal{L}_{\text{lb,sp,cp}}$, respectively. We obtain the specialization loss during the training period as Table 1, from which it can be observation that the introduction of coupling loss can reduce the specialization loss.

## 6 NEW TRAINING OBJECTIVES FOR MOE MODELS

**Combined Training Objective.** With both the intra-layer specialization and the cross-layer coupling losses, we integrate them into the training objective for token $x_i$ as regularization terms:

$$\mathcal{L}_{\text{lb,sp,cp}}(x_i) := \mathcal{L}(x_i) + \mathcal{R}_{\text{lb}}(x_i) + \lambda_{\text{sp}} \mathcal{R}_{\text{sp}}(x_i) + \lambda_{\text{cp}} \mathcal{R}_{\text{cp}}(x_i), \tag{7}$$

where $\mathcal{L}(x_i)$ is the primary language modeling loss, $\mathcal{R}_{\text{lb}}(x_i)$ is the standard load-balancing regularization, and $\lambda_{\text{sp}}$, $\lambda_{\text{cp}}$ are hyperparameters controlling the strength of specialization and coupling regularization, respectively. This joint optimization simultaneously promotes expert specialization and routing stability while maintaining balanced token utilization.

**Compatibility with Load Balancing.** An important consideration for MoE regularization is its interaction with load balancing, which supports training stability and computational efficiency. We show that our proposed losses are naturally compatible with this requirement, as they operate on complementary principles that do not conflict with standard load-balancing objectives.

(1) $\mathcal{R}_{\text{sp}}$ is Compatibility with $\mathcal{R}_{\text{lb}}$. Given a token input space $\mathcal{P}^{(l)}$ at layer $l$, minimizing $\mathcal{R}_{\text{sp}}$ aims to partition this space into $E$ disjoint subspaces $\{\mathcal{P}^{(l,e)}\}_{e=1}^{E}$, ensuring each expert specializes on distinct inputs. Concurrently, minimizing $\mathcal{R}_{\text{lb}}$ enforces balanced utilization where $|\mathcal{P}^{(l,e)}| = |\mathcal{P}^{(l,\nu)}|$ for all experts $e$ and $\nu$. These objectives operate orthogonally: specialization determines the partitioning scheme (non-overlapping regions), while load balancing constrains partition sizes (equal cardinality). Proposition 3 (Appendix A.3) constructively proves that disjoint, equal-sized partitions exist, establishing theoretical compatibility between specialization and load-balancing objectives.

Table 2: Validation perplexity (↓) across model scales and auxiliary-loss configurations.

| Losses | Vanilla MoE | | | DeepSeek-style MoE | | |
|---|---|---|---|---|---|---|
| | Small | Medium | Large | Small | Medium | Large |
| $\mathcal{L}_{lb}$ | 14.01 | 12.50 | 9.68 | 13.54 | 12.33 | 9.56 |
| $\mathcal{L}_{lb,sp,cp}$ | 13.75 | 12.27 | 9.48 | 13.37 | 12.16 | 9.47 |
| $\mathcal{L}_{lb,z}$ | 13.80 | 12.33 | 9.52 | 13.40 | 12.07 | 9.46 |
| $\mathcal{L}_{lb,z,sp,cp}$ | 13.63 | 12.17 | 9.42 | 13.30 | 11.99 | 9.39 |

(2) $\mathcal{R}_{cp}$ is Compatibility with $\mathcal{R}_{lb}$. The coupling loss $\mathcal{R}_{cp}$ and load-balancing loss $\mathcal{R}_{lb}$ also operate on fundamentally different axes. While $\mathcal{R}_{cp}$ enforces token-wise consistency across layers—ensuring each token follows a stable expert pathway through network depth—$\mathcal{R}_{lb}$ enforces batch-wise balance within each layer, distributing the total workload evenly among experts. These constraints are non-conflicting: a routing strategy can simultaneously maintain consistent per-token paths (satisfying coupling) while ensuring different tokens take different paths to achieve aggregate balance (satisfying load distribution). Proposition 4 (Appendix A.3) formally proves that an optimal routing configuration exists that minimizes $\mathcal{R}_{cp}$ while maintaining perfect load balance.

Propositions 3 and 4 establish that our regularization losses are compatible with load-balancing constraints, ensuring both training efficiency and model performance are preserved. This compatibility enables the specialization and coupling losses to function as plug-and-play modules that enhance expert differentiation without disrupting computational balance.

> **Takeaway 3.** *When the intra-layer specialization and cross-layer coupling losses are optimized, load balancing among experts can be simultaneously preserved.*

## 7 EXPERIMENTS

In this section, we use $\mathcal{L}_a$ to represent the loss that combined by regularization a. For example, we use $\mathcal{L}_z$ to denote the router z-loss (Fedus et al., 2022) and we use $\mathcal{L}_{lb,sp,cp}$ to denote the loss that combined $\mathcal{R}_{lb}$, $\mathcal{R}_{sp}$, and $\mathcal{R}_{cp}$. Experimental details and additional experiments are in Appendix B.

**Comparison of validation perplexity.** We evaluate on C4 dataset (Raffel et al., 2020) across models with different sizes for both Vanilla and DeepSeek-style MoE. Table 2 reports validation perplexity under different auxiliary-loss configurations. It can be observed that the proposed $\mathcal{R}_{sp}$ and $\mathcal{R}_{cp}$ consistently improve performance when added to standard objectives. Compared to training with $\mathcal{L}_{lb}$ alone, there appears to be a significant improvement in the validation performance by introduction $\mathcal{R}_{sp}$ and $\mathcal{R}_{cp}$ on all scales. When involving z-loss, adding the proposed two regularization terms yields further gains and the lowest overall perplexities , indicating that our losses complement rather than replace established objectives.

Furthermore, these improvements are architecture-agnostic. For the Vanilla MoE model, the combined application of $\mathcal{R}_{sp}$ and $\mathcal{R}_{cp}$ leads to a consistent reduction in perplexity. Similarly, on DeepSeek-style MoE—which already integrates shared experts—the same objectives yield further enhancements. Notably, in medium and large-scale configurations, the Vanilla MoE enhanced with $\mathcal{R}_{sp}$ and $\mathcal{R}_{cp}$ even outperforms the DeepSeek-style variant that includes a shared expert, achieving lower perplexity while maintaining the same number of routed experts and without additional activated capacity. Therefore, targeted loss functions designed to improve specialization and path coherence can compete with or surpass architectural variants and router modifications, while remaining plug-and-play across diverse MoE designs.

**Downstream Task Evaluations for pre-trained MoE models.** We evaluate the pre-trained MoE models on supervised fine-tuning tasks (see Appendix for details; (Raffel et al., 2020)) and seven zero-shot benchmarks: BoolQ (Clark et al., 2019), ARC-Easy and ARC-Challenge (Clark et al., 2018), TruthfulQA-MC2 (Lin et al., 2022), PIQA (Bisk et al., 2020), MMLU (Hendrycks et al., 2021), and HellaSwag (Zellers et al., 2019) as outlined in Table 3. For each experimental setup, the process was conducted three times with different random seeds to ensure robustness.

Table 3: Zero-shot accuracy of *Vanilla MoE* and *DeepSeek-style MoE* across seven benchmarks (↑).

| Model | Loss | BoolQ | ARC-E | ARC-C | Truthful QA-MC2 | PIQA | MMLU | Hella Swag | Avg. |
|---|---|---|---|---|---|---|---|---|---|
| Vanilla MoE | $\mathcal{L}_{lb}$ | 0.570 (0.003) | 0.452 (0.003) | 0.204 (0.003) | 0.432 (0.001) | 0.622 (0.005) | 0.247 (0.002) | 0.268 (0.002) | 0.399 |
| | $\mathcal{L}_{lb,sp,cp}$ | 0.578 (0.003) | 0.462 (0.002) | 0.210 (0.004) | 0.451 (0.003) | 0.627 (0.002) | 0.253 (0.002) | 0.275 (0.004) | **0.408** |
| | $\mathcal{L}_{lb,z}$ | 0.567 (0.003 | 0.457 (0.004) | 0.205 (0.002) | 0.433 (0.003) | 0.629 (0.002) | 0.250 (0.001) | 0.267 (0.004) | 0.401 |
| | $\mathcal{L}_{lb,z,sp,cp}$ | 0.589 (0.003) | 0.453 (0.004) | 0.206 (0.006) | 0.445 (0.003) | 0.637 (0.003) | 0.257 (0.002) | 0.274 (0.003) | **0.409** |
| DS-style MoE | $\mathcal{L}_{lb}$ | 0.578 (0.002) | 0.453 (0.001) | 0.205 (0.003) | 0.438 (0.003) | 0.631 (0.002) | 0.248 (0.001) | 0.269 (0.002) | 0.403 |
| | $\mathcal{L}_{lb,sp,cp}$ | 0.584 (0.001) | 0.452 (0.003) | 0.206 (0.005) | 0.457 (0.002) | 0.635 (0.002) | 0.255 (0.003) | 0.277 (0.005) | **0.410** |
| | $\mathcal{L}_{lb,z}$ | 0.564 (0.002) | 0.453 (0.002) | 0.205 (0.002) | 0.444 (0.002) | 0.628 (0.001) | 0.252 (0.001) | 0.270 (0.004) | 0.402 |
| | $\mathcal{L}_{lb,z,sp,cp}$ | 0.575 (0.002) | 0.461 (0.004) | 0.214 (0.004) | 0.452 (0.004) | 0.642 (0.003) | 0.257 (0.002) | 0.280 (0.002) | **0.412** |

Table 4: Evaluation score on Qwen3-30B-A3B-Instruct-2507 finetuning tasks. The last four rows stands for the performance for mmlu dataset with different domains.

| Dataset | Metric | $\mathcal{L}_{lb}$ | $\mathcal{L}_{lb,sp,cp}$ |
|---|---|---|---|
| openai_humaneval | humaneval_pass@1 | 92.07 | **95.73** |
| gsm8k | accuracy | 93.33 | **94.16** |
| math_prm800k_500 | accuracy | 94.00 | **94.20** |
| mmlu | naive_average | 78.97 | **79.86** |
| mmlu-weighted | weighted_average | 76.35 | **77.10** |

Across both architectures, the addition of $\mathcal{R}_{cp}$ and $\mathcal{R}_{sp}$ enhances zero-shot accuracy in synergy with load-balance loss and z-loss. For the Vanilla MoE, integrating $\mathcal{R}_{sp}$ and $\mathcal{R}_{cp}$ with load-balance regularization leads to a marked improvement in average accuracy. Further gains are observed when $\mathcal{R}_{sp}$ and $\mathcal{R}_{cp}$ are applied in combination with load-balance loss and z-loss. In the DeepSeek-style MoE, a similar trend emerges: the inclusion of $\mathcal{R}_{sp}$ and $\mathcal{R}_{cp}$ alongside $\mathcal{R}_{lb}$ boosts average performance, while the loss with full set of regularization ($\mathcal{L}_{lb,z,sp,cp}$) achieves the highest overall accuracy, outperforming both $\mathcal{L}_{lb,z}$ and $\mathcal{L}_{lb}$, and yielding superior results on benchmarks such as ARC-E, ARC-C, and PIQA.

Although individual benchmarks show slight variations, the consistent upward trend in average accuracy demonstrates that $\mathcal{R}_{cp}$ and $\mathcal{R}_{sp}$ effectively complement load-balance loss and z-loss, contributing to downstream improvements across model families.

**Finetuning tasks evaluation.** We fine-tune `Qwen3-30B-A3B-Instruct-2507` model on an internal corpus of 38B tokens (see details in Appendix C). We evaluate on a broad suite of reasoning and knowledge-intensive benchmarks, including HumanEval (Chen et al., 2021), GSM8K (Cobbe et al., 2021), math500_PRM800K_dataset (Lightman et al., 2023), and MMLU (Hendrycks et al., 2021). Across nearly all settings, incorporating $\mathcal{R}_{cp}$ and $\mathcal{R}_{sp}$ outperforms the baseline, yielding consistent gains on reasoning-oriented tasks as well as aggregate knowledge measures as Table 4. While a minor fluctuation is observed on the humanities subset of MMLU, the overall trend remains positive, confirming that our objectives not only sharpen specialization in pre-training but also transfer effectively to finetuning adaptation.

**Scalability of the Auxiliary Loss.** We conduct scalability experiments on the MoE model with small size by varying both the number of activated experts and the total number of experts. As shown in Figure 3, our auxiliary losses consistently yield lower perplexity compared to the baseline across both scaling axes. The performance gains remain stable as the number of activated experts increases, and they also persist as the total expert pool expands. These results demonstrate that the

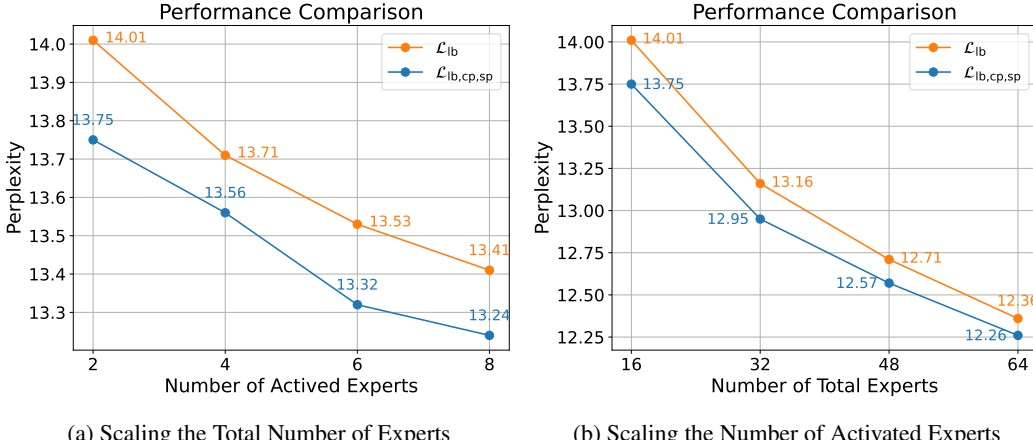

(a) Scaling the Total Number of Experts   (b) Scaling the Number of Activated Experts

Figure 3: Scalability performance on the model with small size: (a) varying total experts; (b) varying activated experts.

proposed objectives generalize robustly across different scaling configurations, highlighting their broad applicability and effectiveness regardless of model size or routing capacity.

# 8   CONCLUSION

We presented two plug-and-play losses that directly optimize expert specialization in MoE models. The intra-layer specialization loss ($\mathcal{R}_{\mathrm{sp}}$) penalizes activation similarity between experts processing identical tokens, while the cross-layer coupling loss ($\mathcal{R}_{\mathrm{cp}}$) maximizes joint routing probabilities across adjacent layers to establish coherent expert pathways. These losses require no architectural modifications, integrate seamlessly with existing objectives, and are theoretically grounded. Empirically, our approach improves performance across all tested scales and MoE variants while increasing inference throughput through stable expert paths.

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

# A PROOF FOR THE PROPOSITIONS.

In this section, we present the proofs for the proposed propositions. And we also present two propositions to formally present Takeaway 3.

## A.1 NEARLY ORTHOGONAL INTERMEDIATES LEAD TO NEARLY ORTHOGONAL GRADIENTS

Here we present the proof of Proposition 1, which provides a formal presentation for Takeaway 1 that nearly orthogonal intermediates can lead to nearly orthogonal gradients of the weights of down-projection.

*Proof.* As the routing weights do not affect the cosine, without loss of generality we assume that the activated experts contribute with equal weights. Then the output of MoE blocks for layer $l$ can be written as

$$E(x_i^{(l)}) := \sum_{e \in \mathbb{A}_i^{(l)}} y_i^{(l,e)}. \tag{8}$$

Then for any $e \in \mathbb{A}_i^{(l)}$ it holds that

$$\frac{\partial \mathcal{L}}{\partial y^{(l,e)}} = \frac{\partial \mathcal{L}}{\partial E(x_i^{(l)})}. \tag{9}$$

As $y_i^{(l,e)} = z_i^{(l,e)} W_{\text{down}}^{(l,e)}$, thus from (9) it comes for any $e \in \mathbb{A}_i^{(l)}$ that:

$$\frac{\partial \mathcal{L}}{\partial W_{\text{down}}^{(l,e)}} = \frac{\partial \mathcal{L}}{\partial y_i^{(l,e)}} \frac{\partial y_i^{(l,e)}}{\partial W_{\text{down}}^{(l,e)}} = \frac{\partial \mathcal{L}}{\partial E(x_i^{(l)})} z_i^{(l,e)}. \tag{10}$$

Using the Frobenius inner-product identity $\langle ab^\top, \, cd^\top \rangle_F = (a^\top c)(b^\top d)$ and $\|ab^\top\|_F = \|a\|_2 \|b\|_2$, we obtain that for $e_1, e_2 \in \mathbb{A}_i^{(l)}$ it holds that

$$\cos\left(\frac{\partial \mathcal{L}}{\partial W_{\text{down}}^{(l,e_1)}}, \frac{\partial \mathcal{L}}{\partial W_{\text{down}}^{(l,e_2)}}\right) = \frac{\left[\left(z_i^{(l,e_1)}\right)^\top z_i^{(l,e_2)}\right] \cdot \left[\frac{\partial \mathcal{L}}{\partial \left(y_i^{(l,e)}\right)^\top} \frac{\partial \mathcal{L}}{\partial y_i^{(l,e)}}\right]}{\left\|z_i^{(l,e_1)}\right\|_2 \cdot \left\|\frac{\partial \mathcal{L}}{\partial \left(y_i^{(l,e)}\right)}\right\|_2 \cdot \left\|z_i^{(l,e_2)}\right\|_2 \cdot \left\|\frac{\partial \mathcal{L}}{\partial \left(y_i^{(l,e)}\right)}\right\|_2}$$

$$= \frac{z_i^{(l,e_1)} \left(z_i^{(l,e_2)}\right)^\top}{\left\|z_i^{(l,e_1)}\right\|_2 \left\|z_i^{(l,e_2)}\right\|_2} = \cos\left(z_i^{(l,e_1)}, z_i^{(l,e_2)}\right). \tag{11}$$

When condisering the case that each expert output is scaled by a positive routing weight, i.e., $\widetilde{y}_i^{(l,e)} = \alpha_i^{(l,e)} \cdot z_i^{(l,e)} W_{\text{down}}^{(l,e)}$, where $\alpha_i^{(l,e)} \in (0,1]$ is the routing weight. Similar to (10), we can obtain that

$$\frac{\partial \mathcal{L}}{\partial W_{\text{down}}^{(l,e)}} = \alpha_i^{(l,e)} \cdot \frac{\partial \mathcal{L}}{\partial E(x_i^{(l)})} z_i^{(l,e)}.$$

Thus the common positive factor cancels in the cosine similarity, leaving the result unchanged. $\square$

## A.2 CROSS-LAYER DEPENDENCY CAN ENHANCE THE SPECIALIZATION

In this subsection, we present the proof for Proposition 2 which supports Takeaway 2 that the cross-layer coupling loss can enhance the intra-layer expert specialization.

*Proof.* From Assumption 2, It can be obtained that:

$$\left\| \frac{x_i^{(l,e)}}{\left\|x_i^{(l,e)}\right\|} - \frac{x_i^{(l+1,e)}}{\left\|x_i^{(l+1,e)}\right\|} \right\|^2 = 2 - 2\cos\left(x_i^{(l,e)}, x_i^{(l+1,e)}\right) \le 2\delta^2. \quad (12)$$

Similarly, it holds that:

$$\left\| \frac{x_i^{(l,e)}}{\left\|x_i^{(l,e)}\right\|} - \frac{r^{(l,e)}}{\left\|r^{(l,e)}\right\|} \right\|^2 \le 2\iota^2, \quad \left\| \frac{x_i^{(l+1,\nu)}}{\left\|x_i^{(l+1,\nu)}\right\|} - \frac{r^{(l+1,\nu)}}{\left\|r^{(l+1,\nu)}\right\|} \right\|^2 \le 2\iota^2. \quad (13)$$

Then from Eq. (12) and Eq. (13), it holds that

$$\left\| \frac{r^{(l,e)}}{\left\|r^{(l,e)}\right\|} - \frac{r^{(l+1,\nu)}}{\left\|r^{(l+1,\nu)}\right\|} \right\|$$

$$\le \left\| \frac{x_i^{(l,e)}}{\left\|x_i^{(l,e)}\right\|} - \frac{x_i^{(l+1,e)}}{\left\|x_i^{(l+1,e)}\right\|} \right\| + \left\| \frac{x_i^{(l,e)}}{\left\|x_i^{(l,e)}\right\|} - \frac{r^{(l,e)}}{\left\|r^{(l,e)}\right\|} \right\| + \left\| \frac{x_i^{(l+1,\nu)}}{\left\|x_i^{(l+1,\nu)}\right\|} - \frac{r^{(l+1,\nu)}}{\left\|r^{(l+1,\nu)}\right\|} \right\| \quad (14)$$

$$\le \sqrt{2}\left(\delta + 2\iota\right).$$

Then it holds that

$$\cos\left(r^{(l,e)}, r^{(l+1,\nu)}\right) = 1 - \frac{1}{2}\left\| \frac{r^{(l,e)}}{\left\|r^{(l,e)}\right\|} - \frac{r^{(l+1,\nu)}}{\left\|r^{(l+1,\nu)}\right\|} \right\|^2 \ge 1 - (\delta + 2\iota)^2. \quad (15)$$

Then we prove (5). Let

$$\tilde{r}^{(l,e_1)} := \frac{r^{(l,e_1)}}{\left\|r^{(l,e_1)}\right\|}, \quad \tilde{r}^{(l+1,\nu_1)} := \frac{r^{(l+1,\nu_1)}}{\left\|r^{(l+1,\nu_1)}\right\|},$$

$$\tilde{r}^{(l,e_2)} := \frac{r^{(l,e_2)}}{\left\|r^{(l,e_2)}\right\|}, \quad \tilde{r}^{(l+1,\nu_2)} := \frac{r^{(l+1,\nu_2)}}{\left\|r^{(l+1,\nu_2)}\right\|}.$$

Then it comes that:

$$\left| \left\langle \tilde{r}^{(l+1,\nu_1)}, \tilde{r}^{(l+1,\nu_2)} \right\rangle \right|$$

$$= \left| \left\langle \tilde{r}^{(l,e_1)}, \tilde{r}^{(l,e_2)} \right\rangle \right| + \left| \left\langle \tilde{r}^{(l,e_1)} - \tilde{r}^{(l+1,\nu_1)}, \tilde{r}^{(l,e_2)} \right\rangle \right| + \left| \left\langle \tilde{r}^{(l,e_1)}, \tilde{r}^{(l,e_2)} - \tilde{r}^{(l+1,\nu_2)} \right\rangle \right|$$

$$+ \left| \left\langle \tilde{r}^{(l,e_1)} - \tilde{r}^{(l+1,\nu_1)}, \tilde{r}^{(l,e_2)} - \tilde{r}^{(l+1,\nu_2)} \right\rangle \right| \quad (16)$$

$$\le \varepsilon + 2\sqrt{2}\left(\delta + 2\iota\right) + 2\left(\delta + 2\iota\right)^2,$$

where the last inequality is from (14). Then we finish the proof of this lemma. □

### A.3 COMPATIBILITY BETWEEN LOAD BALANCE CONDITION AND IN- AND CROSS-LAYER REGULARIZATION

In this section, we present the complete statements and proofs of Proposition3 and Proposition4 so as to prove the Takeaway 3 which regards the compatibility between the load balancing condition, the intra-layer specialization loss, and the cross-layer coupling loss.

Before presenting the proof, we note that *exact* load balancing is not achievable when the batch size is not divisible by the number of experts. However, since the imbalance per expert is at most one token, and the batch size in practice is large, this discrepancy is negligible. Thus, in this subsection, we assume the batch size is divisible by the number of experts without loss of generality.

The following proposition demonstrates that load balancing can be maintained under conditions of expert orthogonality, illustrating the compatibility between the intra-layer specialization loss and load balancing:

**Proposition 3.** *Suppose $k = 1$. For $e = 1, 2, \cdots, E$, denote $\mathcal{P}^{(l,e)}$ as the input space in which all the token can activate the e-th expert in layer l. Then there is always possible that the token space $\mathcal{P}^{(l,1)}, \mathcal{P}^{(l,2)}, \cdots, \mathcal{P}^{(l,E)}$ are convex, connected, and disjoint. Moreover, each $\mathcal{P}^{(l,e)}$ contains $B/E$ elements for the batch of input tokens. (See proof in Appendix A.3.)*

*Proof.* Since $E \mid B$, let $m = \dfrac{B}{E}$. We aim to partition the input set $\{x_1^{(l,e)}, x_2^{(l,e)}, \cdots, x_B^{(l,e)}\} \subset \mathbb{R}^h$ into $E$ convex connected subsets of equal size $m$. Pick any nonzero vector $a \in \mathbb{R}^h$. For each token $x_i^{(l,e)}$, compute the scalar projection

$$u_i = a^\top x_i^{(l,e)}, \quad i = 1, \ldots, N. \tag{17}$$

Without loss of generality, sort them in increasing order:

$$u_{(1)} \leq u_{(2)} \leq \cdots \leq u_{(N)}. \tag{18}$$

As $N_E \mid N$, we can divide this ordered list into $E$ consecutive blocks of size $m$. Specifically, denote

$$B_e := \{u_{((e-1)m+1)}, \, u_{((e-1)m+2)}, \, \ldots, \, u_{(em)}\}, \quad e = 1, \ldots, E. \tag{19}$$

Now define $E - 1$ hyperplanes of the form

$$H_e = \{\, x \in \mathbb{R}^n : a^\top x + b_r = 0\}, \qquad e = 1, \ldots, E - 1, \tag{20}$$

where each $b_e$ is chosen to satisfy that $-b_e \in \big(u_{(em)}, u_{(em+1)}\big)$, which means that the hyperplane lies strictly between the last element of block $B_r$ and the first element of block $B_{e+1}$.

These hyperplanes split $\mathbb{R}^h$ into $E$ slabs:

$$P_r = \{\, x : a^\top x \in [\alpha_{e-1}, \alpha_e]\}, \quad e = 1, \ldots, E, \tag{21}$$

where $\alpha_0 < \alpha_1 < \cdots < \alpha_E$ are thresholds satisfying

$$u_{(em)} < \alpha_e < u_{(em+1)}, \quad e = 1, \ldots, E - 1, \tag{22}$$

and we set $\alpha_0 = -\infty$, $\alpha_E = +\infty$ for completeness.

Each region $P_e$ is convex (intersection of halfspaces), connected, and by construction contains exactly $m = B/E$ tokens. Thus, if we let $\mathcal{P}^{(l,e)} = P_e$, we obtain a partition of the token space into $E$ disjoint convex connected subset with equal token counts, which proves the theorem. $\qquad\square$

Then we consider the compatibility between the coupling loss. Formally, for a given input $x_i^{(l)}$ and expert $e = 1, 2, \cdots, E$, define a binary variable:

$$f_i^{(l,e)} := \chi(\text{The expert } e \text{ in layer } l \text{ is activated})$$

where $\chi$ denotes the indicator function. With the definition of $f_i^{(l,e)}$, we can present the following proposition:

**Proposition 4.** *If we define the coupling loss $\mathcal{R}_{cp}(x_i)$ as Eq. (6), there exists a state that $\mathcal{L}_{cp}$ reach the optimal when satisifying the load balance condition*

$$\sum_{i=1}^{B} f_i^{(l,e)} = \sum_{i=1}^{B} f_i^{(l,\nu)}$$

*for any $e, \nu \in \{1, 2, \cdots, E\}$. (See proof in Appendix A.3.)*

*Proof.* Denote the coupling loss for one given token batch as $\mathcal{L}_{\mathrm{cp}} := \sum_{i=1}^{B} \mathcal{R}_{cp}(x_i)$, then we have:

$$\mathcal{L}_{\mathrm{cp}} = -\sum_{i=1}^{B}\sum_{l=1}^{L-1}\sum_{e=1}^{E}\sum_{\nu \in \mathbb{T}_i^{(l,e)}} s_i^{(l,e)} s_i^{(l+1,\nu)} \geq -\log \sum_{i=1}^{B}\sum_{l=1}^{L-1}\sum_{e=1}^{E} s_i^{(l,e)} \quad = -\log\left(B(L-1)\right), \tag{23}$$

where the equality condition is that for any $\nu \notin \mathbb{T}_i^{(l,e)}$ it holds

$$P_i^{(l,(e,\nu))} = 0, \tag{24}$$

for $i = 1, 2, \cdots, B$ and $e = 1, 2, \cdots, E$.

Recall the load balance condition

$$\sum_{i=1}^{B} f_i^{(l,e)} = \sum_{i=1}^{B} f_i^{(l,\nu)}, \tag{25}$$

where $e, \nu \in \{1, 2, \cdots, E\}$. We now prove that (24) and (25) can be simultaneously satisfied by explicitly constructing the desired condition.

We denote

$$[n] := \{1, 2, \ldots, n\}, \qquad [n]^k := \underbrace{[n] \times \cdots \times [n]}_{k \text{ times}}.$$

And we also define modular addition on $\{1, \ldots, n\}$ by

$$a \oplus_n b := \big((a - 1 + b) \bmod n\big) + 1.$$

Consider $\iota = (\iota_1, \ldots, \iota_k)$, any array in $[E]^k$. We define the following class of functions:

$$\mathcal{F}_{B,E,k} := \left\{ f : [B] \to [E]^k \ \middle|\ \forall i = 1, 2, \cdots, B, \ f(i) := \big(\iota_1 \oplus_{N_E} (i - 1), \ldots, \iota_k \oplus_{N_E} (i - 1)\big) \right\}.$$

Equivalently in component form, it holds that

$$f(i) = \big(f(i)_1, \ldots, f(i)_k\big), \quad (f(i))_r = \big((s_r - 1) + (i - 1)\big) \bmod N_E + 1, \quad r = 1, \ldots, k, \tag{26}$$

where $(f(i))_r$ denotes the $\mathcal{R}$-th element of $f(i)$. Then implies the recursion that

$$\forall i \in \{1, \ldots, B\}, \ \forall r \in \{1, \ldots, k\}, \quad f(i+1)_r = f(i)_r \oplus_{N_E} 1, \qquad f(N_E + 1) = f(1). \tag{27}$$

Now taking any collection of parameters $\eta_i^{(l,\kappa)}$ for $i = 1, 2, \cdots, B$, $l = 1, 2, \cdots, L$, and $\kappa = 1, 2, \cdots, k$ subject to the normalization constraint

$$\sum_{\kappa=1}^{k} \eta_i^{(l,\kappa)} = 1, \quad \forall l \in \{1, 2, \cdots, L\} \text{ and } i \in \{1, 2, \cdots, B\}. \tag{28}$$

We also take $f_1, f_2, \ldots, f_L \in \mathcal{F}_{B,E,k}$. Then define the routing scores $\eta_i^{(l,e)}$ by

$$s_i^{(l,e)} = \begin{cases} \eta_i^{(l,\kappa)}, & \text{if } e = (f_l(i))_\kappa \text{ for some } \kappa \in \{1, \ldots, k\}, \\ 0, & \text{otherwise.} \end{cases} \tag{29}$$

We now verify that the term $s_i^{(l,e)}$ defined in (29) satisfies conditions (24) and (25). Specifically, for Eq. (24) we have

$$P_i^{(l,(e,\nu))} := s_i^{(l,e)} s_i^{(l+1,\nu)} = \begin{cases} \eta_i^{(l,\kappa_1)} \eta_i^{(l+1,\kappa_2)}, & \text{if } e = (f_l(i))_{\kappa_1}, \ \nu = (f_{l+1}(i))_{\kappa_2}, \\ 0, & \text{otherwise.} \end{cases} \tag{30}$$

Thus $\mathbb{T}_i^{(l,e)}$ equals to the set of all the elements of $f_{l+1}(i)$. And for any $\nu \notin \mathbb{T}_i^{(l,e)}$, it holds that $P_i^{(l,(e,\nu))} = 0$.

Moreover, we consider Eq. (25). Recall the recursion property (27) of the selected function. Since $E|B$, in each layer every expert is loaded exactly $\dfrac{Bk}{R}$, which directly gives (25). $\qquad \square$

Table 5: Mixture-of-Experts (MoE) model configurations and training data volumes. 'A. Experts' denotes the activated experts and 'A. Params' denotes the activate parameters.

| Model size | Experts | A. Experts | Params | A. Params | Training Tokens |
|---|---|---|---|---|---|
| Small | 16 | 2 | 0.4B | 80M | 30B |
| Medium | 64 | 4 | 1.1B | 100M | 30B |
| Large | 96 | 6 | 7.0B | 500M | 50B |

Table 6: Ablations for two MoE architectures; metric is perplexity ($\downarrow$).

| Model | $\mathcal{L}_{lb}$ | $\mathcal{L}_{lb,sp}$ | $\mathcal{L}_{lb,cp}$ | $\mathcal{L}_{lb,sp,cp}$ | $\mathcal{L}_{lb,z}$ | $\mathcal{L}_{lb,z,sp}$ | $\mathcal{L}_{lb,z,cp}$ | $\mathcal{L}_{lb,z,sp,cp}$ |
|---|---|---|---|---|---|---|---|---|
| Vanilla MoE | 12.50 | 12.44 | 12.33 | 12.27 | 12.33 | 12.27 | 12.21 | 12.17 |
| DeepSeek-style MoE | 12.33 | 12.29 | 12.22 | 12.16 | 12.07 | 12.05 | 12.00 | 11.99 |

# B EXPERIMENTAL DETAILS AND ADDITIONAL EXPERIMENTS FOR PRE-TRAINING TASKS

## B.1 EXPERIMENTAL SETUP

**Infrastructure.** We integrate two auxiliary loss functions into the Megatron-LM framework (Shoeybi et al., 2019) as a plug-and-play module. By setting the corresponding hyperparameters, these losses can be enabled during MoE training.

**Model architecture.** We evaluate two MoE variants at multiple scales. For the vanilla MoE, we adopt a mainstream design comprising RMS normalization (Zhang & Sennrich, 2019), SwiGLU activations (Shazeer, 2020), and rotary position embeddings (RoPE) (Su et al., 2024); architectural hyperparameters are listed in Table 5. For the DeepSeek-style MoE, we augment the vanilla design with **ONE** shared expert and employ the **auxiliary-loss-free** load balancing strategy (Dai et al., 2024). The hyperparameters $\lambda_{cp}$ and $\lambda_{sp}$ are set to $1 \times 10^{-3}$ and $2 \times 10^{-3}$, respectively. Unless otherwise noted, the load-balance loss weight is set to $1 \times 10^{-2}$, the z-loss weight $\mathcal{R}_z$ (Zoph et al., 2022) to $1 \times 10^{-3}$, and the update step size for the coefficient $b$ in the auxiliary-loss-free load balancing strategy to $1 \times 10^{-3}$.

**Training settings.** Training is performed on the C4-en dataset (Raffel et al., 2020) using the LLaMA-2 tokenizer (Touvron et al., 2023). The small and medium MoE models are trained for 30 billion tokens, and the large MoE model for 50 billion tokens. This token budget exceeds the data size suggested by MoE scaling laws (Clark et al., 2022), providing sufficient signal for convergence. We use AdamW (Loshchilov & Hutter, 2017) optimizer with moment coefficient $\beta_1 = 0.9$, $\beta_2 = 0.999$, and weight decay coefficient 0.1.

## B.2 ABLATION STUDY FOR DIFFERENT REGULARIZATIONS

To evaluate the influence of various regularization techniques on model performance, we performed an ablation study utilizing a medium-scale architecture. The outcomes of this investigation are summarized in Table 6. Our analysis identifies several consistent trends. First, each auxiliary objective demonstrates individual efficacy: for the Vanilla MoE model, the introduction of $\mathcal{R}_{sp}$ leads to a reduction in perplexity, whereas $\mathcal{R}_{cp}$ produces a more substantial improvement. Similarly, in the DeepSeek-style MoE, both regularizers enhance performance, with $\mathcal{R}_{cp}$ yielding a greater effect. Moreover, the two losses exhibit complementarity, as their combined application results in further gains. When integrated with additional components such as $\mathcal{R}_{lb}$, the full regularization set achieves the most pronounced enhancements across both model variants. These patterns indicate that the specialization and coupling mechanisms independently contribute to refining expert behavior and, when employed together, synergize to produce cumulative reductions in perplexity.

Table 7: The load-balance loss during the training process with different loss.

| step | 10K | 20K | 30K | 40K | 50K | 60K |
|------|-----|-----|-----|-----|-----|-----|
| $\mathcal{L}_{\text{lb}}$ | 0.99676 | 0.99635 | 0.99613 | 0.99595 | 0.99566 | 0.99541 |
| $\mathcal{L}_{\text{lb,cp}}$ | 0.99728 | 0.99671 | 0.99648 | 0.99627 | 0.99597 | 0.99570 |
| $\mathcal{L}_{\text{lb,sp}}$ | 0.99709 | 0.99659 | 0.99635 | 0.99616 | 0.99585 | 0.99560 |
| $\mathcal{L}_{\text{lb,sp,cp}}$ | 0.99734 | 0.99683 | 0.99661 | 0.99642 | 0.99611 | 0.99585 |

Table 8: The intra-layer specialization loss $\mathcal{R}_{\text{sp}}$ over training iterations ($\downarrow$).

| **Iterations** | $\mathcal{L}_{\text{lb,sp}}$ | $\mathcal{L}_{\text{lb,sp,cp}}$ |
|------|------|------|
| 30K | 0.020476 | **0.019136** |
| 60K | 0.019924 | **0.018862** |

## B.3 THE IMPACT OF AUXILIARY LOSS ON LOAD BALANCE LOSS

To investigate the impact of the proposed regularization terms $\mathcal{R}_{\text{sp}}$ and $\mathcal{R}_{\text{cp}}$ on the primary training objective, we analyze the load balance loss curves throughout the training process. Table 7 compares the load balance loss of the full model $\mathcal{R}_{\text{lb,sp,cp}}$ with ablation studies involving the baseline configurations $\mathcal{R}_{\text{lb}}$, $\mathcal{R}_{\text{lb,cp}}$, and $\mathcal{R}_{\text{lb,sp}}$.

As illustrated in the main plot, all model variants exhibit rapid and stable convergence. The load balance loss for each configuration declines sharply within the initial 5,000 steps and stabilizes promptly near its optimal value. This demonstrates that incorporating the auxiliary objectives does not hinder the model's capacity to learn the primary load balancing task.

A close examination of the final 10,000 training steps enables a more detailed comparison. Although the inclusion of $\mathcal{R}_{\text{sp}}$ and $\mathcal{R}_{\text{cp}}$ leads to a slight elevation in the final load balance loss, the extent of this increase is negligible. Specifically, at the 60,000-step point, the baseline model $\mathcal{R}_{\text{lb}}$ attains a loss of approximately 0.9955, while the full model with both auxiliary losses ($\mathcal{R}_{\text{lb,sp,cp}}$) reaches approximately 0.9958. This constitutes a minor deviation of less than **0.04%**, which is inconsequential.

## B.4 TRAINING DYNAMICS OF SPECIALIZATION AND COUPLING LOSSES

To provide concrete evidence for the narrative of expert specialization and path stabilization, we analyze the evolution of the proposed auxiliary losses during training. We monitor the intra-layer specialization loss $\mathcal{R}_{\text{sp}}$ under both the baseline objective $\mathcal{L}_{\text{lb,sp}}$ and our full objective $\mathcal{L}_{\text{lb,sp,cp}}$, along with the cross-layer coupling loss $\mathcal{R}_{\text{cp}}$ under $\mathcal{L}_{\text{lb,sp,cp}}$, as summarized in Table 8 and Table 9. Lower values of $\mathcal{R}_{\text{sp}}$ indicate stronger specialization, while for $\mathcal{R}_{\text{cp}}$—defined as a negative quantity in Eq. (6)—more negative values correspond to stronger coupling (i.e., higher joint routing probability along expert paths).

The results reveal several key trends. First, $\mathcal{R}_{\text{sp}}$ decreases rapidly during the early phase of training (1K–10K iterations) and continues to decline, albeit at a slower rate, throughout the entire pretraining schedule up to 60K iterations, without early saturation. Second, the addition of the coupling term consistently leads to lower $\mathcal{R}_{\text{sp}}$ values at every checkpoint compared to the baseline, indicating that cross-layer coupling enhances intra-layer specialization over the course of pretraining. Third, $\mathcal{R}_{\text{cp}}$ exhibits a similar pattern of rapid initial improvement followed by steady progression, becoming progressively more negative across all stages of training. This demonstrates that expert paths continue to sharpen rather than stabilize prematurely.

In summary, both auxiliary objectives exhibit continuous improvement throughout pretraining: $\mathcal{R}_{\text{sp}}$ steadily decreases (with consistently lower values under $\mathcal{L}_{\text{lb,sp,cp}}$ than under $\mathcal{L}_{\text{lb,sp}}$), and $\mathcal{R}_{\text{cp}}$ becomes increasingly negative up to 60K steps. These dynamics substantiate the claim that experts continue to specialize and paths continue to stabilize over time, offering deeper insight beyond downstream accuracy metrics alone.

## B.5 PRE-TRAINING RESULTS WITH RANDOM SEEDS

Table 9: Cross-layer coupling loss $\mathcal{R}_{cp}$ under $\mathcal{L}_{lb,sp,cp}$ (more negative is better).

| Iterations | 0.5K | 1K | 3K | 5K | 10K | 20K | 30K | 60K |
|---|---|---|---|---|---|---|---|---|
| $\mathcal{R}_{cp}$ | -0.2662 | -0.2845 | -0.2986 | -0.3051 | -0.3115 | -0.3186 | -0.3228 | **-0.3321** |

Table 10: Validation perplexity for medium model scale with three random repetitions ($\downarrow$).

| Losses | Vanilla MoE | DeepSeek-style MoE |
|---|---|---|
| $\mathcal{L}_{lb}$ | 12.50 (0.01) | 12.33 (0.02) |
| $\mathcal{L}_{lb,sp,cp}$ | 12.26 (0.01) | 12.15 (0.02) |
| $\mathcal{L}_{lb,z}$ | 12.33 (0.02) | 12.07 (0.01) |
| $\mathcal{L}_{lb,z,sp,cp}$ | 12.17 (0.01) | 11.98 (0.01) |

To rigorously demonstrate that the reported improvements are attributable to the auxiliary regularization and are statistically significant rather than resulting from optimization noise, we conducted repeated pre-training experiments using medium-sized models for both the Vanilla MoE and DeepSeek-style architectures. The experimental configuration remains identical to that described in Appendix B.1.

As illustrated in the Table 10, for the Vanilla MoE, the comparison between $\mathcal{L}_{lb}$ versus $\mathcal{L}_{lb,sp,cp}$ shows an improvement from 12.50 to 12.26, corresponding to an approximately 1.9% relative reduction, with a standard deviation across seeds of only 0.01. Furthermore, when all auxiliary terms are included, $\mathcal{L}_{lb,z}$ versus $\mathcal{L}_{lb,z,sp,cp}$ improves from 12.33 to 12.17 , with standard deviations ranging from 0.01 to 0.02. These findings confirm consistent and statistically meaningful gains in validation performance.

### B.6 HYPERPARAMETER SENSITIVITY IN PRE-TRAINING

The validation performance for the pre-training tasks, as presented in Table 2, is based on a fixed hyperparameter selection described in Appendix B.1. To examine the sensitivity of the hyperparameters $\lambda_{cp}$ and $\lambda_{sp}$, we performed a hyperparameter sweep around the default values using a medium-sized model. Validation perplexity (where lower values indicate better performance) was measured under the following variations:

- With $\lambda_{sp}$ fixed at $2 \times 10^{-3}$, we varied $\lambda_{cp}$ from 0.2 to 2 times the default value of $1 \times 10^{-3}$.
- With $\lambda_{cp}$ fixed at $1 \times 10^{-3}$, we varied $\lambda_{sp}$ from 0.5 to 3 times the default value of $2 \times 10^{-3}$.

The results, shown in Tables 11 and 12, demonstrate that the model performance remains stable across a broad interval. The perplexity changes are limited to less than 1% relative to the optimum. The heuristic choice of $\lambda_{cp} = 10^{-3}$ and $\lambda_{sp} = 2 \times 10^{-3}$ yields near-optimal results, and deviations cause only minor degradation.

The sensitivity study and scaling rules will be detailed in the appendix of the revised manuscript to emphasize the robustness of the method, which does not require meticulous, model-dependent hyperparameter tuning.

### B.7 QUANTITATIVE COMPARISON WITH DEEPSEEKMOE-STYLE LOAD BALANCING

To directly address whether training with our proposed specialization induces more expert specialization than DeepSeekMoE's auxiliary-loss-free load balancing, we compare two training objectives including $\mathcal{L}_{lb}$ and $\mathcal{L}_{lb,cp,sp}$ over the small model with configurations in Table 5. As a proxy for expert specialization and routing coherence, we measure every 1000 iterations the percentage of tokens whose top-1 expert assignment remains unchanged between consecutive checkpoints. Higher values correspond to more stable token–expert assignments, lower routing entropy, and, via Proposition 4, more persistent expert-specific gradient directions.

Table 11: Perplexity with fixed $\lambda_{sp}$ and varied $\lambda_{cp}$.

| $\lambda_{cp}$ | $2 \times 10^{-4}$ | $5 \times 10^{-4}$ | $1 \times 10^{-3}$ | $2 \times 10^{-3}$ |
|---|---|---|---|---|
| PPL | 12.41 | 12.35 | **12.27** | 12.30 |

Table 12: Perplexity with fixed $\lambda_{cp}$ and varied $\lambda_{sp}$.

| $\lambda_{sp}$ | $5 \times 10^{-3}$ | $1 \times 10^{-3}$ | $2 \times 10^{-3}$ | $3 \times 10^{-3}$ |
|---|---|---|---|---|
| PPL | 12.32 | 12.30 | **12.27** | 12.29 |

Table 13: The fraction of tokens that keep the same experts between checkpoints.

| Iteration range | 1K–2K | 2K–3K | 4K–5K | 9K–10K | 19K–20K | 29K–30K | 59K–60K |
|---|---|---|---|---|---|---|---|
| $\mathcal{L}_{lb,sp}$ | 0.4746 | 0.6056 | 0.6601 | 0.6987 | 0.7450 | 0.7864 | 0.9011 |
| $\mathcal{L}_{lb,sp,cp}$ | 0.4898 | 0.6213 | 0.6757 | 0.7187 | 0.7594 | 0.7935 | 0.9067 |

Table 14: Hyperparameters for the fine-tuning task under Qwen3-30B models.

| Hyperparameters | Value |
|---|---|
| Global batch size | 64 |
| Learning rate | 8e-5 |
| Epochs | 3 |
| Sequence length | 32768 |
| $\lambda_{lb}^{\diamond}$ | 1e-3 |
| $\lambda_{sp}$ | 2e-4 |
| $\lambda_{cp}$ | 1e-4 |

$^{\diamond}$ The coefficient of the regularization of load-balancing.

The results, as detailed in Table 3, demonstrate that across all training stages, $\mathcal{L}_{lb,cp,sp}$ consistently enhances routing stability by 1–2 absolute points (approximately 2%–4% relative improvement) compared to $\mathcal{L}_{lb}$. The gains are especially significant during early training phases when routing ambiguity is most severe, which aligns precisely with the regime addressed by Proposition 2. Furthermore, the benefits persist even at later stages (e.g., 59K–60K iterations), indicating that expert assignments maintain greater consistency over time.

In conjunction with Table 13, where $\mathcal{L}_{lb,sp,cp}$ reduces the intra-layer specialization loss $R_{sp}$ relative to $\mathcal{L}_{lb,sp}$, these findings confirm that our method further sharpens expert differentiation beyond the capabilities of auxiliary-loss-free load balancing alone. This improvement is consistent with our theoretical framework: reduced activation similarity promotes more orthogonal gradients (as per Proposition 4), while enhanced routing stability supports stronger and more coherent expert paths (in line with Proposition 2).

## C  FINETUNING TASKS EVALUATION

We fine-tune `Qwen3-30B-A3B-Instruct-2507` model on an internal corpus of 38B tokens under identical training hyperparameters listed in Table 14. We evaluate on a broad suite of reasoning and knowledge-intensive benchmarks, including HumanEval (Chen et al., 2021), GSM8K (Cobbe et al., 2021), math500_PRM800K_dataset (Lightman et al., 2023), and MMLU (Hendrycks et al., 2021). Across nearly all settings, incorporating $\mathcal{R}_{cp}$ and $\mathcal{R}_{sp}$ outperforms the baseline, yielding consistent gains on reasoning-oriented tasks as well as aggregate knowledge measures as Table 15. While a minor fluctuation is observed on the humanities subset of MMLU, the overall trend remains positive, confirming that our objectives not only sharpen specialization in pre-training but also transfer effectively to finetuning adaptation.

We further provide the detailed per-dataset results corresponding to Table 4, covering all reasoning and MMLU subject benchmarks. As shown in Table 16. We observe consistent improvements across most of the datasets. While minor fluctuations are observed in some individual categories (e.g., humanities), the overall trend remains positive.

Table 15: Evaluation score on Qwen3-30B-A3B-Instruct-2507 finetuning tasks. The last four rows stands for the performance for mmlu dataset with different domains.

| Dataset | Metric | $\mathcal{L}_{\text{lb}}$ | $\mathcal{L}_{\text{lb,sp,cp}}$ |
|---|---|---|---|
| openai_humaneval | humaneval_pass@1 | 92.07 | **95.73** |
| gsm8k | accuracy | 93.33 | **94.16** |
| math_prm800k_500 | accuracy | 94.00 | **94.20** |
| mmlu | naive_average | 78.97 | **79.86** |
| mmlu-weighted | weighted_average | 76.35 | **77.10** |
| mmlu-humanities | naive_average | **75.90** | 75.42 |
| mmlu-stem | naive_average | 87.38 | **88.97** |
| mmlu-social-science | naive_average | 75.91 | **77.11** |
| mmlu-other | naive_average | 72.59 | **73.52** |

Table 16: Full evaluation of *Qwen3-30B-A3B-Instruct-2507*. Our method adds $\mathcal{R}_{\text{cp}}(\lambda_{\text{cp}} = 2 \times 10^{-4})$ and $\mathcal{R}_{\text{sp}}(\lambda_{\text{sp}} = 10^{-4})$.

| Dataset | $\mathcal{L}_{\text{lb}}$ | $\mathcal{L}_{\text{lb,sp,cp}}$ | Dataset | $\mathcal{L}_{\text{lb}}$ | $\mathcal{L}_{\text{lb,sp,cp}}$ |
|---|---|---|---|---|---|
| HumanEval (pass@1) | 92.07 | **95.73** | GSM8K (accuracy) | 93.33 | **94.16** |
| PRM800K-500 | 94.00 | **94.20** | College Biology | 85.42 | **90.97** |
| College Chemistry | 73.00 | **78.00** | College CS | 90.00 | **90.00** |
| College Math | 97.00 | **98.00** | College Physics | 97.06 | **97.06** |
| Elec. Engineering | 81.38 | 80.00 | Astronomy | 84.87 | **87.50** |
| Anatomy | 68.89 | **73.33** | Abstract Algebra | 96.00 | **97.00** |
| Machine Learning | 83.04 | **83.93** | Clinical Knowledge | 78.49 | 75.09 |
| Global Facts | 55.00 | 51.00 | Management | 66.99 | **71.84** |
| Nutrition | 73.86 | **79.74** | Marketing | 79.49 | **80.77** |
| Prof. Accounting | 81.91 | **82.27** | High School Geography | 81.82 | 74.75 |
| International Law | 77.69 | 76.03 | Moral Scenarios | 69.61 | 68.38 |
| Computer Security | 74.00 | **79.00** | HS Microeconomics | 86.55 | **88.24** |
| Professional Law | 56.45 | **58.41** | Medical Genetics | 83.00 | **85.00** |
| Prof. Psychology | 72.22 | **72.71** | Jurisprudence | 80.56 | 74.07 |
| World Religions | 78.36 | 77.78 | Philosophy | 69.45 | **71.70** |
| Virology | 52.41 | 48.19 | HS Chemistry | 90.64 | **91.63** |
| Public Relations | 54.55 | **62.73** | HS Macroeconomics | 86.92 | 85.13 |
| Human Sexuality | 77.10 | **80.15** | Elementary Math | 94.44 | **94.97** |
| HS Physics | 92.05 | 91.39 | HS Computer Science | 89.00 | **91.00** |
| HS European Hist. | 78.79 | **79.39** | Business Ethics | 66.00 | **74.00** |
| Moral Disputes | 67.92 | **70.52** | HS Statistics | 89.81 | **90.74** |
| Miscellaneous | 79.05 | 78.54 | Formal Logic | 91.27 | **93.65** |
| HS Gov/Politics | 80.31 | **80.83** | Prehistory | 75.62 | 75.00 |
| Security Studies | 64.08 | **68.57** | HS Biology | 85.16 | **86.77** |
| Logical Fallacies | 80.37 | **82.21** | HS World History | 80.17 | 76.79 |
| Prof. Medicine | 79.41 | **82.72** | HS Math | 97.41 | **98.15** |
| College Medicine | 78.03 | **78.03** | HS US History | 80.39 | 76.47 |
| Sociology | 69.65 | **77.11** | Econometrics | 78.07 | **78.95** |
| HS Psychology | 79.63 | **81.10** | Human Aging | 69.96 | 68.61 |
| US Foreign Policy | 80.00 | 75.00 | Conceptual Physics | 91.06 | **91.06** |

## D INFERENCE ACCELERATION

To leverage the benefits of the specialization loss and coupling loss during the inference period, we implement a path-aware placement and bucketing strategy. This involves estimating a cross-layer expert co-activation matrix from a held-out dataset, greedily co-locating strongly coupled experts on the same GPU shard via graph partitioning, and performing a lightweight pre-routing pass through the first MoE router to bucket sequences according to early expert decisions. These buckets are

Table 17: Throughput comparison (samples/s; ↑) on four standard benchmarks. Here 'SO' means the system optimization.

| Model size | Loss | MMLU | GSM8K | HumanEval | Math500 |
|---|---|---|---|---|---|
| Small | $\mathcal{L}_{lb}$ *w.o.* SO | 161.3 (1.00×) | 26.2 (1.00×) | 35.7 (1.00×) | 6.9 (1.00×) |
|  | $\mathcal{L}_{lb}$ *w.* SO | 164.9 (1.03×) | 26.5 (1.01×) | 36.6 (1.02×) | 7.0 (1.01×) |
|  | $\mathcal{L}_{lb,sp,cp}$ *w.* SO | 170.4 (1.06×) | 27.0 (1.03×) | 37.5 (1.05×) | 7.1 (1.03×) |
| Medium | $\mathcal{L}_{lb}$ *w.o.* SO | 157.4 (1.00×) | 25.9 (1.00×) | 27.7 (1.00×) | 6.1 (1.00×) |
|  | $\mathcal{L}_{lb}$ *w.* SO | 162.7 (1.03×) | 26.2 (1.01×) | 28.6 (1.03×) | 6.2 (1.01×) |
|  | $\mathcal{L}_{lb,sp,cp}$ *w.* SO | 165.7 (1.05×) | 26.6 (1.03×) | 29.4 (1.06×) | 6.3 (1.03×) |
| Large | $\mathcal{L}_{lb}$ *w.o.* SO | 96.9 (1.00×) | 15.0 (1.00×) | 12.8 (1.00×) | 3.9 (1.00×) |
|  | $\mathcal{L}_{lb}$ *w.* SO | 96.9 (1.00×) | 15.0 (1.00×) | 12.8 (1.00×) | 3.9 (1.00×) |
|  | $\mathcal{L}_{lb,sp,cp}$ *w.* SO | 103.5 (1.07×) | 15.7 (1.05×) | 13.7 (1.07×) | 4.2 (1.08×) |

Table 18: The iteration time and peak memory with different loss

| Model size | Loss | Iteration time (ms/iteration) | Peak memory (GB) |
|---|---|---|---|
| Small | $\mathcal{L}_{lb}$ | 405.9 (1.0000×) | 43.5 (1.0000×) |
|  | $\mathcal{L}_{lb,sp,cp}$ | 413.6 (1.0190×) | 43.6 (1.0023×) |
| Medium | $\mathcal{L}_{lb}$ | 518.4 (1.0000×) | 60.6 (1.0000×) |
|  | $\mathcal{L}_{lb,sp,cp}$ | 526.5 (1.0156×) | 60.7 (1.0016×) |
| Large | $\mathcal{L}_{lb}$ | 2927.8 (1.0000×) | 73.1 (1.0000×) |
|  | $\mathcal{L}_{lb,sp,cp}$ | 2942.4 (1.0049×) | 73.3 (1.00027×) |

then assigned to shards hosting the corresponding experts, ensuring that most subsequent dispatches remain local.

We evaluate our approach on MoE models of varying scales under 8 Nvidia A100 80G GPUs with expert parallelism. The number of parallel devices are set to 8 and the microbatch size is set to 1. A baseline model trained solely with $\mathcal{R}_{lb}$ is compared against our variant trained with $\mathcal{R}_{lb,sp,cp}$. While the baseline uses default round-robin expert placement and uniform batching, our model employs the path-aware scheme described above. We also apply identical system-level optimizations to both the load-balancing baseline and our model. This design cleanly separates the acceleration attributable to engineering infrastructure from that enabled by structural properties—specifically, stronger cross-layer expert coupling and lower routing entropy—induced by our proposed losses.

As summarized in Table 17, throughput improves consistently across model sizes and benchmarks—without any architectural modifications or additional parameters. These results demonstrate that reducing routing ambiguity through $\mathcal{R}_{sp}$ and $\mathcal{R}_{cp}$ directly enhances system-level efficiency by streamlining token-to-expert execution paths. With the inference throughput, it can be observed that our proposed auxiliary losses improve model perplexity while simultaneously enhancing inference efficiency through reduced routing entropy. By promoting sharper expert specialization and stronger cross-layer coupling, tokens follow more consistent expert paths, which in an expert parallelism setup improves cache locality and reduces All-to-All communication overhead.

# E COMPARISON WITH RECENT AUXILIARY-LOSS METHODS FOR SPECIALIZATION

Several recent studies have introduced auxiliary loss functions aimed at improving expert specialization and routing efficacy in Mixture-of-Experts (MoE) models. In this section, we present a conceptual analysis and empirical evaluation comparing our approach with a representative method by (Guo et al., 2025a), which combines an orthogonality loss with a *variance* loss applied to the routing logits.

Table 19: Validation perplexity for the small size model under different kinds of loss ($\downarrow$).

| Method | $\mathcal{L}_{\text{lb}}$ | $\mathcal{L}_{\text{lb,sp,cp}}$ | $\mathcal{L}_{\text{lb,o,v}}$ |
|---|---|---|---|
| PPL | 12.50 | **12.27** | 24.86 |

Table 20: Downstream evaluation performance across multiple 16B-class models.

| Method | Model | MMLU | MMLU-Pro | BBH | GPQA | MBPP | GSM8K | MATH500 |
|---|---|---|---|---|---|---|---|---|
| With Aux Loss | | $29.27_{\pm0.10}$ | $19.47_{\pm2.50}$ | $26.92_{\pm2.30}$ | $21.15_{\pm0.40}$ | $31.36_{\pm1.10}$ | $15.70_{\pm2.40}$ | $5.47_{\pm1.50}$ |
| Loss-Free Balancing | | $30.71_{\pm2.10}$ | $16.81_{\pm0.70}$ | $32.99_{\pm1.00}$ | $20.63_{\pm1.60}$ | $32.80_{\pm1.40}$ | $21.28_{\pm0.40}$ | $5.83_{\pm1.30}$ |
| GShard | DeepSeek-MoE-16B | $27.05_{\pm2.00}$ | $20.48_{\pm0.60}$ | $29.83_{\pm1.80}$ | $24.28_{\pm2.30}$ | $34.50_{\pm1.70}$ | $27.12_{\pm1.30}$ | $8.20_{\pm1.50}$ |
| ST-MoE | | $34.23_{\pm2.20}$ | $19.71_{\pm0.80}$ | $36.91_{\pm1.90}$ | $20.35_{\pm0.90}$ | $36.34_{\pm1.50}$ | $30.10_{\pm2.00}$ | $7.08_{\pm0.40}$ |
| $\mathcal{L}_{\text{lb,o,v}}$ | | $33.35_{\pm2.20}$ | $24.87_{\pm1.20}$ | $37.52_{\pm1.40}$ | $25.15_{\pm0.40}$ | $40.03_{\pm0.40}$ | $35.00_{\pm1.00}$ | $10.82_{\pm0.30}$ |
| $\mathcal{L}_{\text{lb,sp,cp}}$ | | $\mathbf{37.26}_{\pm0.40}$ | $\mathbf{26.32}_{\pm1.25}$ | $\mathbf{39.46}_{\pm0.34}$ | $\mathbf{27.58}_{\pm1.90}$ | $\mathbf{42.24}_{\pm0.20}$ | $\mathbf{36.02}_{\pm1.33}$ | $\mathbf{11.70}_{\pm0.40}$ |
| With Aux Loss | | $33.23_{\pm2.10}$ | $28.40_{\pm0.20}$ | $34.80_{\pm1.40}$ | $24.92_{\pm0.80}$ | $41.23_{\pm0.20}$ | $44.79_{\pm2.10}$ | $42.03_{\pm1.40}$ |
| Loss-Free Balancing | | $30.23_{\pm0.80}$ | $30.75_{\pm2.10}$ | $34.21_{\pm1.10}$ | $26.33_{\pm0.60}$ | $36.02_{\pm2.30}$ | $43.35_{\pm0.70}$ | $39.76_{\pm1.10}$ |
| GShard | DeepSeek-V2-Lite | $30.86_{\pm1.10}$ | $29.13_{\pm0.80}$ | $37.67_{\pm0.30}$ | $24.34_{\pm2.10}$ | $37.00_{\pm2.10}$ | $45.39_{\pm1.50}$ | $43.61_{\pm2.10}$ |
| ST-MoE | | $32.68_{\pm2.10}$ | $30.28_{\pm2.10}$ | $38.78_{\pm0.90}$ | $22.33_{\pm0.40}$ | $39.72_{\pm2.30}$ | $47.78_{\pm1.80}$ | $46.74_{\pm0.50}$ |
| $\mathcal{L}_{\text{lb,o,v}}$ | | $35.59_{\pm0.50}$ | $37.37_{\pm0.20}$ | $38.84_{\pm1.70}$ | $28.76_{\pm0.10}$ | $43.53_{\pm2.40}$ | $50.94_{\pm2.40}$ | $49.33_{\pm2.40}$ |
| $\mathcal{L}_{\text{lb,sp,cp}}$ | | $\mathbf{42.51}_{\pm0.56}$ | $\mathbf{39.42}_{\pm0.87}$ | $\mathbf{46.03}_{\pm0.97}$ | $\mathbf{31.04}_{\pm1.80}$ | $\mathbf{46.82}_{\pm1.10}$ | $\mathbf{52.64}_{\pm1.28}$ | $\mathbf{51.40}_{\pm1.20}$ |
| With Aux Loss | | $35.82_{\pm1.40}$ | $36.10_{\pm1.50}$ | $47.17_{\pm0.70}$ | $30.72_{\pm1.90}$ | $47.34_{\pm1.50}$ | $82.32_{\pm1.50}$ | $57.03_{\pm1.60}$ |
| Loss-Free Balancing | | $27.40_{\pm0.10}$ | $31.91_{\pm2.10}$ | $42.45_{\pm0.50}$ | $29.27_{\pm1.80}$ | $44.92_{\pm1.30}$ | $79.34_{\pm0.70}$ | $57.77_{\pm0.50}$ |
| GShard | Moonlight-16B-A3B | $36.06_{\pm0.90}$ | $30.65_{\pm0.50}$ | $49.20_{\pm1.70}$ | $31.13_{\pm1.10}$ | $49.85_{\pm0.50}$ | $84.62_{\pm0.80}$ | $56.09_{\pm2.20}$ |
| ST-MoE | | $33.03_{\pm0.90}$ | $26.83_{\pm1.70}$ | $46.78_{\pm0.30}$ | $30.93_{\pm1.50}$ | $47.97_{\pm2.20}$ | $84.45_{\pm0.90}$ | $57.61_{\pm1.60}$ |
| $\mathcal{L}_{\text{lb,o,v}}$ | | $40.36_{\pm2.20}$ | $34.90_{\pm0.30}$ | $52.42_{\pm1.80}$ | $32.01_{\pm0.90}$ | $47.77_{\pm1.00}$ | $87.62_{\pm2.20}$ | $59.64_{\pm0.20}$ |
| $\mathcal{L}_{\text{lb,sp,cp}}$ | | $\mathbf{51.74}_{\pm2.58}$ | $\mathbf{41.02}_{\pm0.87}$ | $\mathbf{62.56}_{\pm0.53}$ | $\mathbf{34.92}_{\pm1.60}$ | $\mathbf{53.32}_{\pm0.20}$ | $\mathbf{87.67}_{\pm1.10}$ | $\mathbf{59.85}_{\pm0.20}$ |

## E.1 CONCEPTUAL COMPARISON

Our method integrates two complementary components: the *intra-layer specialization* loss $\mathcal{L}_{\text{sp}}$, which promotes orthogonality in the representations of co-activated experts, thereby aligning their parameter gradients along orthogonal directions (see Proposition 1), and the *cross-layer coupling* loss $\mathcal{L}_{\text{cp}}$, which enforces consistency in expert selection across adjacent layers, reducing routing ambiguity (see Proposition 2). These losses operate on principles of *information geometry and path consistency* and are compatible with standard router-stabilization techniques, such as the $z$-loss and logit clipping.

In contrast, the approach by (Guo et al., 2025a) incorporates an orthogonality term along with a *variance-maximization* objective on the routing logits, explicitly encouraging high logit dispersion to enhance discrimination. While increased dispersion can sharpen top-$k$ selections, it lacks inherent control over logit magnitudes, potentially leading to adverse interactions with softmax temperature and $z$-loss penalties. Specifically, unregulated variance amplification often causes prematurely peaked routing distributions or numerical instabilities (e.g., gradient spikes), increasing sensitivity to learning rate and initialization in large-scale pre-training. Our design mitigates these issues by regularizing *activations and paths* rather than directly inflating raw logit variance.

## E.2 EXPERIMENTAL EVALUATION

We first evaluate pre-training stability using perplexity on the C4 dataset. We implemented the method from (Guo et al., 2025a) (denoted as $\mathcal{L}_{\text{lb,o,v}}$) and compared it against the baseline load-balancing loss $\mathcal{L}_{\text{lb}}$ and our full objective $\mathcal{L}_{\text{lb,sp,cp}}$. Pre-training with a medium-scale configuration on C4 yielded the results summarized in Table 19. As anticipated, maximizing routing-logit variance resulted in severe training instability, whereas our consistency-driven losses achieved lower perplexity than the baseline.

Next, we assess downstream performance across diverse benchmarks. We evaluated multiple models with approximately 16 billion parameters including DeepSeek-MoE-16B (Dai et al., 2024), DeepSeek-V2-Lite (Liu et al., 2024), and Moonlight-16B-A3B (Liu et al., 2025). The evaluation tasks include MMLU, MMLU-Pro, BBH, GPQA, MBPP, GSM8K, and MATH500. Even when applying stabilization measures to ensure the method from (Guo et al., 2025a) completes fine-tuning, our approach consistently achieves superior scores, as reported in Table 20.

In summary, our consistency-based formulation demonstrates stronger accuracy and stability. The combination of $\mathcal{L}_{\text{sp}}$ and $\mathcal{L}_{\text{cp}}$ outperforms variance-maximization methods in both pre-training and downstream settings. These gains align with our theoretical framework: (i) orthogonalizing expert activations yields orthogonal gradient directions, reducing parameter interference; (ii) cross-layer coupling concentrates routing probability mass along consistent paths, diminishing ambiguity and consolidating expert specialization. Together, these effects enhance final model quality and training dynamics for large-scale applications.

## F    ANALYSIS FOR THE COMPUTATIONAL AND MEMORY EFFICIENCY

In this section, we present a series of analysis for the computation and memory overhead for the gradient evaluation with our proposed auxiliary regularization.

### F.1    THEORETICAL ANALYSIS FOR COMPUTATIONAL AND MEMORY OVERHEAD

Here we present a theoretical analysis for computational and memory overhead. From a computational perspective, both losses are lightweight relative to the model's core operations (attention mechanisms and feed-forward networks)

**Intra-Layer Specialization Loss ($\mathcal{R}_{\text{sp}}$).**

- **Computational Complexity**: This loss requires computing pairwise cosine similarities between activations of the top-k selected experts. For a hidden dimension $d$ and $k$ activated experts, the per-token complexity is $\mathcal{O}(k^2 \cdot d)$. In standard MoE configurations, $k$ typically assumes small values (e.g., 2 or 4), while $d$ represents a large dimension (e.g., 4096). Consequently, $k^2 \ll d$, rendering the cost of $\mathcal{O}(k^2 \cdot d)$ negligible compared to the standard FFN transformation cost of $\mathcal{O}(k \cdot d^2)$.

- **Scalability**: Crucially, this computational cost depends solely on the number of activated experts $k$, rather than the total number of experts $E$. This implies that even as the total expert count $E$ scales to hundreds or thousands (as in "Mixture of Million Experts" architectures), as long as the activated expert count $k$ remains small, the computational overhead of $\mathcal{R}_{sp}$ remains both constant and minimal.

- **Memory Requirements**: No additional memory allocation is necessary, as this loss reuses intermediate activations $z^{(l,e)}$ already computed during the forward pass.

**Cross-Layer Coupling Loss ($\mathcal{R}_{\text{cp}}$).**

- **Computational Complexity**: This loss operates exclusively on scalar routing logits. Specifically, it involves basic statistical operations on token routing scores across consecutive layers. These operations avoid complex matrix computations involving high-dimensional hidden states.

- **Memory Requirements**: This loss requires storing a lightweight tensor of dimensions $E \times E \times L$ to track expert transition statistics. Since the number of experts $E$ is typically much smaller than the hidden dimension $d$, the memory consumption of this tensor is negligible.

### F.2    EMPIRICAL WALL-CLOCKED TIME AND MEMORY ANALYSIS

Empirical results are fully consistent with our theoretical complexity analysis. We systematically measured training throughput (in ms/iteration) and peak GPU memory consumption (in GB) across the Small (0.4B), Medium (1.1B), and Large (7.0B) model configurations used in Appendix B.1, with all experiments conducted on a uniform hardware configuration consisting of 8 A100 GPUs.

Our benchmarking results, as shown in Table 18, demonstrate that the overhead introduced by $\mathcal{R}_{\text{sp}}$ and $\mathcal{R}_{\text{cp}}$ is negligible: the combined auxiliary losses introduce only $0.5\%$ to $1.9\%$ additional latency, with the relative overhead exhibiting a decreasing trend as model scale increases (reducing to approximately $0.5\%$ for the 7B parameter model), indicating favorable scaling characteristics of our method, while the additional memory footprint is minimal ($< 0.3\%$), empirically confirming that our approach does not impose additional hardware requirements.

## LLMS USAGE

In this paper, generative LLMs were used solely for writing polishing, such as grammar and wording improvements. All LLM-edited content was manually verified to ensure compliance with ICLR policies, and authors bear full responsibility for the submission.

