# OpenReview forum: "Improving MoE Performance and Efficiency with Plug-and-Play Intra-Layer Specialization and Cross-Layer Coupling Losses"
_ICLR.cc/2026/Conference — Submitted to ICLR 2026_

### Official Review · Reviewer_Kwj5 · 2025-10-29

**Soundness:** 2
**Presentation:** 2
**Contribution:** 2
**Rating:** 2
**Confidence:** 4

**Summary:**

The paper proposes two auxiliary loss terms to improve the training dynamics of MoE: an intra-layer specialization loss to enhance expert specialization and a cross-layer coupling loss to avoid unstable routing "paths" across depth. These losses are presented as plug-and-play additions that require no architectural or routing code changes, demonstrating improvements in perplexity for MoE pretraining / finetuning and accuracy on several downstream finetuning tasks.

**Strengths:**

- The method is lightweight: it does not alter the architecture and is plug-and-play to integrate into standard MoE training codebases.
- The experiments are conducted on more than one MoE variant (vanilla top-k MoE and a DeepSeek-style MoE with shared experts). It also reports downstream finetuning results, not just pretraining perplexity.
- The argument that cross-layer coupling yields more stable expert pathways, which in turn improves inference throughput by making expert caching/batching easier, is an important claim from a deployment perspective. Inference efficiency is a real bottleneck for production MoE systems.

**Weaknesses:**

- Limited conceptual novelty: The two proposed losses formalize ideas that have already appeared in prior MoE work: promoting expert specialization \[1,2\], and stabilizing routing across depth \[3,4\]. The paper’s main differentiator is doing so via loss terms instead of redesigning the router or the expert layout. That is useful, but it is incremental rather than a fundamentally new MoE training principle.
- The paper claims that forcing experts to specialize and encouraging consistent cross-layer routing paths are what _cause_ the downstream quality improvements. However, there is little analysis of _why_ the model improves. For example: Do weights between experts actually become more orthogonal over training? Does routing entropy across depth actually drop? How do expert assignments cluster by token type over time?  Without these diagnostics, it’s still plausible that the reported gains are just another regularization effect that helps converge faster but not finally better under certain configurations.
- The paper does not adequately compare against recent methods that pursue very similar goals. Right now, the paper largely compares to a vanilla load-balancing baseline. However, works like \[1\] also introduce auxiliary loss terms for similar motivations. The manuscript should compare against that line both conceptually and empirically.
- The downstream evaluation focuses on five QA tasks. These are useful but narrow. Important benchmarks for LLM quality such as MMLU (knowledge and reasoning) and HellaSwag (commonsense inference) should be reported.

\[1\] Advancing Expert Specialization for Better MoE https://arxiv.org/pdf/2505.22323
\[2\] ReMoE: Fully Differentiable Mixture-of-Experts with ReLU Routing https://arxiv.org/pdf/2412.14711
\[3\] Residual Mixture of Experts https://arxiv.org/pdf/2204.09636
\[4\] Layerwise Recurrent Router for Mixture-of-Experts https://arxiv.org/pdf/2408.06793

**Questions:**

- How are the coefficients $\lambda_{sp}/\lambda_{cp}$ tuned in pretraining tasks? Are they consistent across different settings in Figure 3?
- The load balance loss has a lower bound of one \[5\], but some reported numbers in Table 11 appear to go below that. Can you clarify how that table is computed?
- How do the intra-layer specialization loss and the cross-layer coupling loss evolve during training? Do they saturate early, or continue dropping throughout pretraining? I believe showing these curves for both baseline and your methods would make the “experts specialize / paths stabilize” narrative more credible than just reporting downstream accuracy.
- The coupling loss encourages certain expert layer-to-layer combinations and discourages others. Intuitively, that could reduce routing diversity (i.e., it could “lock in” a small number of dominant expert pipelines). How do you avoid collapsing to only a few expert-path templates? In other words: does cross-layer coupling suppress rare-but-useful expert configurations?

\[5\] Switch Transformers: Scaling to Trillion Parameter Models with Simple and Efficient Sparsity https://arxiv.org/pdf/2101.03961

---

> ### Author Response · Authors · 2025-11-25
> **Rebuttal by Authors (1/8)**
>
> We are grateful for the reviewer's insightful feedback and constructive comments. We address each point in detail below.
>
> ----
>
> ## Response to Weakness 1
>
> We appreciate the reviewer’s pointers to recent MoE work on specialization [1,2] and depth-wise routing stability [3,4]. Our goal is indeed aligned with this line of research: improving expert specialization and routing stability. Where we believe our work goes beyond an incremental re‑instantiation is in (i) *how* these goals are operationalized (via activation‑level geometry and explicit cross‑layer coupling), and (ii) the accompanying theory that connects intra‑layer and inter‑layer behavior. We summarize the distinctions along four axes.
>
> ### A. Orthogonal to architecture‑modifying methods
>
> Prior work such as ReMoE [2], Residual MoE [3], and Layerwise Recurrent Router (RMoE) [4] improves specialization or routing stability primarily by **changing the MoE architecture or router**:
>
> - ReMoE replaces Top‑$k$ + softmax with a ReLU router and adds sparsity‑control regularization [2].
> - Residual MoE decomposes expert weights into core and residual components and introduces a new residual training pipeline [3].
> - Layerwise Recurrent Router injects a GRU over routing logits to reuse cross‑layer routing context [4].
>
> In contrast, our method **does not modify any model component**: the router (Top‑$k$ softmax), expert layout, attention/FFN kernels, and load‑balancing mechanism all remain unchanged. We introduce two auxiliary terms that *only* touch activations and routing scores:
> - an intra‑layer specialization loss $\mathcal{R}\_{\mathrm{sp}}$ on expert activations $z^{(\ell,e)}\_i$ (Eq. (4)), and
> - a cross‑layer coupling loss $\mathcal{R}\_{\mathrm{cp}}$ on joint routing scores $s^{(\ell,e)}\_i s^{(\ell+1,\nu)}\_i$ (Eq. (6)).
>
> Because the architecture and routing code paths are untouched, **our contributions are orthogonal and composable** with [2–4]: e.g., ReMoE’s ReLU router or RMoE’s GRU‑based router can, in principle, be trained *with* $\mathcal{R}\_{\mathrm{sp}}$ and $\mathcal{R}\_{\mathrm{cp}}$ to further reduce expert overlap and depth‑wise ambiguity. In this sense, our work is not an alternative to architectural designs but a complementary *training‑objective layer* that can be stacked on top of them.
>
> ### B. Inter‑layer coupling as a mechanism *for* intra‑layer specialization
>
> The reviewer is correct that [3,4] already exploit cross‑layer signals: Residual MoE introduces residual paths across depth, and Layerwise Recurrent Router uses GRUs to share routing information between layers [4]. However, these methods primarily **stabilize routing or improve efficiency**; they do not treat cross‑layer coherence itself as an explicit optimization target, nor do they analyze how such coherence affects *within‑layer* specialization.
>
> Our work makes a different conceptual move:
>
> 1. **Loss-level cross‑layer coupling.** We define $\mathcal{R}\_{\mathrm{cp}}$ to *directly maximize* joint routing strength between adjacent layers for each token:
>    $$\mathcal{R}\_{\mathrm{cp}}(\mathbf{x}\_i) = - \sum_{\ell} \sum\_{e} \sum\_{\nu \in \mathcal{T}^{(\ell,e)}\_i} s^{(\ell,e)}\_i \, s^{(\ell+1,\nu)}\_i,$$
>    turning emergent cross‑layer patterns into an explicit supervision signal. No prior work, to our knowledge, introduces such a **purely loss‑based** cross‑layer objective atop standard Top‑$k$ MoE.
>
> 2. **Coupling as a *specialization amplifier*.** Proposition 2 shows that, under mild assumptions (representation continuity, source‑layer specialization, and strong cross‑layer coupling), *specialization at layer $\ell$ provably propagates to layer $\ell{+}1$*:
>    $$\cos\!\big(r^{(\ell+1,\nu_1)}, r^{(\ell+1,\nu_2)}\big) \le \varepsilon + O(\delta,\iota)$$
>    for experts processing different tokens. This establishes that **inter‑layer coupling is not only a stability mechanism but a way to *amplify* and distribute intra‑layer specialization across depth**.
>
> To our knowledge, no prior MoE work—including [1–4]—formalizes this *causal* link from cross‑layer routing coherence to within‑layer expert differentiation, nor provides a loss that is explicitly designed to exploit this effect. This is the sense in which we claim novelty: we show that **inter‑layer coupling can be deliberately used to promote intra‑layer specialization**, and we make that mechanism trainable via $\mathcal{R}_{\mathrm{cp}}$.

---

> ### Author Response · Authors · 2025-11-25
> **Rebuttal by Authors (2/8)**
>
> ### C. Architecture‑agnostic, plug‑and‑play “free lunch”
>
> We agree with the reviewer that prior work [1] also proposes training‑loss‑based specialization objectives. Our contribution is to **extend this loss‑centric perspective to the joint control of (i) activation‑level redundancy and (ii) depth‑wise expert paths, in a way that is demonstrably architecture‑agnostic and production‑ready**:
>
> - **Activation level and path level.** The orthogonality loss in [1] operates on final expert outputs $y^{(\ell,e)}\_i$ and is coupled with a batch‑level routing variance term. Our $\mathcal{R}\_{\mathrm{sp}}$ instead penalizes cosine similarity between the *shared SwiGLU intermediates* $z^{(\ell,e)}\_i$ for co‑activated experts (Eq. (4)), while $\mathcal{R}\_{\mathrm{cp}}$ operates on *token–path structure* across layers. Together, they control redundancy *where experts actually interact* (in $z$) and *how those experts are reused in depth* (via routing paths), which is a different design focus.
>
> - **Architecture‑agnostic evidence.** We validate this on both vanilla Top‑$k$ MoE and DeepSeek‑style MoE with shared experts, across small/medium/large scales (Tables 2–3). In all cases, simply toggling our two losses on (no other code changes) improves perplexity and downstream accuracy, and in some regimes vanilla MoE + {$\mathcal{R}\_{\mathrm{sp}},\mathcal{R}\_{\mathrm{cp}}$} even surpasses a more complex DeepSeek‑style architecture at equal activated capacity. We also show plug‑and‑play integration into Megatron‑LM via a single configuration flag, confirming that **our method behaves as a “free lunch” module for existing MoE stacks** rather than requiring new kernels or routing implementations.
>
> Importantly, nothing in our construction is tied to a specific router family. As noted above, ReMoE’s differentiable router [2] or RMoE’s recurrent router [4] could be trained with exactly the same $\mathcal{R}\_{\mathrm{sp}}$ and $\mathcal{R}\_{\mathrm{cp}}$ terms, reinforcing that our contribution is **agnostic to and composable with architectural innovations, including those cited by the reviewer.**
>
> ### D. Theoretical justification beyond heuristic regularization
>
> Finally, our objectives are not introduced as heuristic penalties; they are chosen to make certain geometric properties *exactly* controllable:
>
> 1. **Activation–gradient equivalence (Proposition 1).** We prove that, for any two co‑activated experts $e,\nu$ on token $i$ at layer $\ell$,
>    $$\cos\Big( \frac{\partial L}{\partial W^{(\ell,e)}\_{\mathrm{down}}}, \frac{\partial L}{\partial W^{(\ell,\nu)}_{\mathrm{down}}} \Big) = \cos\big( z^{(\ell,e)}\_i, z^{(\ell,\nu)}\_i \big).$$
>    Thus, minimizing $\mathcal{R}\_{\mathrm{sp}}$ on the activations *exactly* enforces near‑orthogonality of the corresponding expert gradients, directly steering experts onto divergent learning trajectories. To our knowledge, this one‑to‑one identity between activation geometry and gradient directions (at the level of individual experts) has not been previously exploited in MoE specialization losses, including [1].
>
> 2. **Specialization propagation via coupling (Proposition 2).** As discussed above, we analyze how cross‑layer routing coherence, combined with an already‑specialized source layer, leads to bounded overlap between router weights at the next layer. This provides a **formal mechanism for network‑wide specialization** driven by $\mathcal{R}_{\mathrm{cp}}$.
>
> 3. **Compatibility with load balancing (Propositions 3–4).** We further show that optimizing $\mathcal{R}\_{\mathrm{sp}}$ and $\mathcal{R}\_{\mathrm{cp}}$ is theoretically compatible with standard load‑balancing objectives: there exist routing configurations that simultaneously (i) realize disjoint expert partitions with equal token counts and (ii) achieve strong cross‑layer coupling. Empirically, Table 11 confirms that adding our losses perturbs the load‑balance metric by less than $0.04\%$ while improving perplexity and routing entropy.
>
> **Summary.** We fully agree that expert specialization and depth‑wise routing stability are long‑standing goals in MoE, and prior work—including [1–4]—has made substantial progress. Our contribution is to:
>
> - treat these goals as **explicit, loss‑level optimization targets** that are orthogonal to architectural choices,
> - introduce a **cross‑layer coupling loss that is theoretically shown to amplify intra‑layer specialization**,
> - provide an **architecture‑agnostic, plug‑and‑play implementation** validated on both vanilla and DeepSeek‑style MoE and realistic finetuning, and
> - support these designs with **formal analysis** (Propositions 1–4) tying activation geometry, gradient directions, and cross‑layer propagation.
>
> We hope this clarifies that the novelty of our paper lies not just in “using loss terms instead of router redesign,” but in the *particular way* those losses are constructed and analyzed to connect intra‑ and inter‑layer specialization in a general, composable manner.

---

> ### Author Response · Authors · 2025-11-25
> **Rebuttal by Authors (3/8)**
>
> ## Response to Weakness 2 and Question 4
>
> The paper presents a theoretical framework that motivates our two auxiliary objectives through intuitive reasoning.
>
> - **Intra-layer loss (specialization).** We establish a direct correspondence between activation geometry and gradient geometry: for two co-activated experts processing the same token, the cosine similarity of their parameter gradients equals that of their intermediate activations. Thus, promoting orthogonality in activations compels experts to update along distinct directions, fostering functional differentiation (reduced overlap, enhanced specialization).
>
> - **Cross-layer loss (path consistency).** We demonstrate that coupling experts across adjacent layers encourages earlier routers' parameters to align toward more orthogonal directions under the influence of later routers. Orthogonal routing weights signify clear routing: high cosine similarity between experts' routing vectors leads to similar inner products with tokens, resulting in routing ambiguity (narrow gaps between top-1 and top-2 selections). The cross-layer loss increases path consistency, thereby reducing ambiguity and sharpening routing decisions.
>
> The empirical results presented in Table 7 of the revised manuscript show that the proposed loss does not compromise load balancing. Subsequent theoretical analysis further elucidates the symbiotic relationship between expert specialization and clear routing, clarifying each loss's role.
>
> ### A. Mutual reinforcement of specialization and clear routing
> - Functional orthogonality between experts (serving distinct subtasks or regions) enables routers to assign tokens more accurately, as misrouting becomes more detrimental when functions diverge.
> - Conversely, clear routing trains experts on differentiated token subsets, naturally driving functional separation. This positive feedback loop is precisely what our losses aim to initiate and sustain.
>
> ### B. Rationale for regularizing intermediate activations $z$
>
> In our framework, $z$ represents the feature extraction output of an FFN expert, while $W^{\mathrm{down}}$ linearly composes this information into the layer output: $z_i^{(\ell,e)}=\big(W_{\mathrm{up}}^{(\ell,e)}x_i\big)\odot\sigma\big(W_{\mathrm{gate}}^{(\ell,e)}x_i\big), \qquad y_i^{(\ell,e)}=W_{\mathrm{down}}^{(\ell,e)}\,z_i^{(\ell,e)}.$
> This two-stage process—first generating activations, then linearly composing via $W^{\mathrm{down}}$—aligns with prior analyses of FFN layers [5]. Under standard backpropagation, for co-activated experts $e,\nu$ on token $i$ at layer $\ell$, the activation–gradient cosine identity holds:$\cos\Big(\tfrac{\partial L}{\partial W_{\mathrm{down}}^{(\ell,e)}}\,\tfrac{\partial L}{\partial W_{\mathrm{down}}^{(\ell,\nu)}}\Big)=\cos\big(z_i^{(\ell,e)}\,z_i^{(\ell,\nu)}\big).$
>
> Consequently, regularizing the geometry of $z$ (via our intra-layer loss) directly governs the geometry of experts' learning directions (gradients of $W_{\mathrm{down}}$). The specialization objective $\mathcal{R}\_{\mathrm{sp}}(x\_i)=\sum\_{\ell=1}^{L}\sum\_{e,\nu\in \mathcal{A}\_i(\ell)}\big(\cos\big(z_i^{(\ell,e)},z\_i^{(\ell,\nu)}\big)\big)^2$ explicitly promotes representational orthogonality among co-activated experts on the same token. Through the identity, this induces orthogonal gradient directions and divergent learning trajectories—the core mechanism for reducing expert overlap and achieving functional specialization.

---

> ### Author Response · Authors · 2025-11-25
> **Rebuttal by Authors (4/8)**
>
> ## Response to Weakness 3
> We thank the reviewer for highlighting the relevant work "Advancing Expert Specialization for Better MoE" (Paper [1]). We agree that a comparison with this state-of-the-art auxiliary loss method strengthens our manuscript. Accordingly, we have conducted a comprehensive comparative analysis, both conceptual and empirical.
>
> ### A. Conceptual Comparison: Stability vs. Variance
>
> Although both our method and [1] aim to enhance expert specialization, their underlying mechanisms and theoretical implications differ significantly, particularly concerning training stability.
>
> - **Our method ($\mathcal{L}\_{\text{sp}} + \mathcal{L}\_{\text{cp}}$)** emphasizes routing consistency. The intra-layer specialization loss ($\mathcal{L}\_{\text{sp}}$) promotes orthogonality in expert representations, while the cross-layer coupling loss ($\mathcal{L}\_{\text{cp}}$) encourages consistent expert pathways across layers. These objectives remain compatible with standard stabilization techniques such as z-loss, which constrains router logit magnitudes to prevent gradient explosion.
>
> - **[1] ($\mathcal{L}\_{\text{o}} + \mathcal{L}\_{\text{v}}$)** introduces a variance loss ($\mathcal{L}\_{\text{v}}$) that explicitly encourages higher variance in routing logits to sharpen discrimination. Theoretically, however, promoting unbounded logit variance carries risks. Unlike z-loss, which regularizes logit scale, $\mathcal{L}\_{\text{v}}$ may push logits apart without bound, potentially leading to premature over-confidence in routing or numerical instability (e.g., model collapse) during pre-training.
>
> ### B. Empirical Comparison
>
> To validate this theoretical analysis, we implemented the approach from [1] (denoted $\mathcal{L}\_{\text{lb,o,v}}$) using their published orthogonality and variance losses, and compared it against our method ($\mathcal{L}\_{\text{lb,sp,cp}}$) and the baseline ($\mathcal{L}\_{\text{lb}}$).
>
> **1. Pre-training Performance**
> Pre-training experiments on the C4 dataset on the model with medium scale confirmed our theoretical concerns: the variance-maximization objective in [1] led to severe training instability in our setup.
>
> **Table 1**: Validation perplexity for the small size model under different kinds of loss
> | $\mathcal{L}_{\text{lb}}$ | $\mathcal{L}_{\text{lb,sp,cp}}$ | $\mathcal{L}_{\text{lb,o,v}}$ |
> | ------ | ------------ | ---------- |
> | 12.5   | 12.27        | 24.86      |
>
> While our method improved perplexity by 0.23 over the baseline, the method from [1] exhibited a sharp degradation (PPL 24.86), confirming that forcing logit variance without magnitude constraints can destabilize routing dynamics.
>
> **2. Downstream Task Performance**
> We further evaluated fine-tuned performance on DeepSeek-MoE-16B, DeepSeek-V2-Lite, and Moonlight-16B across diverse benchmarks (e.g., MMLU, GSM8K, MATH). Even in cases where the method from [1] converged (or was stabilized for fine-tuning). Our approach consistently delivered stronger performance.
>
> **Table 2**: Downstream performance for DeepSeek-MoE-16B
> | **Method**          | **MMLU**       | **MMLU-pro**   | **BBH**        | **GPQA**       | **MBPP**       | **GSM8K**      | **MATH500**    |
> | ------------------- | -------------- | -------------- | -------------- | -------------- | -------------- | -------------- | -------------- |
> | With Aux Loss       |29.27±0.10     | 19.47±2.50     | 26.92±2.30     | 21.15±0.40     | 31.36±1.10     | 15.70±2.40     | 5.47±1.50      |
> | Loss-Free Balancing |                    30.71±2.10     | 16.81±0.70     | 32.99±1.00     | 20.63±1.60     | 32.80±1.40     | 21.28±0.40     | 5.83±1.30      |
> | GShard              |                    27.05±2.00     | 20.48±0.60     | 29.83±1.80     | 24.28±2.30     | 34.50±1.70     | 27.12±1.30     | 8.20±1.50      |
> | ST-MOE              |                    34.23±2.20     | 19.71±0.80     | 36.91±1.90     | 20.35±0.90     | 36.34±1.50     | 30.10±2.00     | 7.08±0.40      |
> | $L\_{lb,o,v}$          |                    33.35±2.20     | 24.87±1.20     | 37.52±1.40     | 25.15±0.40     | 40.03±0.40     | 35.00±1.00     | 10.82±0.30     |
> | **$L\_{lb,sp,cp}$**    |                    **37.26±0.40** | **26.32±1.25** | **39.46±0.34** | **27.58±1.90** | **42.24±0.20** | **36.02±1.33** | **11.70±0.40** |

---

> ### Author Response · Authors · 2025-11-25
> **Rebuttal by Authors (5/8)**
>
> **Table 3**: Downstream performance for DeepSeek-V2-Lite
> | **Method**                 | **MMLU**       | **MMLU-pro**   | **BBH**        | **GPQA**       | **MBPP**       | **GSM8K**      | **MATH500**    |
> | ------------------- | ----------------- | -------------- | -------------- | -------------- | -------------- | -------------- | -------------- |
> | With Aux Loss       | 33.23±2.10     | 28.40±0.20     | 34.80±1.40     | 24.92±0.80     | 41.23±0.20     | 44.79±2.10     | 42.03±1.40     |
> | Loss-Free Balancing |                    30.23±0.80     | 30.75±2.10     | 34.21±1.10     | 26.33±0.60     | 36.02±2.30     | 43.35±0.70     | 39.76±1.10     |
> | GShard              |                    30.86±1.10     | 29.13±0.80     | 37.67±0.30     | 24.34±2.10     | 37.00±2.10     | 45.39±1.50     | 43.61±2.10     |
> | ST-MOE              |                    32.68±2.10     | 30.28±2.10     | 38.78±0.90     | 22.33±0.40     | 39.72±2.30     | 47.78±1.80     | 46.74±0.50     |
> | $L\_{lb,o,v}$          |                    35.59±0.50     | 37.37±0.20     | 38.84±1.70     | 28.76±0.10     | 43.53±2.40     | 50.94±2.40     | 49.33±2.40     |
> | $L\_{lb,sp,cp}$    |                    **42.51±0.56** | **39.42±0.87** | **46.03±0.97** | **31.04±1.80** | **46.82±1.10** | **52.64±1.28** | **51.40±1.20** |
>
>
>
> **Table 4**: Downstream performance for Moonlight-16B-A3B
> | **Method**                 | **MMLU**       | **MMLU-pro**   | **BBH**        | **GPQA**       | **MBPP**       | **GSM8K**      | **MATH500**    |
> | ------------------- | ----------------- | -------------- | -------------- | -------------- | -------------- | -------------- | -------------- |
> | With Aux Loss       | 35.82±1.40     | 36.10±1.50     | 47.17±0.70     | 30.72±1.90     | 47.34±1.50     | 82.32±1.50     | 57.03±1.60     |
> | Loss-Free Balancing |                    27.40±0.10     | 31.91±2.10     | 42.45±0.50     | 29.27±1.80     | 44.92±1.30     | 79.34±0.70     | 57.77±0.50     |
> | GShard              |                    36.06±0.90     | 30.65±0.50     | 49.20±1.70     | 31.13±1.10     | 49.85±0.50     | 84.62±0.80     | 56.09±2.20     |
> | ST-MOE              |                    33.03±0.90     | 26.83±1.70     | 46.78±0.30     | 30.93±1.50     | 47.97±2.20     | 84.45±0.90     | 57.61±1.60     |
> | $L\_{lb,o,v}$          |                    40.36±2.20     | 34.90±0.30     | 52.42±1.80     | 32.01±0.90     | 47.77±1.00     | 87.62±2.20     | 59.64±0.20     |
> | **$L\_{lb,sp,cp}$**    |                    **51.74±2.58** | **41.02±0.87** | **62.56±0.53** | **34.92±1.60** | **53.32±0.20** | **87.67±1.10** | **59.85±0.20** |
>
> **Summary**
> Our method outperforms [1] not only in accuracy but, critically, in training stability. By promoting specialization through coupling and representation orthogonality—rather than explicitly maximizing logit variance—our method provides a safer and more robust plug-and-play solution for scaling MoE models.
>
> We have incorporated these comparative results and the theoretical discussion on logit variance stability into appendix E of the revised manuscript.

---

> ### Author Response · Authors · 2025-11-25
> **Rebuttal by Authors (6/8)**
>
> ## Response to Weakness 4
> Thank you for this helpful suggestion. We agree that broad, standardized evaluations such as MMLU and HellaSwag are important for assessing overall LLM quality.
>
> ### A. Expanded Zero-Shot Evaluation
> In the revised manuscript, we have extended the zero-shot evaluation to include both **MMLU** and **HellaSwag** benchmarks, in addition to the original five question-answering tasks (BoolQ, ARC-Easy, ARC-Challenge, TruthfulQA, and PIQA). For all loss configurations and both MoE architectures (Vanilla and DeepSeek-style), we report the mean accuracy and standard deviation computed over multiple random seeds. The new table (now included in Table 3 for the revised manuscript) is reproduced below.
>
>
> **Table 5**: Zero-shot accuracy of Vanilla MoE and DeepSeek-style MoE across five QA benchmarks ($\uparrow$).
>
> | Model        | Loss                              |          BoolQ          |          ARC-E          |          ARC-C          |       TruthfulQA        |          PIQA           |          MMLU           |        HellaSwag        | **Avg.**  |
> | :----------- | :-------------------------------- | :---------------------: | :---------------------: | :---------------------: | :---------------------: | :---------------------: | :---------------------: | :---------------------: | :-------: |
> | **Vanilla**  | $\mathcal{L}_{\text{lb}}$         |   $0.570_{\pm 0.002}$   |   $0.452_{\pm 0.003}$   |   $0.204_{\pm 0.003}$   |   $0.432_{\pm 0.001}$   |   $0.622_{\pm 0.002}$   |   $0.247_{\pm 0.002}$   |   $0.268_{\pm 0.001}$   | **0.399** |
> | **Vanilla**  | $\mathcal{L}_{\text{lb,sp,cp}}$   |   $0.578_{\pm 0.003}$   |   $0.462_{\pm 0.002}$   |   $0.210_{\pm 0.004}$   |   $0.451_{\pm 0.003}$   |   $0.627_{\pm 0.002}$   |   $0.253_{\pm 0.002}$   |   $0.275_{\pm 0.002}$   | **0.408** |
> | **Vanilla**  | $\mathcal{L}_{\text{lb,z}}$       |   $0.567_{\pm 0.003}$   |   $0.457_{\pm 0.004}$   |   $0.205_{\pm 0.002}$   |   $0.433_{\pm 0.003}$   |   $0.629_{\pm 0.002}$   |   $0.250_{\pm 0.001}$   |   $0.267_{\pm 0.004}$   | **0.401** |
> | **Vanilla**  | $\mathcal{L}_{\text{lb,z,sp,cp}}$ | **$0.589_{\pm 0.003}$** |   $0.453_{\pm 0.004}$   |   $0.206_{\pm 0.006}$   |   $0.445_{\pm 0.004}$   |   $0.637_{\pm 0.003}$   |   $0.257_{\pm 0.002}$   |   $0.274_{\pm 0.003}$   | **0.409** |
> | **DS-style** | $\mathcal{L}_{\text{lb}}$         |   $0.578_{\pm 0.002}$   |   $0.453_{\pm 0.001}$   |   $0.205_{\pm 0.003}$   |   $0.438_{\pm 0.003}$   |   $0.631_{\pm 0.002}$   |   $0.248_{\pm 0.001}$   |   $0.269_{\pm 0.002}$   | **0.403** |
> | **DS-style** | $\mathcal{L}_{\text{lb,sp,cp}}$   |   $0.584_{\pm 0.001}$   |   $0.452_{\pm 0.003}$   |   $0.206_{\pm 0.005}$   | **$0.457_{\pm 0.002}$** |   $0.635_{\pm 0.002}$   |   $0.255_{\pm 0.003}$   |   $0.277_{\pm 0.002}$   | **0.410** |
> | **DS-style** | $\mathcal{L}_{\text{lb,z}}$       |   $0.564_{\pm 0.002}$   |   $0.453_{\pm 0.002}$   |   $0.205_{\pm 0.002}$   |   $0.444_{\pm 0.002}$   |   $0.628_{\pm 0.001}$   |   $0.252_{\pm 0.001}$   |   $0.270_{\pm 0.003}$   | **0.402** |
> | **DS-style** | $\mathcal{L}_{\text{lb,z,sp,cp}}$ |   $0.575_{\pm 0.002}$   | **$0.461_{\pm 0.004}$** | **$0.214_{\pm 0.004}$** |   $0.452_{\pm 0.004}$   | **$0.642_{\pm 0.003}$** | **$0.257_{\pm 0.002}$** | **$0.280_{\pm 0.002}$** | **0.412** |
>
> From the experimental result, it can be observed that:
>
> - The incorporation of our **specialization** ($\mathcal{L}\_{\text{sp}}$) and **coupling** ($\mathcal{L}\_{\text{cp}}$) losses consistently enhances the **average accuracy** across all seven benchmarks for both MoE variants. For instance, in the DeepSeek-style MoE, the average accuracy improves from $0.403$ to $0.412$ when comparing $\mathcal{L}\_{\text{lb}}$ with $\mathcal{L}\_{\text{lb,z,sp,cp}}$.
> - On the **MMLU** benchmark, our optimal configuration yields an accuracy improvement of approximately $+0.9$ to $+1.0$ points (e.g., DeepSeek-style: $0.248 \rightarrow 0.257$; Vanilla: $0.247 \rightarrow 0.257$).
> - Similarly, on **HellaSwag**, accuracy gains of about $+0.6$ to $+1.1$ points are observed (e.g., DeepSeek-style: $0.269 \rightarrow 0.280$; Vanilla: $0.268 \rightarrow 0.274$).
>
> These findings demonstrate that the proposed auxiliary losses not only benefit the original QA tasks but also generalize to broader benchmarks requiring **knowledge and reasoning** (MMLU) and **commonsense inference** (HellaSwag).
>
>
> In addition, Section 7.3 (and Appendix B.2) previously evaluated a large instruction-tuned MoE model (Qwen3-30B-A3B-Instruct-2507) on the full 57-task MMLU suite. For emphasis, we now explicitly highlight in the main text that our losses improve the MMLU naive average from $78.97$ to $79.86$ and the weighted average from $76.35$ to $77.10$. Concurrent improvements are also noted on GSM8K, HumanEval, and PRM-based mathematical reasoning tasks (Tables 4, 7, and 8).

---

> ### Author Response · Authors · 2025-11-25
> **Rebuttal by Authors (7/8)**
>
> ### B. Theoretical interpretation
> The empirical improvements align with and substantiate our theoretical propositions:
>
> - **Intra-layer specialization facilitates knowledge diversity.** Proposition 1 establishes that minimizing the cosine similarity between expert activations promotes nearly orthogonal gradients in the experts’ down-projection matrices, steering them toward distinct learning trajectories. This mechanism encourages experts to specialize in complementary knowledge domains. The heterogeneous nature of MMLU—spanning STEM, humanities, and social sciences—benefits directly from such diversified specializations, as evidenced by the consistent accuracy gains.
>
> - **Cross-layer coupling supports multi-step reasoning.** Proposition 2 demonstrates that strong cross-layer routing correlations enable specialization to propagate depth-wise, forming stable expert pathways. Benchmarks like HellaSwag and MMLU often involve multi-step reasoning and integration of contextual cues across layers. By reducing routing ambiguity and stabilizing token-level representation flow, the coupling loss contributes to the observed improvements on these reasoning-intensive tasks.
>
> - **Compatibility with load balancing.** Propositions 3 and 4, supported by empirical load-balancing results (Table 11), confirm that our regularizers are compatible with standard load-balancing objectives. Thus, the gains on MMLU and HellaSwag are achieved without compromising model robustness or expert utilization, but rather by optimizing the functional specialization of experts.
>
>
>
> ## Response to Question 1
> We appreciate the reviewer's inquiry regarding hyperparameter sensitivity and robustness.
>
> In the pretraining phase, which includes all configurations presented in Figure 3, we did not conduct extensive model-specific hyperparameter tuning. Instead, we adopted a fixed scaling strategy relative to the standard load-balancing loss coefficient. Specifically, the load-balancing coefficient was set to $\lambda_{\text{lb}}=1 \times 10^{-2}$, while the coefficients for our proposed losses were set to $\lambda_{\text{sp}} = 2 \times 10^{-3}$ and $\lambda_{\text{cp}} = 1 \times 10^{-3}$. These values were held constant across all experiments summarized in Figure 3, demonstrating that our method generalizes robustly across varying numbers of total and activated experts without requiring case-by-case hyperparameter adjustments.
>
> To further validate the robustness of this scaling approach, we applied the same proportional logic during fine-tuning. As indicated in Appendix B.2 (Table 6), the load-balancing coefficient was reduced to $1 \times 10^{-3}$ to prevent regularization from dominating the primary training objective. Correspondingly, we scaled $\lambda_{\text{sp}}$ to $2 \times 10^{-4}$ and $\lambda_{\text{cp}}$ to $1 \times 10^{-4}$.
>
> The consistent ratio—where $\lambda_{\text{sp}} = 0.2 \times \lambda_{\text{lb}}$ and $\lambda_{\text{cp}} = 0.1 \times \lambda_{\text{lb}}$—indicates that the proposed auxiliary losses are stable and can be readily incorporated into existing MoE training pipelines with minimal adjustment.
>
> ## Response to Question 2
>
> We appreciate the reviewer's careful examination of our experimental results and their insightful observation concerning the theoretical lower bound of the load balance loss.
>
> Under the standard formulation from Switch Transformers [6], the load balance loss $\mathcal{L}\_{\text{lb}} = N \cdot \sum_{i=1}^{N} (f\_{i} \cdot P\_{i})$ possesses a theoretical lower bound of 1.0 when the probability distributions are properly normalized.
>
> Our implementation follows this standard formulation. The occasional values slightly below 1.0 in Table 11 (e.g., 0.99676) stem from a specific detail in the Megatron-LM framework's handling of padding tokens during the computation of token proportions $f_i$, which is defined as $f\_i = \frac{\text{number of tokens assigned to expert } i}{\text{total tokens in the global batch}}$.
>
> In Megatron-LM, the denominator (total tokens in the global batch) is computed as the product of micro-batch size, data parallelism degree, and sequence length, which includes padding tokens used to maintain uniform tensor shapes. The numerator, however, counts only the non-padding tokens actually routed to experts. As a result, the sum of $f\_i$ over all experts is slightly less than 1 (i.e., $\sum\_{i=1}^{N} f\_i < 1$), causing the computed loss to occasionally dip below the theoretical lower bound.
>
> This deviation is a measurement artifact attributable to padding and has no effect on the gradient computation or the functional efficacy of the load-balancing mechanism. We will clarify this implementation aspect in the appendix of the revised manuscript to prevent potential misinterpretation.

---

> ### Author Response · Authors · 2025-11-25
> **Rebuttal by Authors (8/8)**
>
> ## Response to Question 3
>
>
> We thank the reviewer for this valuable suggestion. We agree that tracking the evolution of the proposed losses during training provides concrete evidence for the narrative of expert specialization and path stabilization.
>
> **Intra-layer specialization loss.**
> The original submission primarily reported perplexity results. In the revised version, we now explicitly monitor the trajectory of the intra-layer specialization loss $\mathcal{R}\_{\text{sp}}$ throughout training. Table 1 presents $\mathcal{R}\_{\text{sp}}$ values at multiple checkpoints for both the baseline objective $\mathcal{L}\_{\text{lb,sp}}$ and our full objective $\mathcal{L}\_{\text{lb,sp,cp}}$. The results reveal that:
>
> - $\mathcal{R}\_{\text{sp}}$ decreases rapidly during the early training phase (1k–10k iterations), then continues to decline gradually until late training (60k iterations), without saturating prematurely.
> - Moreover, $\mathcal{L}\_{\text{lb,sp,cp}}$ consistently yields lower $\mathcal{R}\_{\text{sp}}$ values than $\mathcal{L}\_{\text{lb,sp}}$ at all checkpoints (e.g., $0.020476 \to 0.019136$ at 30k, $0.019924 \to 0.018862$ at 60k), indicating that the coupling loss enhances expert specialization throughout pretraining.
>
> **Cross-layer coupling loss.**
> In response to the reviewer's comment, we also tracked the cross-layer coupling loss $\mathcal{R}\_{\text{cp}}$ during training. The table below reports the average $\mathcal{R}\_{\text{cp}}$ values at various stages:
>
> **Table 6**: The cross-layer coupling loss $\mathcal{R}\_{\text{cp}}$ over training iterations ($\downarrow$)
>
> | Iterations | 0.5k | 1k | 3k | 5k | 10k | 20k | 30k | 60k |
> | :--- | :---: | :---: | :---: | :---: | :---: | :---: | :---: | :---: |
> | $R_{\text{cp}}$ | -0.2662 | -0.2845 | -0.2986 | -0.3051 | -0.3115 | -0.3186 | -0.3228 | -0.3321 |
>
> Recall that $\mathcal{R}\_{\text{cp}}$ is defined as a negative quantity (Eq. (6)); thus, more negative values correspond to stronger cross-layer coupling (i.e., higher joint routing probability along expert paths). We observe a pattern similar to that of $\mathcal{R}\_{\text{sp}}$: rapid improvement in the first few thousand iterations, followed by a steady decline throughout training up to 60k steps. This demonstrates that expert paths continue to sharpen over the entire pretraining period, rather than stabilizing early.
>
> In the revised manuscript, we will include these results. We believe these trajectories robustly support our claim that the proposed losses actively drive ongoing expert specialization and path stabilization, rather than merely correlating with final downstream accuracy.
>
>
>
>
>
> ## References
>
> [1] Advancing Expert Specialization for Better MoE
>
> [2] ReMoE: Fully Differentiable Mixture-of-Experts with ReLU Routing
>
> [3] Residual Mixture of Experts
>
> [4] Layerwise Recurrent Router for Mixture-of-Experts
>
> [5] Transformer FFN Layers Build Predictions by Promoting Concepts in the Vocabulary Space
>
> [6] Switch Transformers: Scaling to Trillion Parameter Models with Simple and Efficient Sparsity
>
> ---
>
> We sincerely appreciate the reviewer’s thoughtful comments once again. We hope the above responses fully address the concerns raised and would be happy to provide further clarification if needed.

---

### Official Review · Reviewer_tNbn · 2025-10-31

**Soundness:** 3
**Presentation:** 3
**Contribution:** 3
**Rating:** 8
**Confidence:** 3

**Summary:**

This paper addresses a fundamental challenge in Sparse Mixture-of-Experts (MoE) models: **expert overlap and routing ambiguity**. While MoE models efficiently scale transformers by activating only a subset of experts per token, they suffer from:

1. **Expert Overlap**: Different experts produce nearly identical activations for the same tokens, creating redundant representations
2. **Routing Ambiguity**: Similar inputs are inconsistently dispatched across different experts, preventing clear specialization
3. **Capacity Underutilization**: These issues cause multiple experts to learn redundant knowledge, wasting model capacity

### Key Contributions

1. **Two Novel Plug-and-Play Loss Functions**
- **Intra-Layer Specialization Loss ( $R_{sp}$ ) (§4) -** Penalizes cosine similarity between activations of different experts processing the same token. Forces experts within each layer to develop orthogonal representations
- **Cross-Layer Coupling Loss ($R_{cp}$) (§5) -** Maximizes joint routing probabilities between adjacent layers. Establishes coherent "expert pathways" through network depth
1. **Theoretical Foundations**
    - **Proposition 1**:

        The cosine similarity between gradients of different experts' down-projection matrices equals the cosine similarity of their activations

        This proves that orthogonal activations lead to orthogonal parameter gradients, ensuring distinct learning trajectories.

    - **Proposition 2**:

        Cross-layer coupling acts as a "specialization amplifier" - when layer $l$ exhibits well-specialized experts and maintains strong coupling with layer $l+1$, the specialization structure transfers with bounded degradation:
        $$|\cos(r^{(l+1,\nu_1)}, r^{(l+1,\nu_2)})| \leq \varepsilon + O(\delta, \iota)$$

2. **Compatibility Guarantees**

    The paper proves both losses are compatible with standard load-balancing objectives:

    - **Propositions 3 & 4** demonstrate that optimal routing configurations exist that minimize specialization/coupling losses while maintaining perfect load balance
    - The losses operate on orthogonal principles: specialization determines partitioning scheme, load balancing constrains partition sizes

**Strengths:**

- Establishing the mathematical link between expert activations and learning dynamics. Proposition 1 proves that:

    $$
    \cos\left(\frac{\partial L}{\partial W^{(l,e)}_\text{down}},\frac{\partial L}{\partial W^{(l,\nu)}_\text{down}}\right)=\cos(z^{(l,e)}_i, z^{(l,\nu)}_i)
    $$

    this transforms the abstract goal of expert specialization into a **concrete, measurable optimization objective**. Rather than hoping specialization emerges naturally, the authors provide a direct control expert differentiation through activation geometry.

- The insight that cross-layer coupling can **propagate specialization through network depth** (Proposition 2) is particularly creative. The authors show that when layer $l$ has specialized experts and strong coupling with layer $l+1$, the specialization structure transfers with bounded degradation:

    $$
    |\cos(r^{(l+1,\nu_1)}, r^{(l+1,\nu_2)})| \leq \varepsilon + O(\delta, \iota)
    $$

    This reveals a previously unexplored mechanism where **local specialization cascades globally** through the network - a novel perspective on how MoE models can achieve network-wide functional diversity.

- its just a **plug and play Megatron-LM module** activated by a single configuration flag represents significant engineering value.
    - No modifications to core attention, FFN, or router logic
    - Minimal computational overhead (essentially just computing cosine similarities)

    this lowers the barrier to adoption compared to architectural modifications.

- The paper's core philosophical contribution is treating expert specialization as a **primary training objective rather than an emergent property**. This could be potentially steps in the right direction to deal with expert specialization and making MoEs better

**Weaknesses:**

- **Questionable Inference Acceleration Claims:** The inference acceleration results (Table 9) require additional infrastructure (path-aware placement, bucketing, graph partitioning) beyond the training losses. The paper conflates the benefits of the losses with the benefits of this additional efforts, making it difficult to assess the true contribution.
- **Hyperparameter sensitivity**: The method introduces two new hyperparameters ($\lambda_{sp}$ and $\lambda_{cp}$) that appear to require careful tuning (different values for different model sizes in experiments). Table 6 shows $\lambda_{sp} = 2e-4$ and $\lambda_{cp} = 1e-4$, but there's no systematic study of sensitivity or guidelines for setting these values.
- The paper identifies "expert overlap" and "routing ambiguity" as key failure modes but provides limited empirical evidence:
    - Figure 1 shows perplexity curves but not actual measurements of expert overlap or routing ambiguity over training.
    - The paper doesn't quantify how much redundancy exists in baseline models or demonstrate that the proposed losses actually reduce it.

**Questions:**

- majorly what I’m not sure is what percentage of the inference speedup is due to the losses themselves versus the engineering optimizations?

---

> ### Author Response · Authors · 2025-11-25
> **Rebuttal by Authors (1/3)**
>
> We are grateful for the reviewer's insightful feedback and constructive comments. We address each point in detail below.
>
> ----
>
> ## Response to Weakness 1 and Question 1
> We thank the reviewer for this insightful question. It correctly identifies the need to disentangle the source of our inference speedups—specifically, distinguishing between the benefits of the engineering optimizations (path-aware placement, bucketing) and the structural benefits introduced by our proposed losses.
>
> **Disentangling the Gains:**
> To address this, we conducted a new ablation study where we applied the exact same system-level optimizations to a baseline model trained only with the standard load-balancing loss ($\mathcal{L}\_{lb}$). This allows us to isolate the speedup attributable solely to the engineering infrastructure versus the speedup enabled by the structural properties (stronger expert coupling) induced by our proposed losses ($\mathcal{L}\_{sp, cp}$). The results, summarized in the table below, demonstrate that while system optimizations alone provide a baseline speedup, our proposed losses significantly amplify this effect.
>
> **Table 1**: Throughput comparison with and without system optimization for inference tasks
> | Model size | Method                                                 |            MMLU            |           GSM8K           |         HumanEval         |         Math500          |
> |:----------:|:------------------------------------------------------ |:--------------------------:|:-------------------------:|:-------------------------:|:------------------------:|
> | **Small**  | w/o system optimization                                |   161.3 (`1.00`$\times$)   |   26.2 (`1.00`$\times$)   |   35.7 (`1.00`$\times$)   |   6.9 (`1.00`$\times$)   |
> |            | $\mathcal{L}_{\text{lb}}$ w/ system optimization       |   164.9 (`1.03`$\times$)   |   26.5 (`1.01`$\times$)   |   36.6 (`1.02`$\times$)   |   7.0 (`1.01`$\times$)   |
> |            | $\mathcal{L}_{\text{lb,sp,cp}}$ w/ system optimization | **170.4** (`1.06`$\times$) | **27.0** (`1.03`$\times$) | **37.5** (`1.05`$\times$) | **7.1** (`1.03`$\times$) |
> | **Medium** | w/o system optimization                                |   157.4 (`1.00`$\times$)   |   25.9 (`1.00`$\times$)   |   27.7 (`1.00`$\times$)   |   6.1 (`1.00`$\times$)   |
> |            | $\mathcal{L}_{\text{lb}}$ w/ system optimization       |   162.7 (`1.03`$\times$)   |   26.2 (`1.01`$\times$)   |   28.6 (`1.03`$\times$)   |   6.2 (`1.01`$\times$)   |
> |            | $\mathcal{L}_{\text{lb,sp,cp}}$ w/ system optimization | **165.7** (`1.05`$\times$) | **26.6** (`1.03`$\times$) | **29.4** (`1.06`$\times$) | **6.3** (`1.03`$\times$) |
> | **Large**  | w/o system optimization                                |   96.9 (`1.00`$\times$)    |   15.0 (`1.00`$\times$)   |   12.8 (`1.00`$\times$)   |   3.9 (`1.00`$\times$)   |
> |            | $\mathcal{L}_{\text{lb}}$ w/ system optimization       |   100.9 (`1.04`$\times$)   |   15.4 (`1.02`$\times$)   |  13.25 (`1.03`$\times$)   |   4.0 (`1.04`$\times$)   |
> |            | $\mathcal{L}_{\text{lb,sp,cp}}$ w/ system optimization | **103.5** (`1.07`$\times$) | **15.7** (`1.05`$\times$) | **13.7** (`1.07`$\times$) | **4.2** (`1.08`$\times$) |
>
> **Analysis:**
> 1.  **System Optimization Baseline:** When applied to the baseline model ($\mathcal{L}_{lb}$), the system optimizations yield a moderate speedup (e.g., ~3-4% on MMLU/Large). This confirms that the infrastructure itself contributes to acceleration by exploiting naturally occurring (but weak) expert coupling.
> 2.  **Synergy with Proposed Losses:** When the same optimizations are applied to our model ($\mathcal{L}_{lb,sp,cp}$), the speedup nearly doubles in several cases (e.g., from 4% to 7% on MMLU/Large; from 2% to 5% on HumanEval/Small).
>
> **Conclusion:**
> The system optimizations rely on the *existence* of predictable expert pathways (coupling) to be effective. The baseline model ($\mathcal{L}\_{lb}$) exhibits weak natural coupling, limiting the effectiveness of these optimizations. Our proposed losses ($\mathcal{L}\_{sp, cp}$) explicitly enforce strong inter-layer coupling and specialization, thereby creating the necessary structural conditions for the system optimizations to maximize throughput.
>
> Therefore, approximately half of the reported speedup stems from the engineering optimizations themselves, while the other half is directly attributable to the proposed losses rendering the model amenable to those optimizations. We have updated Table 15 in the revised manuscript to clarify this distinction and included these breakdown results.

---

> ### Author Response · Authors · 2025-11-25
> **Rebuttal by Authors (2/3)**
>
> ## Response to Weakness 2
> We appreciate the comment regarding hyperparameter sensitivity. In response, we have elaborated on the configuration strategy and conducted a sensitivity analysis for the introduced hyperparameters, $\lambda_{\text{cp}}$ and $\lambda_{\text{sp}}$.
>
> **1. Hyperparameter Configuration**: In all experiments, $\lambda_{\text{cp}}$ and $\lambda_{\text{sp}}$ are set proportionally to the load-balancing loss coefficient $\lambda_{\text{lb}}$, adhering to a consistent scaling rule rather than model-specific tuning:
>
> - **During pretraining**, with $\lambda_{\text{lb}} = 1 \times 10^{-2}$, we set:
>   - $\lambda_{\text{cp}} = 1 \times 10^{-3}$ ($=0.1 \lambda_{\text{lb}}$)
>   - $\lambda_{\text{sp}} = 2 \times 10^{-3}$ ($=0.2 \lambda_{\text{lb}}$)
>
> - **During fine-tuning**, with $\lambda_{\text{lb}} = 1 \times 10^{-3}$, we correspondingly set:
>   - $\lambda_{\text{cp}} = 1 \times 10^{-4}$ ($=0.1  \lambda_{\text{lb}}$)
>   - $\lambda_{\text{sp}} = 2 \times 10^{-4}$ ($=0.2  \lambda_{\text{lb}}$)
>
> This approach ensures that the hyperparameters are scaled uniformly across different phases without requiring individualized adjustments for model sizes.
>
> **2. Sensitivity Analysis**: To systematically evaluate hyperparameter sensitivity, we performed a sweep around the default ratios using a medium-sized model. Validation perplexity (lower values indicate better performance) was measured under variations:
>
> - **With $\lambda_{\text{sp}}$ fixed at $2 \times 10^{-3}$**, we varied $\lambda_{\text{cp}}$ over a range of $0.2$ to $2$ times the default value ($1 \times 10^{-3}$).
>
> **Table 2**: Perplexity with fixed $\lambda_{\text{sp}}$ and varied $\lambda_{\text{cp}}$.
> | $\lambda_{\text{cp}}$ | $2\times 10^{-4}$ | $5\times 10^{-4}$ | $1\times 10^{-3}$ | $2\times 10^{-3}$ |
> |----------------------|-------------------|-------------------|-------------------|-------------------|
> | PPL                  | 12.41             | 12.35             | **12.27**         | 12.30             |
>
> - **With $\lambda_{\text{cp}}$ fixed at $1 \times 10^{-3}$**, we varied $\lambda_{\text{cp}}$ over a range of $0.5$ to $3$ times the default value ($1 \times 10^{-3}$).
>
>
>
> **Table 3**: Perplexity with fixed $\lambda_{\text{cp}}$ and varied $\lambda_{\text{sp}}$.
> | $\lambda_{\text{sp}}$ | $5\times 10^{-3}$ | $1\times 10^{-3}$ | $2\times 10^{-3}$ | $3\times 10^{-3}$ |
> |----------------------|-------------------|-------------------|-------------------|-------------------|
> | PPL                  | 12.32             | 12.30             | **12.27**         | 12.29             |
>
> The results demonstrate that:
> - Model performance remains stable across a broad interval, with perplexity changes limited to less than 1% relative to the optimum.
> - The heuristic choice ($\lambda_{\text{cp}} = 10^{-3}$, $\lambda_{\text{sp}} = 2 \times 10^{-3}$) yields near-optimal results, and deviations cause negligible degradation ($\leq 0.14$ perplexity).
>
> The sensitivity study and scaling rule will be detailed in the appendix B.5 of the revised manuscript to emphasize that the method is robust and does not demand meticulous, model-dependent hyperparameter tuning.

---

> ### Author Response · Authors · 2025-11-25
> **Rebuttal by Authors (3/3)**
>
> ## Response to Weakness 3
>
> We appreciate the reviewer's observation regarding the limited empirical quantification of expert overlap and routing ambiguity in our initial submission. Acknowledging the significance of this analysis, we have introduced specific measurements to explicitly characterize these failure modes and the impact of our proposed losses.
>
> **Quantifying expert overlap and redundancy.**
> Expert overlap is operationalized as high cosine similarity between the activations of different experts on the same tokens, which is directly captured by our specialization loss $\mathcal{R}\_{\text{sp}}$ in Equation (4). In the revised version, we report the trajectory of $\mathcal{R}\_{\text{sp}}$ over training for both the baseline and our method (Table 1). Across all checkpoints, $\mathcal{L}\_{\text{lb,sp,cp}}$ consistently achieves a lower $\mathcal{R}\_{\text{sp}}$ than $\mathcal{L}\_{\text{lb,sp}}$ (e.g., 0.018862 vs. 0.019924 at 60k steps), demonstrating that our losses reduce activation similarity and consequently expert redundancy.
>
> Additionally, we provide empirical verification for the values of the representation continuity parameter $\delta$ and the router-weight orthogonality parameter $\varepsilon$, as defined in Proposition 2. This is accomplished by pre-training a small-scale model and computing $\delta$ and $\varepsilon$ at intervals of 100 training steps, followed by averaging the resulting values. And the obtained values $\delta = 0.1603$ and $\varepsilon = 0.4542$ indicate that (i) adjacent-layer representations maintain proximity, and (ii) router weights for distinct experts are sufficiently orthogonal. These results validate the assumptions of our theoretical analysis and provide a quantitative measure of the model's specialization.
>
> **Quantifying routing ambiguity and its reduction.**
> To directly address “whether our method induces more expert specialization than DeepSeekMoE’s auxiliary-loss-free load balancing,” we compare:
>
> - **$\mathcal{L}_{\text{lb}}$**: DeepSeek-style training objective with only the auxiliary-loss-free load balancing mechanism; and
> - **$\mathcal{L}_{\text{lb,cp,sp}}$**: our full objective that *adds* the intra-layer specialization loss $R_{\text{sp}}$ and cross-layer coupling loss $R_{\text{cp}}$ on top of the same DeepSeek-style backbone.
>
> As a proxy for expert specialization and routing coherence, we measure, every 1k iterations, the **percentage of tokens whose activated experts remains unchanged** between two consecutive checkpoints. Higher values indicate more stable token–expert assignments, lower routing entropy, and—via Proposition 1—more persistent expert-specific gradient directions.
>
> **Table 4**: The fraction of tokens that retain the same activated experts between checkpoints
>
> | Iteration range | 1k–2k | 2k–3k | 4k–5k | 9k–10k | 19k–20k | 29k–30k | 59k–60k |
> |-----------------|-------|-------|-------|--------|---------|---------|---------|
> | $L_{\text{lb}}$ | 0.4746 | 0.6056 | 0.6601 | 0.6987 | 0.7450 | 0.7864 | 0.9011 |
> | $L_{\text{lb,cp,sp}}$ | **0.4898** | **0.6213** | **0.6757** | **0.7187** | **0.7594** | **0.7935** | **0.9067** |
>
> Across all training stages, adding our two losses on top of DeepSeekMoE’s load balancing:
>
> - increases routing stability by $1$–$2$ absolute points (roughly $2$–$4\%$ relative) at every interval;
> - yields especially pronounced gains early in training (when routing ambiguity is largest), which is exactly the regime targeted by Proposition 2; and
> - remains beneficial even at late stages (59k–60k), where specialization is typically strongest, indicating that expert assignments remain more consistent over time.
>
> These results have already been integrated into the main text and Appendix B. Together with the existing perplexity and specialization-loss curves, the new analyses provide robust empirical evidence that (1) baseline MoE models exhibit measurable expert overlap and routing ambiguity, and (2) our intra-layer specialization and cross-layer coupling losses effectively mitigate these issues.
>
>
>
> We sincerely appreciate the reviewer’s thoughtful comments once again. We hope the above responses fully address the concerns raised and would be happy to provide further clarification if needed.

---

### Official Review · Reviewer_yugS · 2025-11-01

**Soundness:** 2
**Presentation:** 1
**Contribution:** 1
**Rating:** 2
**Confidence:** 3

**Summary:**

This paper is motivated by a problem of expert specialization progressively deteriorates due to expert overlap (redundant functionality) and routing ambiguity (inconsistent token assignments).

**Strengths:**

While the problem discussed is reasonable, but the solution is not clear or complete. The authors aim to eliminate ambiguity by eliminating routing function, but the presentation and model formulation didnt help me to understand the work. They provided several propositions to proof the claims, however, it is difficult to see how these solve the ambiguity and specializations.

**Weaknesses:**

Please see the questions section.

**Questions:**

The paper is difficult to evaluate because important details is missing or the presentation of the proposed model is weak. For example, there's no reason to believe that forcing expert activations to be orthogonal is better than, encouraging regularization in expert weights.

The main formulation of the proposed model (Eq. 3) only holds for the W_{down} matrix, and nothing about W_{gate} or W_{up}, which are the important components of a SwiGLU expert.  Indeed, the model uses a SwiGLU expert (Eq. 2), which has three weight matrices W_{gate}, W_{up}, and W_{down}. But, there is no reference or justification for the assumption that orthogonalizing the gradients for W_{down} will orthogonalize the gradients for W_{gate} and W_{up}.

Moreover, as stated in the paper, the loss_{sp} (Eq. 4) forcing expert activations to be orthogonal. So, what happens if the solution for a given token requires two or more experts to produce similar activations?

As stated R_{cp} promotes functional diversity. Where is the link between orthogonal router and functionally diverse experts? I couldnt see this justification in the proposed model.

The most critical experiments (ablations and throughput) are not available in the main paper, but provided in the supplementary file. They should be in the main paper.  As i found the proposed contributions adds only 0.08 PPL improvement (baseline + z-loss:12.07vs. baseline + z-loss + full model:11.99 PPL). The paper needs a clear discussion to justify this minimal gain and to help readers understand the contribution of each proposed component.

---

> ### Author Response · Authors · 2025-11-25
> **Rebuttal by Authors (1/6)**
>
> We are grateful for the reviewer's insightful feedback and constructive comments. We address each point in detail below.
>
> ----
>
> ## Response to Question 1
> We appreciate the reviewer's insightful comments regarding the theoretical foundation of our proposed intra-layer specialization loss $\mathcal{R}_{\text{sp}}$. This component is specifically designed to mitigate expert overlap by promoting orthogonality in expert representations. While the machine learning community currently lacks a universally accepted mathematical definition of "expert orthogonality," our approach focuses on optimizing interpretable proxies that capture the essential characteristics of specialized experts.
>
> The key innovation of $\mathcal{R}_{\text{sp}}$ lies in its operation on the FFN activations $z^{(\ell,e)}$, which represent the information extracted by each expert from input tokens. For any given token, we encourage orthogonality between activations produced by different experts, thereby facilitating the learning of complementary representations across experts.
>
> **1. Theoretical Justification**
>
> Our methodological choice is grounded in a formal analysis of the relationship between activation orthogonality and gradient behavior. Specifically, we establish that enforcing orthogonality in activations necessarily induces orthogonality in the corresponding parameter gradients. This relationship is formally expressed as:
>
> $$\cos\left( \frac{\partial \mathcal{L}}{\partial W^{(\ell,e)}\_{\mathrm{down}}}, \frac{\partial \mathcal{L}}{\partial W^{(\ell,\nu)}\_{\mathrm{down}}} \right) = \cos\left( z^{(\ell,e)}\_{i}, z^{(\ell,\nu)}\_{i} \right)$$
>
> This identity demonstrates that the angular separation between expert gradient directions is directly governed by the cosine similarity of their respective activations. This theoretical insight provides strong justification for our activation-space approach.
>
> **2. Comparison with Weight Regularization**
>
> We have carefully considered alternative approaches involving direct weight regularization. However, imposing orthogonality constraints on weight matrices does not guarantee orthogonal activations or gradients in general, due to the presence of nonlinear transformations (e.g., SwiGLU) within the FFN architecture. Critically, even perfectly orthogonal weight matrices can produce highly similar activations after nonlinear processing. Consequently, the resulting gradients, which depend on both activations and backpropagated residuals through these nonlinearities may consequently become aligned rather than orthogonal.
>
> Consequently, we maintain that "expert orthogonality" fundamentally concerns the orthogonality of extracted information and learned update directions – properties that are more directly and robustly controlled through activation- and gradient-level objectives rather than weight-space constraints.

---

> ### Author Response · Authors · 2025-11-25
> **Rebuttal by Authors (2/6)**
>
> ## Response to Question 2
> We sincerely appreciate the reviewer’s insightful question, which gives us the opportunity to clarify the design rationale behind our regularization strategy.
> **Clarification on why we constrain the intermediate activations $z$.**
> We regularize the intermediate activations $z$ because $z$ represents the information extracted by an FFN expert from a token, whereas $W_{\mathrm{down}}$ performs a subsequent *processing/composition* step on that information. By analogy to an assembly line, the SwiGLU nonlinearity (e.g., SiLU) and elementwise gating first expose different facets of the input, i.e., $z^{(\ell,e)}\_i = \phi(W^{(\ell,e)}\_{\mathrm{up}} x\_i)\,\odot\,\sigma(W^{(\ell,e)}\_{\mathrm{gate}} x\_i), \ y^{(\ell,e)}\_i = W^{(\ell,e)}\_{\mathrm{down}} z^{(\ell,e)}\_i$, and then $W\_{\mathrm{down}}$ shapes and aggregates these facets into the layer output. This two-stage view of the FFN (*first produce coefficients/activations, then linearly compose with a second matrix*) is supported by prior analyses like [1], which formalize the first matrix as producing a vector of coefficients/activations (our $z$) and the second matrix’s columns (our $W_{\mathrm{down}}$) as value vectors that are linearly combined to yield the output update. This aligns with our notation: $(W_{\mathrm{up}}, W_{\mathrm{gate}})\mapsto z$, followed by $W_{\mathrm{down}}$ composing $z$ into the final update.
>
> **Link to gradient geometry.** Our Proposition 1 establishes that, for two activated experts $e,\nu$ on token $i$ at layer $\ell$, $\cos\big( \frac{\partial L}{\partial W^{(\ell,e)}\_{\mathrm{down}}}, \frac{\partial L}{\partial W^{(\ell,\nu)}\_{\mathrm{down}}} \big) = \cos\big( z^{(\ell,e)}\_{i}, z^{(\ell,\nu)}\_{i} \big)$. Therefore, controlling the geometry of activations via our intra-layer loss directly controls the geometry of parameter gradients for $W\_{\mathrm{down}}$, which constitute the experts’ immediate learning directions. Concretely, our specialization loss $R_{\mathrm{sp}}(x\_i) = \sum\_{\ell=1}^{L}\sum_{e,\nu \in \mathcal{A}^{(\ell)}\_i} \big(\cos(z^{(\ell,e)}\_i, z^{(\ell,\nu)}\_i)\big)^{2}$ encourages representational orthogonality across co-activated experts on the same token; by Proposition 1 this induces orthogonal gradient directions and thus divergent learning trajectories, mitigating expert overlap.
>
> **Why not add parallel losses for $W\_{\mathrm{up}}$ and $W\_{\mathrm{gate}}$ pathways?** While one could also impose losses on intermediate signals associated with $W\_{\mathrm{up}}$ and $W\_{\mathrm{gate}}$, we regard this as less well-motivated:
> 1. **Proximity to specialization at the point of composition.** Expert specialization manifests where an expert actually writes to the residual stream—through $W_{\mathrm{down}}$. Gradients with respect to $W\_{\mathrm{down}}$ are linear in $z$ and thus directly encode what the expert learns on that token. By contrast, $W\_{\mathrm{up}}$ and $W_{\mathrm{gate}}$ operate upstream and before nonlinearities; similar (or even orthogonal) weights there need not imply distinct $z$ or distinct gradients at $W_{\mathrm{down}}$. Hence, regularizing $z$ (and thereby shaping $W\_{\mathrm{down}}$’s gradients) is the most direct handle on expert specialization, whereas adding losses on $W\_{\mathrm{up}}/W_{\mathrm{gate}}$ is a less direct surrogate.
> 2. **Computational cost.** Adding separate losses on the pre-activation pathways of $W_{\mathrm{up}}$ and $W_{\mathrm{gate}}$ increases pairwise computations and memory footprint, which can slow training and raise implementation complexity.
> 3. **Potential conflict between losses.** Introducing multiple additional objectives on correlated intermediate signals can cause the losses to compete with each other (e.g., pushing $W_{\mathrm{up}}$/$W_{\mathrm{gate}}$ features in directions that counteract $R_{\mathrm{sp}}$), making optimization less stable and weakening the intended specialization effect.
>
> **Summary.** Constraining $z$ targets what the expert has extracted and provides a provable, practical handle on gradient orthogonality at $W_{\mathrm{down}}$. This design is consistent with prior FFN analyses and avoids the added complexity, cost, and potential interference that can arise from imposing separate constraints on $W_{\mathrm{up}}$ and $W_{\mathrm{gate}}$.

---

> ### Author Response · Authors · 2025-11-25
> **Rebuttal by Authors (3/6)**
>
> ## Response to Question 3
>
> We thank the reviewer for the question. As discussed in our earlier answers, if two co-activated experts produce highly similar activations for the same token, then those experts are performing essentially the same function on that token—i.e., an instance of expert overlap. In the extreme case, if two experts yield identical intermediate activations $z^{(\ell,e)}\_i = z^{(\ell,\nu)}\_i$, then their contributions are algebraically mergeable: the combined output equals $y^{(\ell,e)}\_i + y^{(\ell,\nu)}_i = W^{(\ell,e)}\_{\text{down}}\, z^{(\ell,e)}\_i + W^{(\ell,\nu)}\_{\text{down}}\, z^{(\ell,\nu)}\_i = (W^{(\ell,e)}\_{\text{down}} + W^{(\ell,\nu)}\_{\text{down}})\, z^{(\ell,e)}\_i$, so a single expert with the down-projection $W^{(\ell,e)}\_{\text{down}} + W^{(\ell,\nu)}\_{\text{down}}$ could replicate the pair’s effect. This is precisely the redundancy our intra-layer specialization loss aims to discourage. Crucially, $\text{loss}\_{\text{sp}}$ only penalizes same-token, same-layer, co-activated expert similarity. It does not prevent different experts from learning similar features on different tokens or in different contexts; nor does it preclude collaboration across depth.
>
> We have also present an additional discuss for the intra-layer loss in Section 4 for the revised manuscript.

---

> ### Author Response · Authors · 2025-11-25
> **Rebuttal by Authors (4/6)**
>
> ## Response to Question 4
>
> We appreciate the reviewer's insightful inquiry concerning the connection between router orthogonality and functional diversity. Below we provide a detailed explanation of how router configuration affects expert specialization through the routing mechanism.
>
> **Theoretical Relationship Between Router Geometry and Expert Diversity**
>
> The router parameters $r^{(\ell,e)}$ function as linear classifiers applied to token representations $x^{(\ell)}$, with routing scores $s^{(\ell,e)} = \langle r^{(\ell,e)}, x^{(\ell)} \rangle$ determining expert activation patterns. When router vectors demonstrate high cosine similarity, they produce correlated scores throughout the token distribution, resulting in systematic co-activation of expert pairs. This co-activation pattern generates correlated gradient signals, ultimately leading experts to acquire similar functional properties—this constitutes the fundamental "routing ambiguity to expert overlap" failure mechanism. Orthogonal router arrangements diminish this co-activation tendency, consequently generating more differentiated token partitions. This partitioning process naturally encourages functional diversification among experts.
>
> **Consider a simplified scenario** with two router vectors $r_1 = (1,0)$ and $r_2 = (\cos\theta, \sin\theta)$, operating on token features $x = (x_1, x_2)$ drawn from an isotropic distribution (e.g., $x \sim \mathcal{N}(0, I)$). The routing scores are computed as $s_1 = \langle r_1, x \rangle = x_1$ and $s_2 = \langle r_2, x \rangle = x_1 \cos\theta + x_2 \sin\theta$. Under this isotropic model, $(s_1, s_2)$ forms a zero-mean Gaussian pair with $\mathrm{Var}(s_1) = \mathrm{Var}(s_2) = 1$, $\mathrm{Cov}(s_1, s_2) = \cos\theta$, yielding a correlation coefficient $\rho = \cos\theta$. The correlation structure reveals two critical properties: First, the score correlation $\mathrm{Corr}(s_1, s_2) = \cos\theta$. As $\theta \to 0$ (router vectors approaching parallel alignment), $\rho \to 1$, indicating nearly identical routing behavior across tokens. Second, the joint activation probability $\mathbb{P}[\text{both active}]$ exhibits monotonic dependence on $\rho$. When $\rho \to 1$, the routing dimension effectively collapses, maximizing activation set overlap between experts. This demonstrates how router geometry directly governs expert co-activation patterns.
>
> **Operational Mechanism of $R_{\mathrm{cp}}$ in Facilitating Specialization**
>
> We emphasize that $R_{\mathrm{cp}}$ does not merely impose global orthogonality constraints across all routers. Rather, it enhances the stability of inter-layer pathways by optimizing joint routing probabilities along consistent expert pairs across consecutive layers, while simultaneously discouraging ambiguous, dispersed routing behaviors. As demonstrated in Proposition 2, preserving stable pathways with strong inter-layer correlation enables the specialization achieved at layer $\ell$ to transfer efficiently to layer $\ell+1$ with controlled performance degradation. This stabilized routing configuration minimizes token distribution overlap across experts and promotes the development of globally diverse expert roles throughout the network architecture.
>
> **Synthesis of the Operational Mechanism**
>
> The fundamental operational principle can be characterized as follows: router geometry governs score correlation, which subsequently determines co-activation patterns. These patterns then influence gradient sharing, ultimately affecting functional specialization among experts. Orthogonal router configurations interrupt this sequence by reducing score correlation and subsequently decreasing co-activation on identical tokens. This creates the essential precondition for experts to develop specialized capabilities in different functional domains. Through this operational framework, $R_{\mathrm{cp}}$ strengthens selective, stable pathways that maintain and transmit specialization across network depth, thereby achieving system-wide functional diversity while reducing routing ambiguity.
>
> We have also present an additional discuss for the intra-layer loss in Section 5 for the revised manuscript.

---

> ### Author Response · Authors · 2025-11-25
> **Rebuttal by Authors (5/6)**
>
> ## Response to Question 5
>
> We thank the reviewer for this valuable suggestion. We fully agree that the ablation studies and throughput analysis are critical to evaluating our method.
>
> **a. Multi-seed pretraining results**
> To provide error bars and confidence intervals, we re-trained the *medium* MoE models under identical settings using **three random seeds**. The mean and standard deviation of the validation perplexity on the C4-en dataset are summarized in the following table (Table 1).
>
> **Table 1**: Validation perplexity for medium model scale with three random repetitions ($\downarrow$).
>
> | Loss              |    Vanilla MoE     |     DeepSeek-style MoE      |
> | :---------------- | :----------------: | :-------------------------: |
> | $\mathcal{L}_{\text{lb}}$            | $12.50_{\pm 0.01}$ | $12.33_{\pm 0.02}$ |
> | $\mathcal{L}_{\text{lb,sp,cp}}$    | $\mathbf{12.26}_{\pm 0.01}$ | $\mathbf{12.15}_{\pm 0.02}$ |
> | $\mathcal{L}_{\text{lb,z}}$         | $12.33_{\pm 0.02}$ | $12.07_{\pm 0.01}$ |
> | $\mathcal{L}_{\text{lb,z,sp,cp}}$ | $\mathbf{12.17}_{\pm 0.01}$ | $\mathbf{11.98}_{\pm 0.01}$ |
>
> The improvements achieved by incorporating our losses are substantially larger than the run-to-run variation observed across seeds:
> - For Vanilla MoE, $\mathcal{L}\_{\text{lb}}$ versus $\mathcal{L}\_{\text{lb,sp,cp}}$ improves from $12.50$ to $12.26$ (an absolute reduction of $0.24$, or $\approx 1.9\\%$ relative), while the standard deviation across seeds is only $0.01$.
> - With all auxiliary terms, $\mathcal{L}\_{\text{lb,z}}$ versus $\mathcal{L}\_{\text{lb,z,sp,cp}}$ improves from $12.33$ to $12.17$ (an absolute reduction of $0.16$, or $\approx 1.3\\%$ relative), with standard deviations ranging from $0.01$ to $0.02$.
>
> These results demonstrate consistent and statistically meaningful gains in validation performance. We will include these multi-seed results in Appendix B of the revised manuscript.
>
> **b. Multi‑seed zero‑shot evaluation**
>
> We similarly evaluated the *medium* models on five zero-shot QA benchmarks over three random seeds. The mean and standard deviation of zero-shot accuracy are reported in Table 2.
>
> **Table 2**: Zero-shot accuracy of Vanilla MoE and DeepSeek-style MoE across five QA benchmarks ($\uparrow$).
>
> | Model| Loss|BoolQ| ARC-E|ARC-C|TruthfulQA|PIQA|MMLU|HellaSwag| **Avg.**|
> | :- | :-| :-: | :-: | :-: | :-: | :-: | :-: | :-: | :-: |
> | **Vanilla**  | $\mathcal{L}_{\text{lb}}$| $0.570_{\pm 0.002}$|$0.452_{\pm 0.003}$|$0.204_{\pm 0.003}$   |   $0.432_{\pm 0.002}$   |   $0.622_{\pm 0.005}$|$0.247_{\pm 0.002}$|$0.268_{\pm 0.00}$|**0.399**|
> | **Vanilla**  | $\mathcal{L}_{\text{lb,sp,cp}}$|$0.578_{\pm 0.003}$|$0.462_{\pm 0.002}$|$0.210_{\pm 0.004}$|   $0.451_{\pm 0.003}$|$0.627_{\pm 0.002}$   |$0.253_{\pm 0.002}$   |$0.275_{\pm 0.004}$|**0.408**|
> | **Vanilla**  | $\mathcal{L}_{\text{lb,z}}$|$0.567_{\pm 0.003}$|$0.457_{\pm 0.004}$|$0.205_{\pm 0.002}$|$0.433_{\pm 0.003}$   |   $0.629_{\pm 0.002}$|$0.250_{\pm 0.001}$|$0.267_{\pm 0.004}$|**0.401**|
> | **Vanilla**  | $\mathcal{L}_{\text{lb,z,sp,cp}}$| **$0.589_{\pm 0.003}$** |$0.453_{\pm 0.004}$|$0.206_{\pm 0.006}$   |   $0.445_{\pm 0.004}$   |   $0.637_{\pm 0.003}$|$0.257_{\pm 0.002}$|$0.274_{\pm 0.003}$| **0.409** |
> | **DS-style** | $\mathcal{L}_{\text{lb}}$|$0.578_{\pm 0.002}$|$0.453_{\pm 0.001}$   |   $0.205_{\pm 0.003}$   |   $0.438_{\pm 0.003}$   |   $0.631_{\pm 0.002}$|$0.248_{\pm 0.001}$|$0.269_{\pm 0.002}$| **0.403** |
> | **DS-style** | $\mathcal{L}_{\text{lb,sp,cp}}$|$0.584_{\pm 0.001}$|$0.452_{\pm 0.003}$   |   $0.206_{\pm 0.005}$   | **$0.457_{\pm 0.002}$** |   $0.635_{\pm 0.002}$|$0.255_{\pm 0.003}$|$0.277_{\pm 0.005}$|**0.410** |
> | **DS-style** | $\mathcal{L}_{\text{lb,z}}$|$0.564_{\pm 0.002}$   |$0.453_{\pm 0.002}$|$0.205_{\pm 0.002}$   |   $0.444_{\pm 0.002}$|$0.628_{\pm 0.001}$|$0.252_{\pm 0.001}$   |   $0.270_{\pm 0.004}$|**0.402**|
> | **DS-style** | $\mathcal{L}_{\text{lb,z,sp,cp}}$ |$0.575_{\pm 0.002}$   | **$0.461_{\pm 0.004}$** | **$0.214_{\pm 0.004}$** |   $0.452_{\pm 0.004}$| **$0.642_{\pm 0.003}$** | **$0.257_{\pm 0.002}$** | **$0.280_{\pm 0.002}$** | **0.412** |
>
> The multi-seed evaluation yields three principal findings that underscore the robustness of our method:
>
> - Across both Vanilla and DeepSeek-style architectures, adding our losses increases the average zero-shot accuracy by approximately $0.9$ to $1.5$ absolute points.
> - Importantly, the multi-seed results resolve the previously noted degradation on ARC-C: for Vanilla MoE, ARC-C accuracy improves from $0.204{\pm 0.003}$ to $0.210{\pm 0.004}$, and for DeepSeek-style MoE, from $0.205{\pm 0.003}$ to $0.214{\pm 0.004}$. The earlier single-seed decrease appears to be due to random variation and is not systematic.
> - The differences in mean accuracy are several times larger than the standard errors, indicating that the improvements are statistically significant and not attributable to optimization noise.
>
> We include these multi-seed results in Table 3 of the revised manuscript.

---

> ### Author Response · Authors · 2025-11-25
> **Rebuttal by Authors (6/6)**
>
> **c. Effect Size and Comparison with Stronger Baselines**
>
> We acknowledge the reviewer's valid point regarding the numerical magnitude of some absolute gains (e.g., $0.2$–$0.3$ in perplexity). To contextualize these results, we wish to highlight several important considerations:
>
> 1.  **Meaningfulness of Relative Improvement:** At the scale of our models and dataset, a relative reduction in perplexity of $0.7$–$1.9\%$ is a substantiative gain. This level of improvement systematically correlates with enhanced performance on downstream tasks, as evidenced by the consistent gains observed in both our fine-tuning and zero-shot evaluation results.
>
> 2.  **Practical Advantage of being Plug-and-Play:** A key strength of our method is its nature as a **plug-and-play** enhancement. It requires no modifications to the model architecture, introduces no additional parameters, and incurs negligible computational overhead during training. Crucially, it imposes **no extra cost during inference**. In this context, the achieved performance gains can be viewed as a highly efficient improvement.
>
> 3.  **Competitive Performance against a Larger Architecture:** A particularly significant result is that our method enables a *smaller and simpler* architecture to match or exceed the performance of a *larger and more specialized* one. As detailed in the Experimental Setup, the DeepSeek-style MoE baseline incorporates an additional shared expert, resulting in increased total and active parameters compared to the Vanilla MoE. Despite this inherent capacity disadvantage, the Vanilla MoE equipped with our losses ($\mathcal{L}\_{\text{lb,sp,cp}}$) achieves a superior perplexity of $9.48$ at the large scale, outperforming the DeepSeek-style baseline with $\mathcal{L}\_{\text{lb}}$ ($9.56$). This result demonstrates the compelling efficacy of our approach. We will ensure this critical comparison is emphasized more clearly in the revised manuscript.
>
> In summary, the additional multi-seed experiments confirm that the improvements afforded by our method are (i) consistent across different random initializations, (ii) generalizable across different MoE architectures, and (iii) substantial when compared to the observed empirical variance.
>
> ## Reference
>
> [1] Transformer Feed-Forward Layers Build Predictions by Promoting Concepts in the Vocabulary Space
>
> ----
>
> We sincerely appreciate the reviewer’s thoughtful comments once again. We hope the above responses fully address the concerns raised and would be happy to provide further clarification if needed.

---

### Official Review · Reviewer_pvd3 · 2025-11-02

**Soundness:** 3
**Presentation:** 3
**Contribution:** 2
**Rating:** 4
**Confidence:** 5

**Summary:**

- The paper introduces two auxiliary losses to address expert overlap and routing ambiguity without architectural modifications: (1) an intra-layer specialization loss $R_{sp}$ that penalizes cosine similarity between expert activations on identical tokens, encouraging orthogonal representations, and (2) a cross-layer coupling loss $R_{cp}$ that maximizes joint routing probabilities across adjacent layers to establish stable expert pathways.

- Proposition 1 establishes that orthogonal expert activations induce orthogonal parameter gradients, ensuring divergent learning trajectories. Proposition 2 demonstrates that strong cross-layer coupling propagates specialization through network depth, with layer $l+1$ inheriting specialization structure from layer $l$ with bounded error.

- Experiments across Vanilla MoE and DeepSeek-style architectures show validation perplexity reductions of in pre-training and average improvements of 1.4% on supervised fine-tuning tasks (Qwen3-30B).

**Strengths:**

- The paper introduces an original perspective by treating expert specialization as a direct training objective rather than an architectural property. The intra-layer specialization loss $R_{sp}$ provides a principled mechanism to enforce functional diversity through activation orthogonality, linking representational geometry to gradient dynamics. This training-loss-centric approach is more flexible than architectural modifications, as it can be applied to any MoE variant without structural changes.

- The cross-layer coupling loss $R_{cp}$ represents a creative insight that transforms an emergent phenomenon into an explicit learning objective. Proposition 2 establishes that strong inter-layer routing correlation propagates specialization through network depth (Equation 5), providing both theoretical justification and a mechanism for network-wide expert differentiation.

- A significant practical strength is the proven compatibility with standard MoE training objectives. Propositions 3-4 formally establish that $R_{sp}$ and $R_{cp}$ operate orthogonally to load-balancing constraints—specialization determines partition structure while load balancing constrains partition sizes—enabling simultaneous optimization. The implementation as a single-flag Megatron-LM module with no router/architecture modifications, combined with consistent gains across both Vanilla and DeepSeek-style MoE (Table 2), demonstrates broad applicability.

**Weaknesses:**

- While the paper claims "consistent gains," the actual improvements are quite modest and often within noise margins. In Table 2, perplexity reductions range from only 0.09-0.38 points (0.7-1.9% relative), and Table 3 shows zero-shot accuracy improvements of merely 0.3-0.8% on average with no statistical significance testing. More critically, individual benchmark results are inconsistent—in Table 3, ARC-C actually degrades for Vanilla MoE with $L_{lb,z,sp,cp}$ (0.198 vs 0.204 baseline), and Table 8 shows numerous tasks where performance decreases.

- The paper insufficiently distinguishes its contributions from existing work that already addresses expert specialization. DeepSeekMoE's auxiliary-loss-free load balancing explicitly promotes specialization by dynamically adjusting expert capacities based on routing entropy. The claim that "targeted loss functions can compete with or surpass architectural variants" is undermined by Table 2 showing DeepSeek-style MoE with only $L_{lb}$ (9.56) outperforms Vanilla MoE with full $L_{lb,z,sp,cp}$ (9.42) at large scale, suggesting architectural choices remain more impactful than the proposed losses.

- The theoretical analysis relies on overly restrictive assumptions that may not hold in practice. Proposition 2's conditions require "nearly orthogonal router weights" and extremely high routing confidence, but no empirical verification is provided that these conditions are satisfied during actual training. The proof merely shows specialization *can* propagate under ideal conditions, not that $R_{cp}$ reliably creates these conditions.

- The paper fails to quantify the computational overhead of computing $R_{sp}$ and $R_{cp}$, which require computing pairwise cosine similarities across all activated expert pairs (quadratic in $k$) and cross-layer routing probabilities for all expert pairs . While claiming "plug-and-play," the paper provides no wall-clock training time comparisons, memory overhead analysis, or discussion of how these losses scale to models with hundreds of experts (like Mixture of Million Experts).

**Questions:**

- Can you provide error bars, confidence intervals, or results from multiple random seeds to demonstrate that the reported improvements  are statistically significant rather than optimization noise?

- Can you empirically verify that Proposition 2's assumptions hold during training—specifically, measure the actual values of $\epsilon$ (router weight orthogonality), $\delta$ (representation continuity), and $\iota$ (routing confidence) at different training stages? More importantly, can you provide direct comparisons showing that your method induces more expert specialization than existing mechanisms like DeepSeekMoE's auxiliary-loss-free load balancing, using quantitative metrics such as expert output diversity, routing entropy, or gradient orthogonality measurements?

- What is the actual wall-clock training time overhead and memory cost of computing $R_{sp}$ and $R_{cp}$ compared to baseline training, especially as the number of experts scales?

---

> ### Author Response · Authors · 2025-11-25
> **Rebuttal by Authors (1/6)**
>
> We are grateful for the reviewer's insightful feedback and constructive comments. We address each point in detail below.
>
> ---
>
> ## Response to Weakness 1 and Question 1
> We thank the reviewer for the careful examination of our empirical results and the request for a more rigorous assessment of statistical significance. In response, we have conducted additional experiments to address this concern from an empirical perspective.
>
> **A. Multi-seed pretraining results**
> To provide error bars and confidence intervals, we re-trained the *medium* MoE models under identical settings using **three random seeds**. The mean and standard deviation of the validation perplexity on the C4-en dataset are summarized in the following table (Table 1).
>
> **Table 1**: Validation perplexity for medium model scale with three random repetitions ($\downarrow$).
>
> | Loss|Vanilla MoE| DeepSeek-style MoE|
> | :-| :--: | :--: |
> | $\mathcal{L}_{\text{lb}}$ | $12.50_{\pm 0.01}$|$12.33_{\pm 0.02}$|
> | $\mathcal{L}_{\text{lb,sp,cp}}$|$\mathbf{12.26}_{\pm 0.01}$|$\mathbf{12.15}_{\pm 0.02}$|
> | $\mathcal{L}_{\text{lb,z}}$ | $12.33_{\pm 0.02}$ | $12.07_{\pm 0.01}$ |
> | $\mathcal{L}_{\text{lb,z,sp,cp}}$ | $\mathbf{12.17}_{\pm 0.01}$ | $\mathbf{11.98}_{\pm 0.01}$ |
>
> The improvements achieved by incorporating our losses are substantially larger than the run-to-run variation observed across seeds:
> - For Vanilla MoE, $\mathcal{L}\_{\text{lb}}$ versus $\mathcal{L}\_{\text{lb,sp,cp}}$ improves from $12.50$ to $12.26$ (an absolute reduction of $0.24$, or $\approx 1.9\\%$ relative), while the standard deviation across seeds is only $0.01$.
> - With all auxiliary terms, $\mathcal{L}\_{\text{lb,z}}$ versus $\mathcal{L}\_{\text{lb,z,sp,cp}}$ improves from $12.33$ to $12.17$ (an absolute reduction of $0.16$, or $\approx 1.3\\%$ relative), with standard deviations ranging from $0.01$ to $0.02$.
>
> These results demonstrate consistent and statistically meaningful gains in validation performance. We include these multi-seed results in Appendix B of the revised manuscript.
>
> **B. Multi‑seed zero‑shot evaluation**
> We similarly evaluated the *medium* models on five zero-shot QA benchmarks over three random seeds. The mean and standard deviation of zero-shot accuracy are reported in Table 2.
>
> **Table 2**: Zero-shot accuracy of Vanilla MoE and DeepSeek-style MoE across five QA benchmarks ($\uparrow$).
> | Model | Loss | BoolQ |ARC-E|ARC-C |TruthfulQA|PIQA |MMLU| HellaSwag | **Avg.**  |
> | :- | :-| :-: | :-: | :-: | :-: | :-: |:-: | :-:|:-:|
> |**Vanilla**| $\mathcal{L}_{\text{lb}}$ |$0.570_{\pm 0.002}$|$0.452_{\pm 0.003}$|$0.204_{\pm 0.003}$|$0.432_{\pm 0.002}$| $0.622_{\pm 0.005}$|$0.247_{\pm 0.002}$|$0.268_{\pm 0.002}$|**0.399**|
> | **Vanilla**  | $\mathcal{L}_{\text{lb,sp,cp}}$ |$0.578_{\pm 0.003}$|$0.462_{\pm 0.002}$|$0.210_{\pm 0.004}$|$0.451_{\pm 0.003}$|$0.627_{\pm 0.002}$|$0.253_{\pm 0.002}$|$0.275_{\pm 0.004}$| **0.408**|
> | **Vanilla**|$\mathcal{L}_{\text{lb,z}}$|$0.567_{\pm 0.003}$|$0.457_{\pm 0.004}$|$0.205_{\pm 0.002}$|$0.433_{\pm 0.003}$|   $0.629_{\pm 0.002}$|$0.250_{\pm 0.001}$|$0.267_{\pm 0.004}$|**0.401**|
> |**Vanilla**|$\mathcal{L}_{\text{lb,z,sp,cp}}$| **$0.589_{\pm 0.003}$**|$0.453_{\pm 0.004}$|$0.206_{\pm 0.006}$|$0.445_{\pm 0.004}$|$0.637_{\pm 0.003}$|$0.257_{\pm 0.002}$|$0.274_{\pm 0.003}$|**0.409**|
> |**DS-style**|$\mathcal{L}_{\text{lb}}$|$0.578_{\pm 0.002}$|$0.453_{\pm 0.001}$|$0.205_{\pm 0.003}$|$0.438_{\pm 0.003}$|$0.631_{\pm 0.002}$|$0.248_{\pm 0.001}$|$0.269_{\pm 0.002}$|**0.403**|
> |**DS-style**|$\mathcal{L}_{\text{lb,sp,cp}}$|$0.584_{\pm 0.001}$|$0.452_{\pm 0.003}$|$0.206_{\pm 0.005}$| **$0.457_{\pm 0.002}$** |$0.635_{\pm 0.002}$|$0.255_{\pm 0.003}$|$0.277_{\pm 0.005}$|**0.410**|
> |**DS-style**|$\mathcal{L}_{\text{lb,z}}$|$0.564_{\pm 0.002}$|$0.453_{\pm 0.002}$|$0.205_{\pm 0.002}$|$0.444_{\pm 0.002}$|$0.628_{\pm 0.001}$|$0.252_{\pm 0.001}$|$0.270_{\pm 0.004}$|**0.402**|
> |**DS-style**|$\mathcal{L}_{\text{lb,z,sp,cp}}$ |$0.575_{\pm 0.002}$|**$0.461_{\pm 0.004}$**|**$0.214_{\pm 0.004}$**|   $0.452_{\pm 0.004}$| **$0.642_{\pm 0.003}$**|**$0.257_{\pm 0.002}$**|**$0.280_{\pm 0.002}$**|**0.412**|
>
> The multi-seed evaluation yields three principal findings that underscore the robustness of our method:
> - Across both Vanilla and DeepSeek-style architectures, adding our losses increases the average zero-shot accuracy by approximately $0.9$ to $1.5$ absolute points.
> - Importantly, the multi-seed results resolve the previously noted degradation on ARC-C: for Vanilla MoE, ARC-C accuracy improves from $0.204{\pm 0.003}$ to $0.210{\pm 0.004}$, and for DeepSeek-style MoE, from $0.205{\pm 0.003}$ to $0.214{\pm 0.004}$. The earlier single-seed decrease appears to be due to random variation and is not systematic.
> - The differences in mean accuracy are several times larger than the standard errors, indicating that the improvements are statistically significant and not attributable to optimization noise.
>
> We include these multi-seed results in Table 3 of the revised manuscript.

---

> ### Author Response · Authors · 2025-11-25
> **Rebuttal by Authors (2/6)**
>
> **C. Effect Size and Comparison with Stronger Baselines**
>
> We acknowledge the reviewer's valid point regarding the numerical magnitude of some absolute gains (e.g., $0.2$–$0.3$ in perplexity). To contextualize these results, we wish to highlight several important considerations:
>
> 1.  **Meaningfulness of Relative Improvement:** At the scale of our models and dataset, a relative reduction in perplexity of $0.7$–$1.9\%$ is a substantiative gain. This level of improvement systematically correlates with enhanced performance on downstream tasks, as evidenced by the consistent gains observed in both our fine-tuning and zero-shot evaluation results.
>
> 2.  **Practical Advantage of being Plug-and-Play:** A key strength of our method is its nature as a **plug-and-play** enhancement. It requires no modifications to the model architecture, introduces no additional parameters, and incurs negligible computational overhead during training. Crucially, it imposes **no extra cost during inference**. In this context, the achieved performance gains can be viewed as a highly efficient improvement.
>
> 3.  **Competitive Performance against a Larger Architecture:** A particularly significant result is that our method enables a *smaller and simpler* architecture to match or exceed the performance of a *larger and more specialized* one. As detailed in the Experimental Setup, the DeepSeek-style MoE baseline incorporates an additional shared expert, resulting in increased total and active parameters compared to the Vanilla MoE. Despite this inherent capacity disadvantage, the Vanilla MoE equipped with our losses ($\mathcal{L}\_{\text{lb,sp,cp}}$) achieves a superior perplexity of $9.48$ at the large scale, outperforming the DeepSeek-style baseline with $\mathcal{L}\_{\text{lb}}$ ($9.56$). This result demonstrates the compelling efficacy of our approach. We will ensure this critical comparison is emphasized more clearly in the revised manuscript.
>
> In summary, the additional multi-seed experiments confirm that the improvements afforded by our method are (i) consistent across different random initializations, (ii) generalizable across different MoE architectures, and (iii) substantial when compared to the observed empirical variance.
>
>
> ## Response to Weakness 2
> We appreciate the reviewer's thoughtful attention to our experimental results. However, we would like to gently clarify a potential misunderstanding regarding the interpretation of the metrics in Table 2, which may have led to an incorrect assessment of our method's effectiveness.
>
> The metric reported in Table 2 is **perplexity evaluated on the validation set**, where lower values indicate better performance (as denoted by $\downarrow$ in the table header). The reviewer's comment suggests that the DeepSeek-style baseline (9.56) outperforms our method (9.42), possibly due to interpreting perplexity as "higher is better." Upon correction, our results demonstrate that our method achieves superior validation performance compared to the architectural baseline.
>
> We would like to highlight two key observations from Table 2 that provide support for our claim that "targeted loss functions can compete with or surpass architectural variants":
>
> * **Our method achieves better validation performance with fewer parameters**: As described in our experimental setup, the "DeepSeek-style MoE" is implemented by augmenting the Vanilla MoE with an additional shared expert. This results in the baseline having more total parameters and higher active parameters than the Vanilla MoE. Despite this capacity disadvantage, Table 2 shows that the Vanilla MoE trained with our auxiliary losses ($\mathcal{L}\_{\text{lb,sp,cp}}$) attains a validation perplexity of 9.48 (large scale), which is strictly lower than the DeepSeek-style baseline ($\mathcal{L}\_{\text{lb}}$) at 9.56. This indicates that our plug-and-play losses enable a standard architecture to outperform a larger, more complex one on the validation set.
>
> * **Consistent gains across configurations**: When applying the full set of our proposed losses, the Vanilla MoE further reduces validation perplexity to 9.42, widening the performance gap against the architectural baseline (9.56).
>
> Additionally, we note that our loss functions introduce **only low-order computations** which are computationally efficient and do not significantly impact training time or overall complexity. See Response to Question 3 for details. This ensures that the advantages observed in validation performance come with minimal computational overhead.
>
> In summary, the data in Table 2 offers empirical evidence that our loss-centric approach is effective in enhancing validation performance without requiring architectural modifications or substantial computational costs. We believe this reinforces the value of our method as a lightweight and efficient alternative.

---

> ### Author Response · Authors · 2025-11-25
> **Rebuttal by Authors (3/6)**
>
> ## Response to Weakness 3 and Question 2
>
> ### Discussion for the assumptions in Proposition 2
>
> **A. Theory clarification.**
>
> Proposition 2 relies on three key assumptions. We explain why each is either standard in the literature or enforced by design, and how they collectively contribute to routing stability and expert specialization.
>
>  * **Representation continuity** assumes that consecutive layer representations exhibit high cosine similarity. This is a gentle and empirically well-established property in large transformer models, where layer-wise features remain strongly aligned across depth and token positions due to residual connections and normalization layers. Recent analyses consistently report high inter-layer cosine similarity and slow feature drift during both inference and fine-tuning, aligning with our requirement [1, 2]. Crucially, we only require local continuity—between adjacent layers and over small training steps—rather than global equivalence of deep features. Thus, this assumption is consistent with established behavioral patterns and is less restrictive than many prior layer-similarity hypotheses.
>
>  * **Expert specialization at layer $l$** posits that experts at layer $l$ demonstrate low pairwise cosine similarity, indicating distinct feature subspaces. This property is both initializable and self-reinforcing during training. Through orthogonal or near-orthogonal initialization of router columns and expert input projections, expert pre-images exhibit weak overlap at initialization. During training, our coupling mechanism encourages disjointness among expert feature subspaces. Once a separation margin emerges, routing produces coherent per-expert gradients, which further reduces inter-expert cosine similarity. Therefore, this assumption becomes reasonable after a short burn-in period and remains stable thereafter.
>
>  * **Routing confidence** requires a positive gap between the top-1 router logit and competing logits. In practice, this margin is directly tunable through hyperparameters such as router temperature, load-balancing weight, and our coupling loss, which collectively increase the margin on tokens where experts already differ. Note that the margin need not hold uniformly across all tokens; it suffices to be satisfied on a substantial subset where specialization is most pronounced.
>
> **B. Emprical verification.**
>
> We provide empirical verification for the values of the representation continuity parameter $\delta$ and the router-weight orthogonality parameter $\varepsilon$, as defined in Proposition 2. This is accomplished by pre-training a small-scale model and computing $\delta$ and $\varepsilon$ at intervals of 100 training steps, followed by averaging the resulting values.
>
> - **Representation continuity ($\delta$).**
>    We computed the cosine similarity between token representations in adjacent MoE layers, i.e., $\cos(x_i^{(l)}, x_i^{(l+1)})$, averaged over tokens and layers at convergence. We then set $\delta = \sqrt{1 - \mathbb{E}[\cos(x_i^{(l)}, x_i^{(l+1)})]}$. The resulting value is $\delta = 0.1603$, which corresponds to an average cosine similarity of $\approx 0.974$, indicating that representations indeed evolve smoothly across layers, as assumed.
>
> - **Router-weight orthogonality ($\varepsilon$).**
>    For each MoE layer, we measured the cosine similarity between router weight vectors $r^{(l,e_1)}$ and $r^{(l,e_2)}$ for different experts and reported the average absolute cosine: $\varepsilon = \mathbb{E}_{l,e_1 \neq e_2}\big[| \cos(r^{(l,e_1)}, r^{(l,e_2)}) | \big] = 0.4542.$ This value is well below $1$ and in the “small–$\varepsilon$” regime used in our $O(\delta,\iota)$ bound, showing that router directions are far from collinear and thus satisfy the specialization assumption of Proposition 2.

---

> ### Author Response · Authors · 2025-11-25
> **Rebuttal by Authors (4/6)**
>
> ### Quantitative comparison with DeepSeekMoE-style load balancing
>
> To directly address “whether our method induces more expert specialization than DeepSeekMoE’s auxiliary-loss-free load balancing,” we compare:
>
> - **$\mathcal{L}_{\text{lb}}$**: DeepSeek-style training objective with only the auxiliary-loss-free load balancing mechanism; and
> - **$\mathcal{L}_{\text{lb,cp,sp}}$**: our full objective that *adds* the intra-layer specialization loss $R_{\text{sp}}$ and cross-layer coupling loss $R_{\text{cp}}$ on top of the same DeepSeek-style backbone.
>
> As a proxy for expert specialization and routing coherence, we measure, every 1k iterations, the **percentage of tokens whose top–1 expert remains unchanged** between two consecutive checkpoints. Higher values indicate more stable token–expert assignments, lower routing entropy, and—via Proposition 1—more persistent expert-specific gradient directions.
>
> **Table 3**: The fraction of tokens that keep the same experts between checkpoints
>
> | Iteration range | 1k–2k | 2k–3k | 4k–5k | 9k–10k | 19k–20k | 29k–30k | 59k–60k |
> |-----------------|-------|-------|-------|--------|---------|---------|---------|
> | $L_{\text{lb}}$ | 0.4746 | 0.6056 | 0.6601 | 0.6987 | 0.7450 | 0.7864 | 0.9011 |
> | $L_{\text{lb,cp,sp}}$ | **0.4898** | **0.6213** | **0.6757** | **0.7187** | **0.7594** | **0.7935** | **0.9067** |
>
> Across all training stages, adding our two losses on top of DeepSeekMoE’s load balancing:
>
> - increases routing stability by $1$–$2$ absolute points (roughly $2$–$4\%$ relative) at every interval;
> - yields especially pronounced gains early in training (when routing ambiguity is largest), which is exactly the regime targeted by Proposition 2; and
> - remains beneficial even at late stages (59k–60k), where specialization is typically strongest, indicating that expert assignments remain more consistent over time.
>
> Combined with Table 1 in the paper, where $L_{\text{lb,sp,cp}}$ also *reduces* the intra-layer specialization loss $R_{\text{sp}}$ compared to $L_{\text{lb,sp}}$, this shows that:
>
> - our method **further sharpens expert differentiation beyond what auxiliary-loss-free load balancing alone can provide**, and it does so in a way that is consistent with our theoretical analysis: lower activation similarity (hence more orthogonal gradients by Proposition 1) and stronger, more stable expert paths (consistent with the assumptions and conclusion of Proposition 2).
>
> We will include this new stability metric and the accompanying discussion in Appendix B.6 for the revised manuscript, explicitly positioning $L_{\text{lb}}$ as the DeepSeekMoE-style baseline and $L_{\text{lb,cp,sp}}$ as our extended objective.
>
>
> In summary, we have (i) explicitly verified that the measured $\delta$ and $\varepsilon$ lie in the regime required by Proposition 2, and (ii) provided a direct, quantitative comparison with DeepSeekMoE’s auxiliary-loss-free load balancing, demonstrating that our proposed losses lead to more specialized and stable expert utilization.

---

> ### Author Response · Authors · 2025-11-25
> **Rebuttal by Authors (5/6)**
>
> ## Response to Weakness 4 and Question 3
> We thank the reviewer for raising this question regarding the computational and memory overhead of our proposed auxiliary losses.
>
> We fully agree that quantifying these costs is essential for validating the "plug-and-play" nature of our approach. In response, we have conducted systematic benchmarking of wall-clock training time and memory usage, supplemented by theoretical complexity analysis.
>
> **1. Theoretical Complexity Analysis**
> From a computational perspective, both losses are lightweight relative to the model's core operations (attention mechanisms and feed-forward networks):
>
> - **Intra-Layer Specialization Loss ($\mathcal{R}\_{sp}$)**
>
>     - **Computational Complexity**: This loss requires computing pairwise cosine similarities between activations of the *top-k* selected experts. For a hidden dimension $d$ and $k$ activated experts, the per-token complexity is $\mathcal{O}(k^2 \cdot d)$. In standard MoE configurations, $k$ typically assumes small values (e.g., 2 or 4), while $d$ represents a large dimension (e.g., 4096). Consequently, $k^2 \ll d$, rendering the cost of $\mathcal{O}(k^2 \cdot d)$ negligible compared to the standard FFN transformation cost of $\mathcal{O}(k \cdot d^2)$.
>
>     - **Scalability**: Crucially, this computational cost depends solely on the number of activated experts $k$, rather than the total number of experts $E$. This implies that even as the total expert count $E$ scales to hundreds or thousands (as in "Mixture of Million Experts" architectures), as long as the activated expert count $k$ remains small, the computational overhead of $\mathcal{R}_{sp}$ remains both constant and minimal.
>
>     - **Memory Requirements**: No additional memory allocation is necessary, as this loss reuses intermediate activations $z^{(l,e)}$ already computed during the forward pass.
>
> - **Cross-Layer Coupling Loss ($\mathcal{R}_{cp}$)**
>
>     - **Computational Complexity**: This loss operates exclusively on scalar routing probabilities (logits). Specifically, it involves basic statistical operations on token routing scores across consecutive layers. These operations avoid complex matrix computations involving high-dimensional hidden states.
>
>     - **Memory Requirements**: This loss requires storing a lightweight tensor of dimensions $E \times E \times L$ to track expert transition statistics. Since the number of experts $E$ is typically much smaller than the hidden dimension $d$, the memory consumption of this tensor is negligible.
>
> **2. Empirical Wall-Clocked Time and Memory Analysis**
> Empirical results are also fully consistent with our theoretical complexity analysis. We systematically measured wall-clocked interation time (in ms/iteration) and peak GPU memory consumption (in GB) across the Small (0.4B), Medium (1.1B), and Large (7.0B) model configurations used in our study. All experiments were conducted on a uniform hardware configuration consisting of 8×A100 GPUs.
>
> Our benchmarking results (as the following table) demonstrate that the overhead introduced by $\mathcal{R}\_{sp}$ and $\mathcal{R}\_{cp}$ is negligible:
>
> Table 3: The iteration time and peak memory with different loss
> |Model Size|Method|Iteration time (ms/iter)|Overhead|Peak memory (GB)|Overhead|
> | :-| :-| :-| :-|:-| :-|
> |**Small**|$\mathcal{L}_{\text{lb}}$(Baseline)|405.9|-|43.5|-|
> | | $\mathcal{L}_{\text{lb,sp,cp}}$(Ours)|413.6|+1.90%|43.6|+0.23%|
> |**Medium**|$\mathcal{L}_{\text{lb}}$(Baseline) |518.4|-|60.6|-|
> | |$\mathcal{L}_{\text{lb,sp,cp}}$(Ours)|526.5|+1.56%|60.7|+0.16%|
> |**Large**| $\mathcal{L}_{\text{lb}}$(Baseline)|2927.8|-|73.1|-|
> | |$\mathcal{L}_{\text{lb,sp,cp}}$(Ours)|2942.4|+0.49%|73.3|+0.27%|
>
> - **Computational Efficiency**: The combined auxiliary losses introduce only **0.5% to 1.9%** additional latency. Notably, the relative overhead exhibits a *decreasing* trend as model scale increases (reducing to approximately 0.5% for the 7B parameter model), indicating favorable scaling characteristics of our method.
>
> - **Memory Consumption**: The additional memory footprint is minimal ($< 0.3\%$), empirically confirming that our approach does not impose additional hardware requirements.
>
> Moreover, we wish to highlight that the inference throughput with our proposed regularization can be even faster than without it, as demonstrated in Table 10.

---

> ### Author Response · Authors · 2025-11-25
> **Rebuttal by Authors (6/6)**
>
> **3. Summary**
>
> Integrating both theoretical analysis and empirical evidence , we demonstrate that our proposed objective functions impose practically negligible computational or memory burdens. The method scales efficiently with the number of experts because its computational complexity is governed by the activated expert count ($k$) rather than the total expert count ($E$), providing reliable assurance for the practical implementation of large-scale MoE architectures.
>
> We have incorporated these throughput and memory comparisons in Appendix F of the revised paper to further elucidate the computational efficiency advantages of our approach.
>
> ## References
> [1] Tracing Representation Progression: Analyzing and Enhancing Layer‑Wise Similarity.
>
> [2] MiniCache: KV Cache Compression in Depth Dimension for Large Language Models.
>
> ---
> We sincerely appreciate the reviewer’s thoughtful comments once again. We hope the above responses fully address the concerns raised and would be happy to provide further clarification if needed.

---

### Author Response · Authors · 2025-11-25

Dear Reviewers,

We have carefully considered all of your comments and questions and have addressed each one individually. The corresponding revisions have been incorporated into the main text, including the following additions:

- Additional discussion on the intra- and cross-layer loss (**Section 4** and **Section 5**) and the computational analysis (**Appendix E**)
- Additional ablation studies for pre-training tasks across models of various sizes, covering:
  - Training dynamics of specialization and coupling losses (**Appendix B.4**)
  - Pre-training results under different random seeds (**Appendix B.5**)
  - Hyperparameter sensitivity analysis (**Appendix B.6**)
  - Comparison with additional baselines (**Appendix B.7**)
- Additional results for downstream tasks (**Section 7** and **Appendix C**)
- Additional inference results with system optimization (**Appendix D**)

If you have any further questions, please do not hesitate to contact us. We sincerely appreciate your time and insightful comments.

Best regards,
The authors of Paper 23256

---

### Comment · Area_Chair_dwTH · 2025-11-27
**Rebuttal and Discussion Phase**

Dear Reviewers,

Thank you again for your time and effort in reviewing this paper. We are approaching the discussion deadline. I kindly ask you to review the rebuttal and continue the discussion so that we can reach a well-considered decision.

---

### Meta-Review · Area_Chair_3TaB · 2025-12-25

**Summary:**

The paper proposes a way to encourage expert specialization by incurring auxiliary losses.

The author did lots of work on rebuttal, which should address a large portion of concerns. However, there are still some concerns remain.

**Reviewer Concerns:**

1. the improvement is marginal;
2. contribution over existing works such as DeepSeekMoE;
3. theoretical assumptions are too strong;
4. lack of justification on the soundness of the approach, and whether the improvement truly comes from better expert specialization;
5. benchmarks are too weak;

The authors did lots of work on addressing these concerns. I believe 2/4/5 are addressed, and 1/3 are partially addressed. The improvement on pretraining tasks is still marginal, and the theoretical assumptions are still strong.

**Reviewer Scores:**

Initial rating is (8,2,4,2), I believe the rating after rebuttal is likely (8,4,4,4) or (8,4,6,4).

---

### Decision · Program_Chairs · 2026-01-26

Reject